# Mutation rates and fitness consequences of mosaic chromosomal alterations in blood

Caroline J. Watson ⬛ ✉ & Jamie R. Blundell ⬛ ✉

Mosaic chromosomal alterations (mCAs) are common in cancers and can arise decades before diagnosis. A quantitative understanding of the rate at which these events occur, and their functional consequences, could improve cancer risk prediction and our understanding of somatic evolution. Using mCA clone size estimates from the blood of approximately 500,000 UK Biobank participants, we estimate mutation rates and fitness consequences of acquired gain, loss and copy-neutral loss of heterozygosity events. Most mCAs have moderate to high fitness effects but occur at a low rate, being more than tenfold less common than equivalently fit single-nucleotide variants. Notable exceptions are mosaic loss of X and Y, which we estimate have roughly 1,000-fold higher mutation rates than autosomal mCAs. Although the way in which most mCAs increase in prevalence with age is consistent with constant growth rates, some mCAs exhibit different behavior, suggesting that their fitness may depend on inherited variants, extrinsic factors or distributions of fitness effects.

Mutations in hematopoietic stem and progenitor cells, which confer a 'Darwinian' fitness advantage, can clonally expand to detectable levels in the blood in a phenomenon known as clonal hematopoiesis[1–4]. Previous studies have developed population genetic frameworks to estimate the mutation rates and associated fitness effects of these mutations[5,6], and these estimates have been validated in subsequent studies leveraging serial sampling[7] and single-cell derived phylogenies[8]. These previous analyses have largely focused on the fitness effects and mutation rates of single-nucleotide variants (SNVs) in cancer-associated genes. However, recent studies have estimated that 60–80% of clonal expansions in healthy blood are driven by mutations outside of cancer-associated genes[6,8], raising the prospect of large numbers of mutations, beyond SNVs, that confer high fitness effects and could have implications for cancer risk.

mCAs are common in hematological malignancies[9,10], and a number of studies have found mCAs in the blood of healthy individuals[11–15]. As with clonal hematopoiesis driven by SNVs, the prevalence of mCAs in blood increases with age[13–17], and certain mCAs are associated with an increased risk of developing hematological malignancies[12,14,18]. However, the rate at which mCAs occur and their fitness consequences remain unknown. Furthermore, it is not clear whether fitness effects

and mutation rates exhibit any age-specific or sex-specific effects and how acquiring a highly fit variant impacts future blood cancer risk.

Here, we apply a population genetic framework to mCA calls from ~500,000 individuals in UK Biobank[14] to estimate the fitness effects and mutation rates of gains, losses and copy-neutral loss of heterozygosity (CN-LOH) events at the chromosomal arm level. Unlike SNVs, for which mutation rates are well understood, robust estimates for mCA mutation rates have been harder to measure. Our estimates reveal that mCAs conferring high fitness effects (growth rates of ≥10% per year) occur at a rate of ~1 per 10 million cells per year, approximately tenfold lower than SNVs that confer equivalent fitness. Although occurring at a relatively low rate, the fitness consequences of these mutations can be dramatic, resulting in an expansion in the size of the mutated clone at rates of up to 15–20% per year. Furthermore, there is a clear association between fitness effect and cancer risk, implying that the acquisition of some highly fit mCAs makes it more likely for clones to achieve malignant potential. The sheer scale of the UK Biobank data coupled with a rational expectation of how the distribution of mCA cell fractions should evolve with age enables us to detect specific mCAs with unexpected age and sex dependence, suggesting that the risk of acquisition and/or expansion of certain

Early Cancer Institute, University of Cambridge, Cambridge Biomedical Campus, Cambridge, Cambridgeshire, UK. ✉e-mail: cw672@cam.ac.uk; jrb75@cam.ac.uk

mCAs may be non-uniform throughout life and may be influenced by extrinsic factors.

## Results

### Mutation rates and fitness effects of mCAs

To estimate the fitness effects and mutation rates of mCAs, we analyzed cell fraction estimates of mCAs from Loh et al.'s study of single-nucleotide polymorphism array data from ~500,000 UK Biobank participants[14] (Supplementary Figs. 1–4). As this study incorporated long-range phase information, the authors were able to detect mCAs at cell fractions as low as 0.7%. Autosomal mCAs were detected in 3.5% of individuals, with 2,389 gain ('+'), 3,718 loss ('−') and 8,185 CN-LOH ('=') events. mCAs spanned a broad range of cell fractions and, as is the case with SNVs[5], the density of mCAs increases rapidly with decreasing cell fraction (65% of mCAs at cell fractions of 0.7–5%). Some mCAs are observed far more often than others, with some being detected hundreds of times (for example, gain of chromosome 12 ('12+'), loss of the q-arm of chromosome 20 ('20q−'), CN-LOH of the q-arm of chromosome 14 ('14q=')) and others were not detected at all (for example, loss of chromosomes 2, 5 and 8 ('2−', '5−', '8−')) (Fig. 1a and Supplementary Fig. 3).

To disentangle how much of this variation is caused by differences in mutation rates versus differences in fitness effects, we adapted our evolutionary framework[5] to quantify the mutation rates and fitness effects of specific mCAs. Cell fraction estimates for a given mCA were log transformed and their density was plotted as a function of this log-transformed cell fraction (Fig. 1b). Plotted this way, the density of a specific mCA is expected to be uniform at low cell fractions, with an amplitude set by the product of the mutation rate ($\mu$) and the stem cell population size multiplied by the symmetric cell division time in years ($N\tau$). The density of the mCA is then expected to decline above a cell fraction that is determined by a combination of the fitness effect ($s$) of the mCA and the age distribution of individuals in the cohort. Therefore, fitting the distribution of cell fractions predicted by our evolutionary framework to the observed density for a specific mCA yields estimates for $N\tau\mu$ and $s$ (ref. 5). As there are robust estimates for $N\tau$ (refs. 5,8,19), we are able to infer the $\mu$ and $s$ per year of an mCA (Fig. 1b,c, Supplementary Note 1 and Supplementary Figs. 5–7).

The mCA densities predicted by our evolutionary framework (Fig. 1c,d, solid lines) closely match the densities observed for specific mCAs. Some mCAs (for example, 3q=) have a high mutation rate, resulting in a large number of observed events, but because they confer only a modest fitness effect, the vast majority are confined to low cell fraction (Fig. 1c,d, yellow). Others (for example, 9q−) have a very low mutation rate, resulting in a modest number of observed events, but because they confer a substantial fitness effect, a considerable fraction is detected at high cell fraction (Fig. 1c,d, blue).

Applying this framework to all mCAs that were observed in at least eight individuals reveals a broad range of fitness effects and mutation rates (Fig. 2a, Extended Data Figs. 1–7 and Supplementary Table 1). The fittest mCAs (for example, 3p−, 17p−) confer fitness effects of >20% per year, enabling a stem cell that acquires one of these mCAs to clonally expand and dominate the entire stem cell pool over a 40-year timescale. The least fit detectable mCAs confer fitness effects of ~6–10% per year. Such variants are unlikely to ever reach high cell fractions (>10%), even in elderly individuals. Examining the mutation rate distribution of fitness effects (DFE) for each class of mCA (loss, gain, CN-LOH) reveals systematic differences between the three broad mCA classes (Fig. 2b). Of the three mCA classes, CN-LOH events occur at the highest rate (combined rate of ~9 × 10$^{-8}$ per cell per year). However, CN-LOH events typically confer modest fitness effects, with most being in a narrow range between ~11 and 13% per year. By contrast, the fitness effects of losses are systematically higher, with most fitness effects being ~12–17% per year. However, as a class, losses occur at a combined rate of ~4 × 10$^{-8}$ per cell per year, ~2.5-fold lower than CN-LOH. Gains appear to have a broad range of fitness effects but occur at the lowest combined mutation rate of ~2 × 10$^{-8}$ per cell per year.

### Sex differences in fitness effects and mutation rates

Previous studies have reported sex biases in the prevalence of certain mCAs; for example, 15q+ is more common in men and 10q− is more common in women[14]. By applying our framework, we can reveal whether sex biases are driven by differences in fitness effect, differences in mutation rate or a combination of these factors. To examine this question, we calculated the sex-specific fitness effect and mutation rate for mCAs that were observed at least ten times in men and in women (Supplementary Figs. 8–14 and Supplementary Table 2). Approximately half of the mCAs (25 out of 60) showed no significant sex-specific differences in either fitness effects or mutation rate. Of the 35 mCAs that showed significant sex differences (Fig. 2c), most had modest differences in fitness effect, with fold differences between 1.04 and 1.42. By contrast, differences in mutation rate were sometimes substantial, with fold differences between 1.3 and 12. For example, we infer that the observed higher prevalence of 10q− in women is due to a ~fourfold higher mutation rate in women, with limited evidence for any sex bias in fitness effect. The observed higher prevalence of 15q+ in men is probably due to a ~12-fold higher mutation rate in men.

### Age dependence of mCAs

Our framework, which assumes that the fitness effects and mutation rates of mCAs remain constant throughout life, predicts how the prevalence of mCAs should increase with age (Fig. 3). Above a certain age, determined by the sequencing sensitivity, the prevalence of a specific mCA is expected to increase linearly at a rate of $N\tau\mu s$. We reasoned that our framework could serve as a null model to identify mCAs whose age prevalence deviates from the prevalence expected, which might highlight interesting biology (Supplementary Note 2). Overall, the observed prevalence of gain and loss events in both men and women is in close agreement with the predicted prevalence (Fig. 3a–c). CN-LOH events, by contrast, show weaker age dependence than expected, particularly in women, possibly pointing to a violation of the underlying assumptions. By quantifying the deviation between the observed and expected prevalence across the three different age groups in UK Biobank, we are able to examine the agreement between the observed and expected age prevalence for specific mCAs (Fig. 3d, Supplementary Figs. 15–22 and Supplementary Table 3). Many mCAs exhibit age dependence broadly in line with predictions (for example, 14q+, 20q−, 22q=). For mCAs exhibiting the expected age dependence, we further challenged our model by testing the age dependence of the clone size distributions (Supplementary Fig. 23). While these show qualitative agreement with the theory, there are consistently more variants in the younger age group than predicted by our framework. Some mCAs show considerable deviation from the expected prevalence in at least one of the two sexes: some show greater age dependence than expected (for example, 4q= in men); others show no age dependence (for example, 2q= in both men and women); and some even show declining age prevalence (for example, 10q− in women) (Fig. 3d).

### mCA fitness effects and cancer risk

We reasoned that the fitness effect of mCAs may be correlated with their subsequent risk of hematological malignancy. Despite 98 mCAs having fitness effects of >10%, only 20 of these were significantly associated with a subsequent hematological malignancy diagnosis during 4–9 years of UK Biobank follow-up[14]. Although this length of follow-up means that we are probably underpowered to detect all mCAs associated with cancer risk, it could be that some mCAs drive benign clonal expansions[20]. Nonetheless, conditioning on mCAs that have a non-zero odds ratio of hematological malignancy, we do indeed find a significant correlation between the fitness effect and odds ratio of

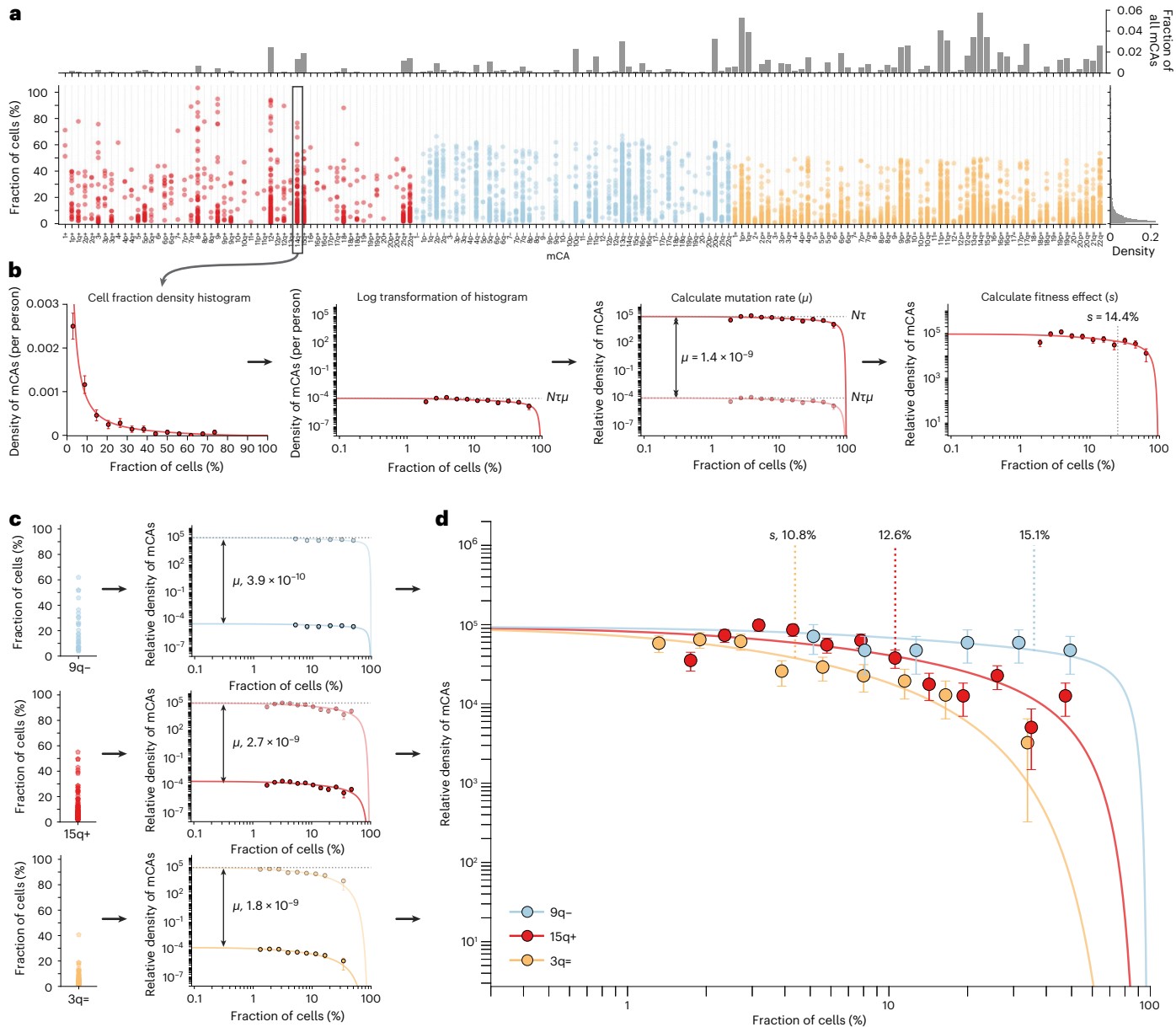

**Fig. 1 | Estimating mCA mutation rates and fitness effects. a**, Distribution of cell fractions for each autosomal mCA that was detected in ≥1 individual in UK Biobank. Red, gains; blue, losses; yellow, CN-LOH events. **b**, Plotting all cell fraction measurements for a particular mCA (for example, 14q+, $n = 147$ observations) as log-binned histograms yields estimates for $s$ (mCA fitness effect) and $N\tau\mu$ (product of the mutation rate, $\mu$, and the stem cell population size multiplied by the symmetric cell division time in years, $N\tau$). Using an estimate for $N\tau$ of ~100,000 allows the mCA-specific mutation rate to be calculated. Using the known distribution of ages in UK Biobank enables $s$ to be calculated.

Data points are presented as mean values ± s.e.m. **c**, Three example mCAs with different fitness effects and mutation rates: 9q− ($n = 28$ observations), 15q+ ($n = 206$ observations) and 3q= ($n = 92$ observations). Data points are presented as mean values ± s.e.m. **d**, The mCA densities predicted by our evolutionary framework (solid lines) closely match the densities (data points) observed for the specific mCAs (9q−, $n = 28$ observations; 15q+, $n = 206$ observations; 3q=, $n = 92$ observations). The greater the fitness effect of the mCA, the faster the clone grows and so the more likely it is to be seen at higher cell fractions. Data points are presented as mean values ± s.e.m.

subsequent blood cancer (Pearson's $R$, 0.63; $P = 9.2 \times 10^{-6}$) (Fig. 4 and Supplementary Table 4).

## Mosaic loss of chromosomes Y and X

Mosaic loss of the Y chromosome (mLOY) and mosaic loss of the X chromosome (mLOX) are commonly observed in blood and are increasingly prevalent with age[13,21]. We reasoned that our framework could determine whether this high prevalence was caused by increased fitness effects or anomalously high mutation rates. To check this reasoning, we considered the cell fraction distribution of all 22,367 mLOY events across the 220,899 men in UK Biobank (Fig. 5a). The distribution

of mLOY cell fractions is in remarkably close agreement with the predictions of our framework, conferring a modest fitness effect ($s$) of ~13% per year, and a mutation rate ($\mu$) of ~9 × 10⁻⁷ per year, which is ~1,000-fold higher than the typical autosomal mCA mutation rate (Fig. 5c and Extended Data Fig. 8a). The close agreement between the predicted and observed age dependence for mLOY provides further support for our fitness effect and mutation rate inferences (Extended Data Fig. 8b).

We next considered the 8,577 mLOX events across 261,890 women. In contrast to mLOY, the observed distribution of mLOX cell fractions is clearly inconsistent with the predictions of our framework based on

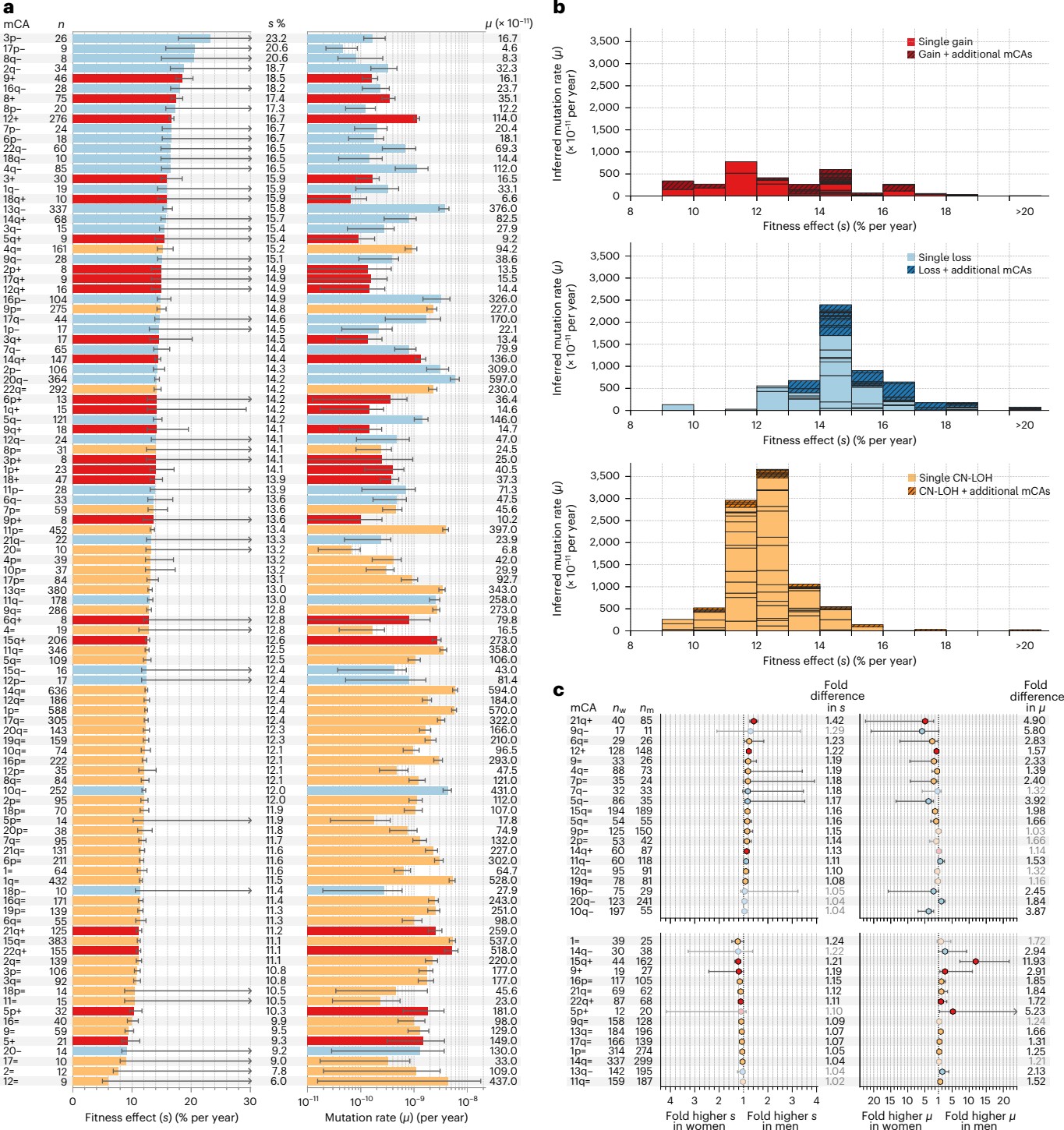

**Fig. 2 | The fitness and mutational landscape of autosomal mCAs. a**, Inferred fitness effects ($s$) and mutation rates ($\mu$) for all autosomal mCAs observed in ≥8 individuals. Red, gains; blue, losses; yellow, CN-LOH events. $n$ indicates the number of individuals (with a single mCA) in whom the mCA was observed. Error bars represent 95% confidence intervals. Some upper error bars are undetermined (arrows) owing to technical limits of detecting high cell fraction events (see Supplementary Note 1). **b**, Mutation rate DFE for gains (red, top plot), losses (blue, middle plot) and CN-LOH events (yellow, bottom plot). Each box within a fitness interval column represents a specific mCA. Darker hatched boxes represent the fitness effects of a specific mCA that was seen in individuals that also harbored ≥1 other mCA. **c**, Fold differences in fitness effects ($s$) and mutation rates ($\mu$) between men and women for gains (red), losses (blue) and CN-LOH (yellow) events that were observed as a single mCA ≥10 times in men and in women and which showed a significant difference in either fitness effect or mutation rate. $n_w$ and $n_m$, number of times each mCA was observed in women and men, respectively. Error bars represent the 95% confidence intervals from the distribution of fold difference between men and women. Upper 95% confidence intervals >150 (see Supplementary Table 2) are indicated by arrows. Data points with no black border and whose fold difference values are in grey indicate mCAs whose fitness effect or mutation rate was not significantly different between men and women.

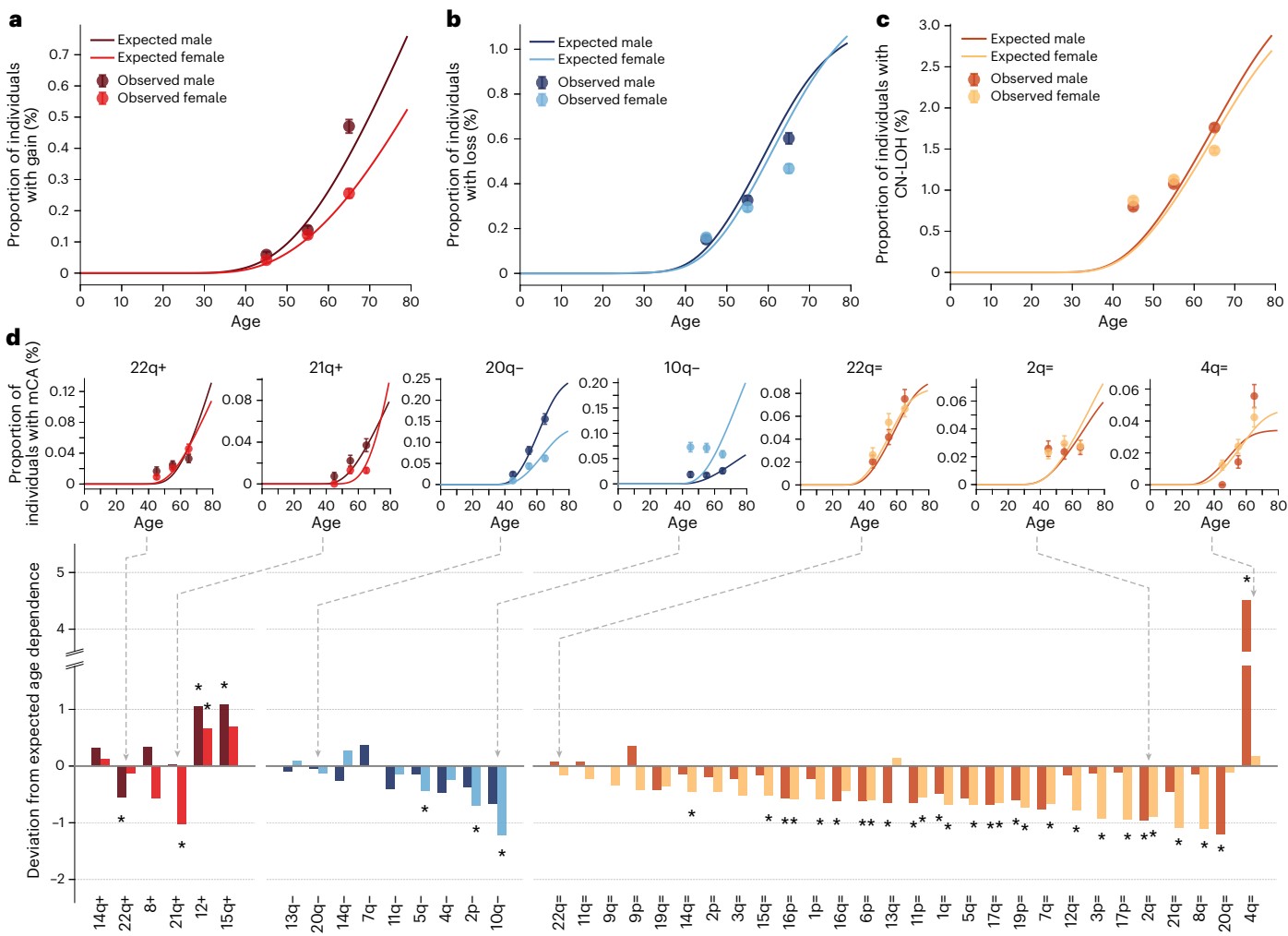

**Fig. 3 | Age dependence of mCAs. a–c,** Observed and expected prevalence of gains (**a**), losses (**b**) and CN-LOH (**c**) events for men and women. Expected prevalence (solid lines) was calculated by summing the expected prevalence of each mCA in the mCA class. Observed number of gain events, 435 (women), 595 (men); observed number of loss events, 915 (women), 917 (men); observed number of CN-LOH events, 3315 (women), 2967 (men). Error bars represent sampling error ±1 s.d. **d,** Deviation from expected age dependence for each mCA observed ≥30 times in men and ≥30 times in women, with examples from each mCA class. Red, gains; blue, losses; yellow, CN-LOH events. Darker shade,

men; lighter shade, women (see Supplementary Figs. 15–17 for age dependence plots for all mCAs). Deviation from expected age dependence was calculated as the relative difference between the observed and expected age dependence gradients. Positive deviation reflects mCAs that show a greater increase in prevalence with age than expected. Negative deviation reflects mCAs that show less increase in prevalence with age than expected. mCAs whose deviation was statistically significant ($P < 0.05$) are highlighted with an asterisk (see Supplementary Note 2 and Supplementary Table 3).

a single fitness effect and single mutation rate (Extended Data Fig. 8c). This inconsistency is further evidenced by the marked discrepancy between the observed and expected age dependence (Extended Data Fig. 8d). The way in which the observed distribution of mLOX cell fractions deviates from our null model, with two distinct plateaus, is highly suggestive of two distinct mLOX events occurring at different rates and conferring different fitness effects. To test this theory, we simulated mLOX events with two possible combinations of mutation rate and fitness effect and observed cell fraction distributions that were very similar to those observed in the UK Biobank data (Extended Data Fig. 9). Therefore, we modified our framework to predict the distribution of cell fractions that would be observed for two distinct fitness effects occurring at two distinct mutation rates for mLOX. This modified framework provided a much closer agreement to the observed cell fraction estimates in UK Biobank (Fig. 5b and Extended Data Fig. 10a). It predicts that there are two possible mLOX events: one that confers a large fitness effect ($s \approx 16\%$ per year) but occurs at a low rate ($\mu \approx 1 \times 10^{-8}$ per year), and one that confers a low fitness effect ($s \approx 7\%$ per year) but

occurs at a rate similar to mLOY ($\mu \approx 1.6 \times 10^{-6}$ per year) (Fig. 5c,d and Supplementary Table 5). Furthermore, this modified framework provides a much closer agreement between the predicted and observed age dependence for mLOX (Extended Data Fig. 10b).

## Discussion

Here, we have adapted our evolutionary framework[5], which uses clone size estimates from large-scale genomic data sets, to estimate mutation rates and fitness effects of mCAs. Our previous inferences of fitness effects[5], stem cell population size ($N\tau$) (ref. 5) and proportion of missing driver mutations[6] have subsequently been validated using longitudinal data[7,22] and single-cell phylogenies[8]. However, these recent studies were unable to provide direct estimates for the mutation rate or fitness effects of mCAs. By extending our framework to mCAs, we have been able to provide estimates for the rate of mutation and fitness consequences of these important genomic alterations.

Unlike somatic SNV mutation rates, which can be estimated from large-scale, single-cell sequencing studies[19,23], somatic mCA mutation

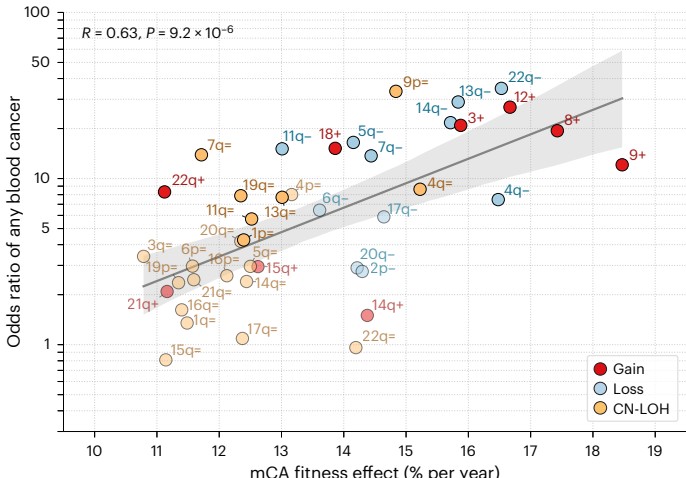

**Fig. 4 | mCA fitness effects and blood cancer risk.** The relationship between inferred fitness effect and odds ratio (OR) of any blood cancer is shown for autosomal mCAs with a non-zero OR[14] and that were observed in ≥30 individuals (n = 41 mCAs) (see Supplementary Table 4). Red, gains; blue, losses; yellow, CN-LOH events. The mCAs highlighted in bold are those that Loh et al.[14] determined to have a statistically significant increased risk of any blood cancer. Pearson's R correlation coefficient, two-tailed P value and 95% confidence intervals (gray shaded area) are shown. The blood cancers were diagnosed >1 year after DNA collection (within 4–9 years of follow-up) in individuals with no previous cancer and with no restriction on blood counts at assessment.

rates have historically been harder to calculate. Our framework allows us to calculate mutation rates for individual mCAs as well as classes of mCAs. The key insight is that the density of mCAs will be determined by the product of $N\tau$ and the mutation rate ($\mu$); therefore, by using recent estimates for $N\tau$ (refs. 5,8,19), it is possible to estimate the mCA mutation rate. Strikingly, the total mutation rate to highly fit mCAs ($s > 10\%$ per year) is over tenfold lower than the total mutation rate to highly fit SNVs (Fig. 6). Recent work has suggested that there is a large amount of positive selection in blood that is not explained by SNVs[6]. Our analysis suggests that even by accounting for the additional positive selection contributed by mCAs, a large fraction of positive selection would remain unexplained. This may point to an important role for a large number of variants driving clonal expansions that reside outside cancer-associated genes and that are individually rare but collectively common.

By considering the cell fraction spectra across individuals for each mCA, our framework enables us to quantify mCA-specific fitness effects. There are 168 different possible mCAs that could have been detected in the UK Biobank data set at the chromosome and chromosomal arm level. Using our framework, we were able to infer the fitness effects of 105 of these mCAs: 86% of possible CN-LOH, 60% of possible losses and 43% of possible gains. The fitness effects of the fittest mCAs appear to be similar to the fitness effects of the fittest SNVs[5], with both conferring selective advantages in the range of ~10–20% per year. It is important to bear in mind the possible limitations of our fitness effect inferences. Owing to technical limitations in detecting low cell fraction mCAs (<1%), it is difficult to estimate fitness effects below $s < 7\%$ per year. Similarly, due to difficulty in distinguishing mosaicism from germline, losses and CN-LOH events 'drop out' of the data at cell fractions >67% and >54% respectively; these upper cell fraction limits of detection[14] mean that our inferences for highly fit losses ($s > 24\%$) and CN-LOH ($s > 25\%$) may underestimate their true fitness (Supplementary Note 1 and Supplementary Tables 6–8).

The key assumptions of our framework are that mCAs are acquired stochastically at a constant rate throughout life and then expand with an mCA-specific intrinsic fitness effect, independent of age, sex or

genetic background. Although the data for most mCAs appears to be consistent with these assumptions, we are able to highlight a number of mCAs that deviate from this simple 'null' model, calling into question one or more of the assumptions for these specific mCAs and allowing us to identify mCAs with potentially interesting biology.

It is likely that cell-extrinsic effects may influence the growth rate and/or mutation rate for some mCAs, as they do for SNVs[24]. For example, we find significant differences in fitness effects and/or mutation rates between men and women for some mCAs (Fig. 2c), suggesting that hormonal influences and/or sex-linked genetic influences may have an effect. For example, 5q− shows a higher mutation rate in women but a higher fitness effect in men, suggesting that extrinsic effects play a role. Indeed, this may be related to the higher prevalence of myelodysplasia (MDS) associated with 5q− ('del(5q) syndrome') in women but its worse prognosis in men[25]. It is also possible that the growth rate conferred by an mCA is explicitly dependent on age, as seen for DNMT3A-mutant clones[7].

It is also possible that the growth rate and/or mutation rate of mCAs are shaped by genetic background and interaction with other somatic variants[17]. A clear example of this identified by our framework is 10q−, which shows almost no age dependence, in sharp contrast to the predictions of our framework (Fig. 3d). One possible explanation for this lack of age dependence could be the presence of a genetic variant that strongly predisposes an individual to acquire 10q− early in life. Indeed, 10q− was highlighted by Loh et al.[13] because they found clear evidence that it was strongly associated with an inherited variant located near a known genomic fragile site on the same chromosome. Further evidence for possible germline-specific effects is that the distribution of mCAs per person (Supplementary Fig. 4) is broader than expected, as has been previously observed for SNVs in blood[2,3] and bladder[26]. Another possibility is that some mCAs provide a 'second hit' to somatic point mutations in certain commonly mutated genes; for example, JAK2[V617F] mutations are found in at least 60% of individuals with 9p CN-LOH events[14]. Our framework does not account for such second-hit events and would require further development to include these effects. To determine whether second-hit events might be biasing our fitness effect estimates, we assessed whether the fitness effect of an mCA that occurs on its own in an individual is significantly different to the fitness effect of the mCA when it co-occurs with at least one other mCA in that person (Supplementary Fig. 24). We found no clear relationship for the fitness effects depending on the number of mCAs; however, it is likely that cell fraction distributions alone would be underpowered to disentangle single-hit and double-hit events.

It is also possible that the growth rate estimates for mCAs and SNVs, particularly at older ages, are influenced by clonal competition from other acquired alterations. Such clonal competition would result in weaker age dependence than is predicted by our model, or even a decline in prevalence with age. Decreasing age prevalence (for example, 20q= in men; Supplementary Fig. 17) is a particularly striking observation that could be caused by fitter variants out-competing less-fit mCAs at older ages; however, other explanations are also possible (for example, the mCA becoming disadvantageous at older age).

Another important assumption in our analysis is that mCAs of a specific type affecting any part of a chromosomal arm have the same fitness effect. In some instances, this simplification is probably reasonable; for example, gain events often affect entire chromosome arms and are therefore homogeneous in the regions affected. However, in other cases, there is likely to be variation in the fitness effect of mCAs affecting different parts of a chromosomal arm. For CN-LOH and loss events, for which there is substantial variation in which specific regions of the chromosome are affected (Supplementary Figs. 1 and 2), the assumption that all events on the same chromosome arm confer the same fitness effect may be more questionable. Where sufficient data existed, we checked the length dependence of our inferences for losses (Supplementary Note 3 and Supplementary Table 9). We found that

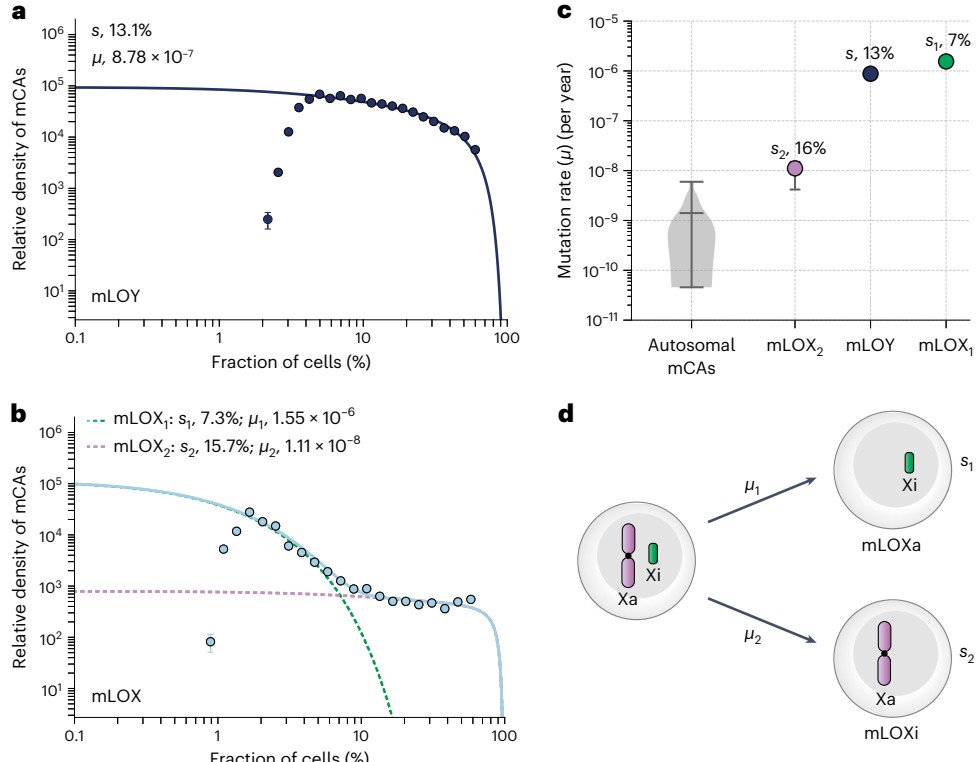

**Fig. 5 | Fitness effects and mutation rates of mLOY and mLOX. a**, The distribution of mLOY cell fractions (data points, $n = 22,367$ observations) is in close agreement with the predictions of our framework (solid lines). Data points are presented as mean values ± s.e.m. **b**, The distribution of mLOX cell fractions (data points, $n = 8577$ observations) is suggestive of a framework involving two distinct mLOX events (blue solid line): one event (mLOX$_1$) occurring at high mutation rate ($\mu_1$) with low fitness ($s_1$) (green dashed line) and another event (mLOX$_2$) occurring at low mutation rate ($\mu_2$) and high fitness ($s_2$) (purple dotted line). Data points are presented as mean values ± s.e.m. **c**, Inferred mutation

rates ($\mu$) for autosomal mCAs (violin plot, $n = 105$ mCAs), mLOX$_1$, mLOY and mLOX$_2$. The middle horizontal line in the violin plot shows the mean autosomal mCA mutation rate ($1.41 \times 10^{-9}$ per year) and the top and bottom horizontal lines indicate the minimum ($4.56 \times 10^{-11}$ per year) and maximum ($5.97 \times 10^{-9}$ per year) autosomal mCA mutation rates. Error bars on the mLOX and mLOY data points represent 95% confidence intervals. Fitness effects ($s$) for mLOX$_1$, mLOY and mLOX$_2$ are also shown. **d**, Possible explanation for two mLOX events: one involving loss of the active X (Xa) and one involving loss of the inactive X (Xi) leading to the generation of mLOXa and mLOXi cells, respectively.

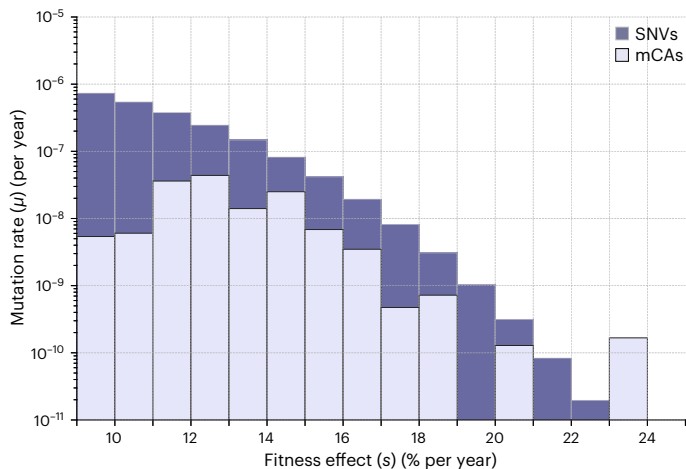

**Fig. 6 | DFE for mCAs versus SNVs.** The mutation rate DFE for all classes of autosomal mCAs (Fig. 2b combined) is shown in light purple, the mutation rate DFE for SNVs across a targeted 'cancer panel' of ~1.1 MB (inferred in Poon et al.[6]) is shown in dark purple. The mutation rate to fitness effects of 10–25% per year is $1.5 \times 10^{-7}$ for autosomal mCAs and $1.5 \times 10^{-6}$ for SNVs.

although there appears to be some length dependence of the mutation rate, inferred fitness effects were largely insensitive to mCA length, suggesting that the underlying cause of the fitness effect of the mCA is confined to a small genomic region.

There were five mCAs not observed at all in the UK Biobank data set: monosomies of chromosomes 2, 5, 8, 16 and 19 (Supplementary Fig. 3). Of note, monosomy 5 is known to be associated with MDS and acute myeloid leukemia (AML) and is associated with poor prognosis[27,28]. Monosomy 16, although rare, is also associated with myeloid malignancies and is similarly associated with poor prognosis[29]. Although the absence of monosomy 5 and 16 in the UK Biobank cohort may simply reflect low mCA-specific mutation rates, their absence could suggest that these events only occur in individuals who then rapidly progress to MDS or AML (that is, they are 'late' events in MDS or AML development).

One of the principles underlying pre-cancerous mutation acquisition and clonal expansion is that the greater the fitness effect of a mutation, the faster the clone will expand and the more likely it is that subsequent mutations will be acquired within the same clone. We find a correlation between higher mCA fitness effects and increased risk of any hematological malignancy. This is consistent with the conclusions from SNVs, in which an increased risk of AML is associated with highly fit SNVs[5]. It is important to note, however, that some mCAs driving clonal expansions may not be associated with a higher risk of malignancy. For example, 3p−, which was observed in 26 individuals and had an inferred fitness effect of -23% per year (Fig. 2), had no evidence of an increased risk of blood cancer[14]. There are several reasons why there may be a deviation from the general association between fitness effect and risk of malignancy. First, there may be additional factors, other than the fitness effect of the initial driver mutation, that are important for subsequent progression to malignancy; for example, interaction with other driver mutations. Second, there is likely to be variability in the

time it takes to progress to malignancy, and therefore the 12 years of follow-up in the UK Biobank data may be insufficient to observe subsequent cancer development in some individuals. Third, some mCAs, although highly 'fit', may actually be protective. Although there is insufficient data to identify low-risk or protective mCAs in these data, there are examples of such mutations in other tissues; for example, *NOTCH1* mutations, which are thought to be protective in the esophagus[20].

Given that mLOY and mLOX are common events in blood, they provide good statistical power for testing the predictions of our framework. These events demonstrate why prevalence, the total number of events detected above technical sensitivity limits across all individuals, can be a misleading statistic (Supplementary Note 4). Our framework reveals that although mLOY is more prevalent than mLOX, the overall mutation rate to mLOX is higher, but it is less prevalent because it confers a modest fitness effect. The behavior of mLOX is a further demonstration of why our framework is an important null model for understanding the fitness effects and mutation rates of mosaic events. mLOX provides the strongest evidence for deviating from this null model and is more consistent with a model in which mLOX can confer two distinct fitness effects that occur at two different rates. Although we do not have direct evidence for the cause of these two distinct events, we speculate that it could be related to the fact that the X chromosome that is lost will be either 'active' or 'inactive'; however other explanations for two distinct fitness effects are possible. Loss of either the 'active' or the 'inactive' X could plausibly result in two distinct effects occurring at two different rates (Fig. 5d). Previous studies have shown that high cell fraction mLOX typically results from the loss of the 'inactive' X chromosome[30], suggesting that this could be the event that drives the higher fitness effect. More generally, these examples demonstrate how quantitative features in the distribution of cell fractions, and their evolution with age, can reveal insights beyond the underlying biology.

## Online content

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

## Methods

### Data used in the analysis

Cell fraction estimates of mCAs generated by Loh et al.[14] from UK Biobank participants aged ~40–70 years were used in our analysis. The mCA calls were generated from 482,789 UK Biobank participants[14], of whom 261,890 were men and 220,899 were women. The North West Multi-centre Research Ethics Committee (MREC) reviewed and approved the UK Biobank scientific protocol and operational procedures (REC reference number 21/NW/0157), and all participants provided signed informed consent at enrollment.

Loh et al.[14] transformed genotyping intensities from the UK Biobank single-nucleotide polymorphism array data into $\log_2 R$ ratios and B-allele frequencies (BAFs) to obtain measures of total and relative allelic intensities, respectively, and incorporated long-range phase information to call mCAs at cell fractions as low as 0.7%. There was a sharp cut-off at cell fractions of ≥67% for losses and ≥54% for CN-LOH events, corresponding to BAF deviations of >0.25 (Fig. 1a). This occurred because the analytical approach used by Loh et al.[14] resulted in heterozygous single-nucleotide polymorphisms 'dropping out' of the data if BAF deviations were >0.25. Autosomal mCAs were called on all chromosomal arms except 13p, 14p, 15p, 21p and 22p (Supplementary Figs. 1 and 2). The majority of autosomal mCAs were most commonly seen in individuals as single events, although some were more commonly found in the context of additional autosomal mCAs (for example, 17p-, 18+) (Supplementary Fig. 3). In individuals for whom an autosomal mCA was detected, the average number was one (Supplementary Fig. 4).

### Maximum likelihood parameter estimation

Our evolutionary framework, which allows estimation of mCA-specific fitness effects (s) and mCA-specific mutation rates (μ), is based on a continuous time branching process for hematopoietic stem cells, as previously described for SNVs[5]. In brief, the framework is based on a simple stochastic branching model of hematopoietic stem cell dynamics, in which mCAs (with an mCA-specific fitness effect, s) are acquired stochastically at a constant rate (μ per year). The framework is adapted from our previous work[5], taking into account cell fraction (rather than variant allele frequency) measurements, the UK Biobank age distribution and the upper cell fraction cut-off for loss and CN-LOH events. How the distribution of cell fractions predicted by our evolutionary framework changes with age (t), the mCA-specific fitness effect (s), the mCA-specific mutation rate (μ), the population size of hematopoietic stem cells (N) and the time in years between successive symmetric cell differentiation divisions (τ) is given by the following expression for the probability density as a function of l = log(cell fraction):

$$\rho(l) = \frac{N\tau\mu}{(1-e^l)} e^{-\frac{e^l}{\phi(1-e^l)}}, \text{ where } \phi = \frac{e^{st}-1}{N\tau s} \tag{1}$$

Fitting the cell fraction distributions predicted by our evolutionary framework to the observed densities for a specific mCA enables us to infer estimates for Nτμ and s. To take into account the varying ages in UK Biobank, predicted densities were calculated by integrating the theoretical density for a given age (equation 1) across the age distribution in UK Biobank (23.8% aged 40–49 years, 33.6% aged 50–59 years, 42.6% aged 60–69 years). For each mCA, a maximum likelihood approach was used for parameter estimation, minimizing the L2 norm between the cumulative log-rescaled densities and the cumulative log-predicted densities, for all logit(cell fractions), to optimize Nτμ and s. Compared to using log cell fractions, logit cell fractions provide an improvement in correctly inferring the fitness effects of highly fit variants, which we extensively checked with simulated data (Supplementary Note 1). This is because very high fitness variants 'bunch up' close to 100% cell fraction, and this feature is more clearly delineated on a logit scale rather than on a log scale. The mCA-specific mutation rate (μ) was estimated

by dividing the inferred Nτμ by an Nτ of ~100,000 (refs. 5,8,19) (Fig. 2a, Extended Data Figs. 1–7 and Supplementary Table 1).

### Sex differences in mCA fitness effects and mutation rates

To calculate sex-specific mutation rates (μ) and fitness effects (s), a maximum likelihood approach was used as described above, for mCAs that were observed as single mCAs in at least ten men and/or at least ten women (Supplementary Figs. 8–14 and Supplementary Table 2). To determine whether any difference between sex-specific mutation rates or fitness effects was statistically significant, a distribution of difference probability curve was calculated (for men versus women) and the P value of the fold difference was taken as the area under the fold-difference probability curve where the difference was ≤1. If P < 0.05, then the sex difference was deemed to be statistically significant.

### Age dependence of mCAs

The prevalence of an mCA within a particular range of cell fractions can be calculated by integrating the mCA's probability density, given in equation 1, as a function of f = cell fraction, over the range of cell fractions ($f_0$ to $f_1$):

$$\int_{f_0}^{f_1} \frac{N\tau\mu}{f(1-f)} e^{-\frac{f}{\phi(1-f)}} df, \text{ where } \phi = \frac{e^{st}-1}{N\tau s} \tag{2}$$

Our framework, which assumes that the fitness effects and mutation rates of mCAs remain constant throughout life, predicts how the prevalence of mCAs should increase with age. The prevalence of a specific mCA is expected to increase approximately linearly at a rate given by Nτμs once the individual is above a certain age, determined by the cell fraction limit of detection ($f_{lim}$) and the mCA-specific fitness effect (s). This is because, provided the limit of detection is less than the cell fraction at which the exponential decline in cell fraction densities occurs (that is, $f_{lim} \ll \phi$), the mCA prevalence can be approximated as:

$$\int_{f_{lim}}^{f_1} \frac{N\tau\mu}{f(1-f)} e^{-\frac{f}{\phi(1-f)}} df \approx N\tau\mu\log\left(\frac{\phi}{f_{lim}}\right) \approx N\tau\mu st + C \tag{3}$$

where $\phi = \frac{e^{st}-1}{N\tau s}$ and $C = -N\tau\mu\log(N\tau s f_{lim})$

**Age dependence of gains, losses and CN-LOH events.** To calculate the expected prevalence of each class of mCA (gains, losses, CN-LOH) as a function of age (Fig. 3a–c), the expected prevalence of each individual mCA within the class (for example, 1=, 1p=, and so on for the CN-LOH class) was calculated by integrating equation 2 between $f_0$ = mCA-class-specific lower limit of detection and $f_1$ = mCA-class-specific upper limit of detection (Supplementary Note 1 and Supplementary Table 6), using the sex-specific μ and s values of each mCA (Supplementary Table 2). The overall expected prevalence for the mCA class was then calculated by summing the expected prevalence of each mCA in the mCA class and then compared to the summed observed prevalence of each mCA in the class (above the mCA class-specific lower limit of detection) across three different age groups (ages 40–50, 50–60 and 60–70 years).

**Age dependence of individual mCAs.** To calculate the expected prevalence of individual mCAs as a function of age (Supplementary Figs. 15–17), the expected prevalence of each mCA (observed ≥30 times in men and ≥30 times in women) was calculated by integrating equation 2 between the $f_0$ = mCA-specific lower limit of detection and $f_1$ = mCA-specific upper limit of detection, using the sex-specific μ and s values of each mCA (Supplementary Table 2). The class-specific upper limit of detection (Supplementary Note 1 and Supplementary Table 6) was used as the upper cell fraction limit of detection. The lowest cell fraction detected for the mCA, multiplied by 1.5 (to reduce the

false-negative rate), was used as the lower limit of detection of the mCA. The expected prevalence was compared to the observed prevalence of the mCA (above the lower limit of detection of the mCA), across three different age groups (ages 40–50, 50–60 and 60–70 years).

**Quantifying deviation from expected age dependence.** To quantify any deviation from the expected age dependence (Fig. 3d), we calculated the relative difference between the gradients of the observed and expected age dependence (Supplementary Fig. 18). A maximum likelihood approach was used to calculate the gradients ($m_{obs}$ or $m_{exp}$), minimizing the L2 norm between the observed or expected prevalence ($p$) and $y = mt + C$ across three age ($t$) groups (ages 40–50, 50–60 and 60–70 years), optimizing $m$ and $C$. To account for error in prevalence, the square distance was multiplied by $T/2p$, where $T$ is the total number of people in the age group. An age group was excluded from both observed and expected gradient estimation if the observed prevalence in that age group was zero (as the prevalence would not be expected to be increasing linearly at that age). To determine whether any deviation from expected age dependence was statistically significant, a distribution of difference probability curve was calculated (for observed versus expected age dependence gradients) and the $P$ value of the difference was calculated as the area under the curve where the difference was ≤0. If $P < 0.05$, then the deviation from expected age dependence was deemed statistically significant. The relative difference, $(m_{obs} - m_{exp}) / m_{exp}$, between the observed and expected age dependence gradients for individual mCAs is shown in Fig. 3d, and the $P$ values are given in Supplementary Table 3.

### mLOY and mLOX

Unlike mLOY, the shape of the mLOX cell fraction distribution was clearly inconsistent with that predicted by our framework. We considered whether the shape of the mLOX cell fraction distribution (Extended Data Fig. 8c) could be explained by two distinct mutational events underlying mLOX: one that occurs at a high mutation rate ($\mu_1$) but confers a small fitness effect ($s_1$) and another that occurs at a much lower rate ($\mu_2$) but confers a substantially larger fitness effect ($s_2$). To test this theory, we generated simulated mLOX calls across ~500,000 people (as in Supplementary Note 1), but with two possible combinations of mutation rate and fitness effect ($\mu_1, s_1$) and ($\mu_2, s_2$) for the mLOX event. The distribution of the simulated mLOX cell fractions (Extended Data Fig. 9a) was indeed similar to the distribution observed for mLOX in UK Biobank (Extended Data Fig. 8c) and attempting to fit a single theory distribution to the simulated mLOX distribution revealed a poor fit to the data and inaccurate inferences for $s$ and $\mu$.

### Inferring mutation rates and fitness effects for two distinct mLOX events.

When the product of the difference between the fitness effects of two mLOX events and the age, $T$, is large enough ($(s_2 - s_1)T \gg 1$), and when the two mutation rates are substantially different ($\mu_1 \gg \mu_2$), a qualitative feature emerges in the distribution characterized by a large peak at low cell fractions (set by the high mutation rate, low fitness event (which falls off at $\phi_1 \approx \exp(s_1T) / N\tau\mu_1 s_1$)), followed by a plateau out to higher cell fractions (set by $\phi_2 \approx \exp(s_2T) / N\tau\mu_2 s_2$). This separation of regimes means that the low cell fraction events are dominated by one event and the high cell fraction events are dominated by the other. Therefore, it is possible to estimate the ($\mu_1, s_1$) and ($\mu_2, s_2$) parameters by fitting to the low cell fraction versus the high cell fraction distributions independently. We divided the histogram of simulated mLOX cell fractions into two sets of data points: the lower cell fraction data points included simulated mLOX events whose cell fractions were below the point at which the fall-off in cell fraction densities starts to plateau (lower split point, vertical dashed line at ~9% cell fraction in Extended Data Fig. 9b). The upper cell fraction data points included mLOX events whose cell fractions were above the cell fraction that was three histogram 'bins' above the lower split point (upper split point,

vertical dashed line at ~14% cell fraction in Extended Data Fig. 9b) to avoid fitting independent distributions to data points for which there is expected to be the most contribution from both distributions together. A theory distribution was then fit to each of the two data point sets, using a maximum likelihood approach to infer the parameters $\mu_1, s_1$ (for the distribution at lower cell fraction) and $\mu_2, s_2$ (for the distribution at higher cell fraction). Using this approach, we were able to infer very close to the ground-truth values for these parameters (Extended Data Fig. 9b).

We then applied this approach to the UK Biobank mLOX data by dividing the mLOX cell fraction histogram into two sets of data points and fitting a theory distribution to each of the two data point sets (Fig. 5b and Extended Data Fig. 10a). This enabled us to infer the fitness effects and mutation rates of two distinct mLOX events: one that occurs with a high mutation rate ($s_1$) but low fitness effect ($s_1$), and another that occurs with a low mutation rate ($\mu_2$) but high fitness effect ($s_2$). These inferred parameters for $s_1, \mu_1, s_2$ and $\mu_2$ were then used to calculate the expected combined prevalence of the two distinct mLOX events as a function of age (Extended Data Fig. 10b).

**Considering multiple independent allosomal mCA events.** A key difference between allosomal and autosomal mCAs is that we infer both mLOY and mLOX to occur at a high mutation rate, such that multiple independent allosomal mCA events may expand at the same time within the same individual. We generated simulated data to test whether the features in the mLOY and mLOX distributions (Fig. 5) may result from the inability to distinguish independent events from the sum (mLOY) or the difference (mLOX) of mCA events within an individual (Supplementary Note 5). However, these effects produced features that were inconsistent with the features that we observed in the real data, suggesting that these effects are not contributing (Supplementary Fig. 26).

### DFE for mCAs versus SNVs

To compare the mutation rate DFE for mCAs to SNVs, we used the DFE for SNVs inferred in a previous study[6]. That study used SNV data from ref. 24—in which 1,236 nonsynonymous variants were detected across 4,160 individuals, using a targeted 'cancer panel' of ~1.1 MB— and parameterized the DFE using an exponential power distribution[6]:

$$\mu s \propto \exp\left[-\left(\frac{s}{\sigma}\right)^p\right] \tag{4}$$

They inferred the free parameters $p$ (shape of distribution) = 3, $\sigma$ (scale of distribution) = 0.1 and total nonsynonymous SNV haploid mutation rate across panel ($\int_{s=0}^{\infty} \mu s ds$) = 7.7 × 10⁻⁶. This DFE is plotted as a histogram and compared to the DFE for mCAs in Fig. 6.

### Statistics and reproducibility

No statistical methods were used to predetermine sample size. No autosomal mCA calls from the supplementary data of Loh et al.[14] were excluded in our analysis of autosomal mCAs. mLOY calls (from UK Biobank Return 3094), which were based on BAF and $\log_2R$ ratio measurements in *PAR1*, were filtered to include only those that were >2 MB in size, to reduce the risk of including focal loss events on the Y chromosome. mLOX calls (from UK Biobank Return 3094) were filtered to include only those that were >125 MB in size, to reduce the risk of including focal loss events on the X chromosome. For fitness effect and mutation rate parameter estimation, mLOX, mLOY and simulated mCA calls were downsampled to 50 calls (using NumPy 1.20.1 numpy. random.sample with numpy.random.seed(seed = 3, version = 2)) to reduce compute time.

### Reporting summary

Further information on research design is available in the Nature Portfolio Reporting Summary linked to this article.

## Data availability

The mCA calls used in our analysis are available from UK Biobank (Return 3094), through an application process described at http://www.ukbiobank.ac.uk/using-the-resource. Autosomal mCA calls are also available in the supplementary data of ref. 14 (https://doi.org/10.1038/s41586-020-2430-6).

## Code availability

All code used in this study is available on the Blundell Lab GitHub page (https://github.com/the-blundell-lab/mCA-mutation-rates-fitness-consequences) and at https://doi.org/10.5281/zenodo.19662539.

## Acknowledgements

We thank I. Matincorena and D. Steensma for their helpful comments on this work. This research was conducted using the UK Biobank Resource under application number 58387. C.J.W. and J.R.B. were supported by the Early Cancer Institute, the Cancer Research UK (CRUK) Cambridge Centre and the NIHR Biomedical Research Centre. J.R.B. is supported by a UK Research and Innovation (UKRI) Future Leaders Fellowship (MR/S031782/1). The funders had no role in study design, data collection and analysis, decision to publish or preparation of the manuscript.

## Author contributions

J.R.B. and C.J.W. conceived the study, developed the theory, analyzed the data and wrote the manuscript.

## Competing interests

The authors declare no competing interests.

## Additional information

**Extended data** is available for this paper at https://doi.org/10.1038/s41588-023-01490-z.

**Correspondence and requests for materials** should be addressed to Caroline J. Watson or Jamie R. Blundell.

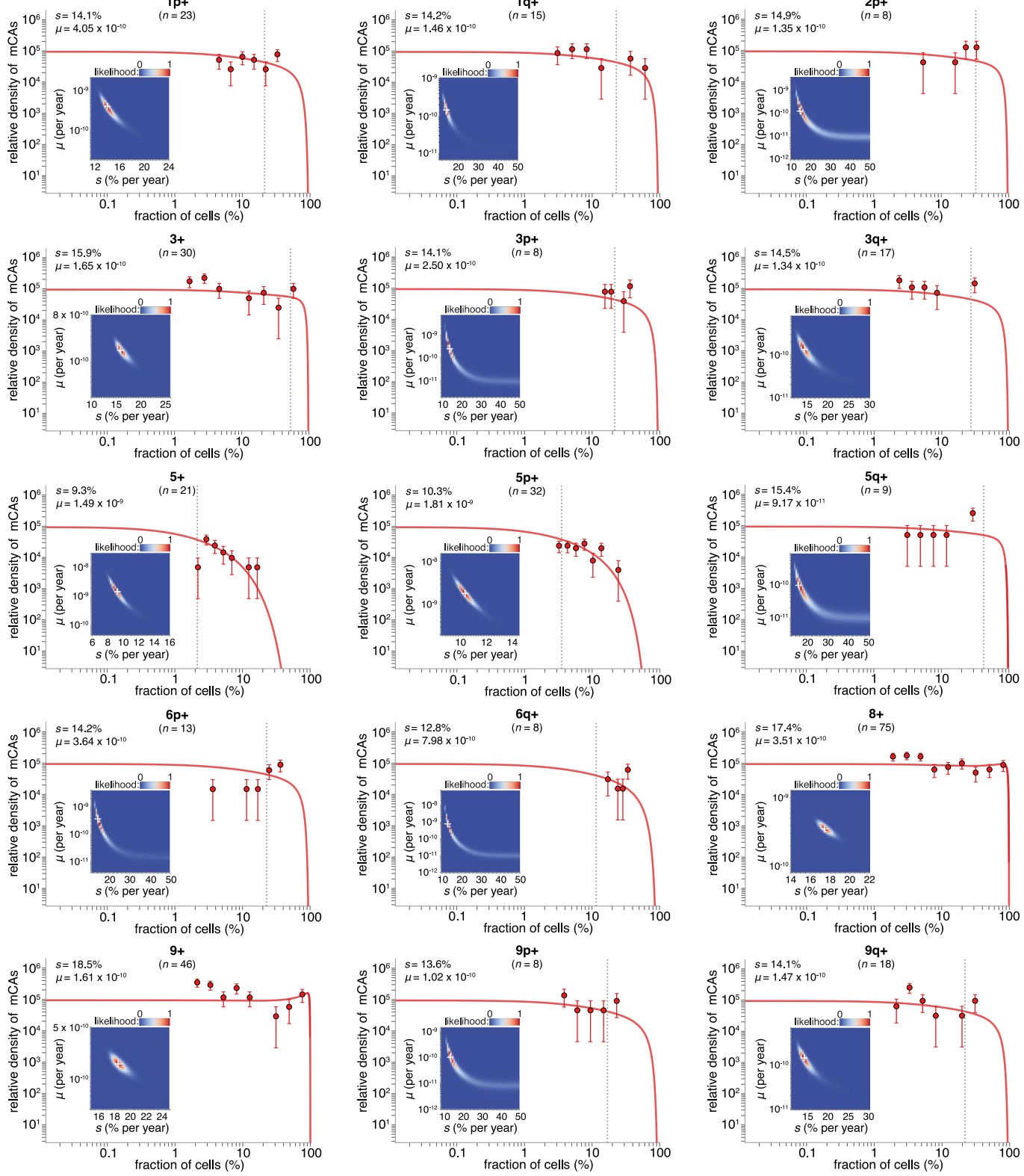

**Extended Data Fig. 1 | Parameter estimation for individual mCAs: part 1.**
The cell fraction probability density histogram is shown for each mCA (datapoints) with the theory distribution (solid line) fitted using maximum likelihood approaches (red = gains, blue = losses, yellow = CN-LOH). Datapoints are presented as mean values +/− SEM. Grey vertical dashed line shows the fitted $\phi$ parameter $\left(\frac{e^{st}-1}{Ns}\right)$, where the exponential fall-off in densities occurs. The white cross on the maximum likelihood heatmap marks the most likely mutation rate ($\mu$) and fitness effect ($s$). $n$ = number of individuals (with a single mCA) in whom the mCA was detected.

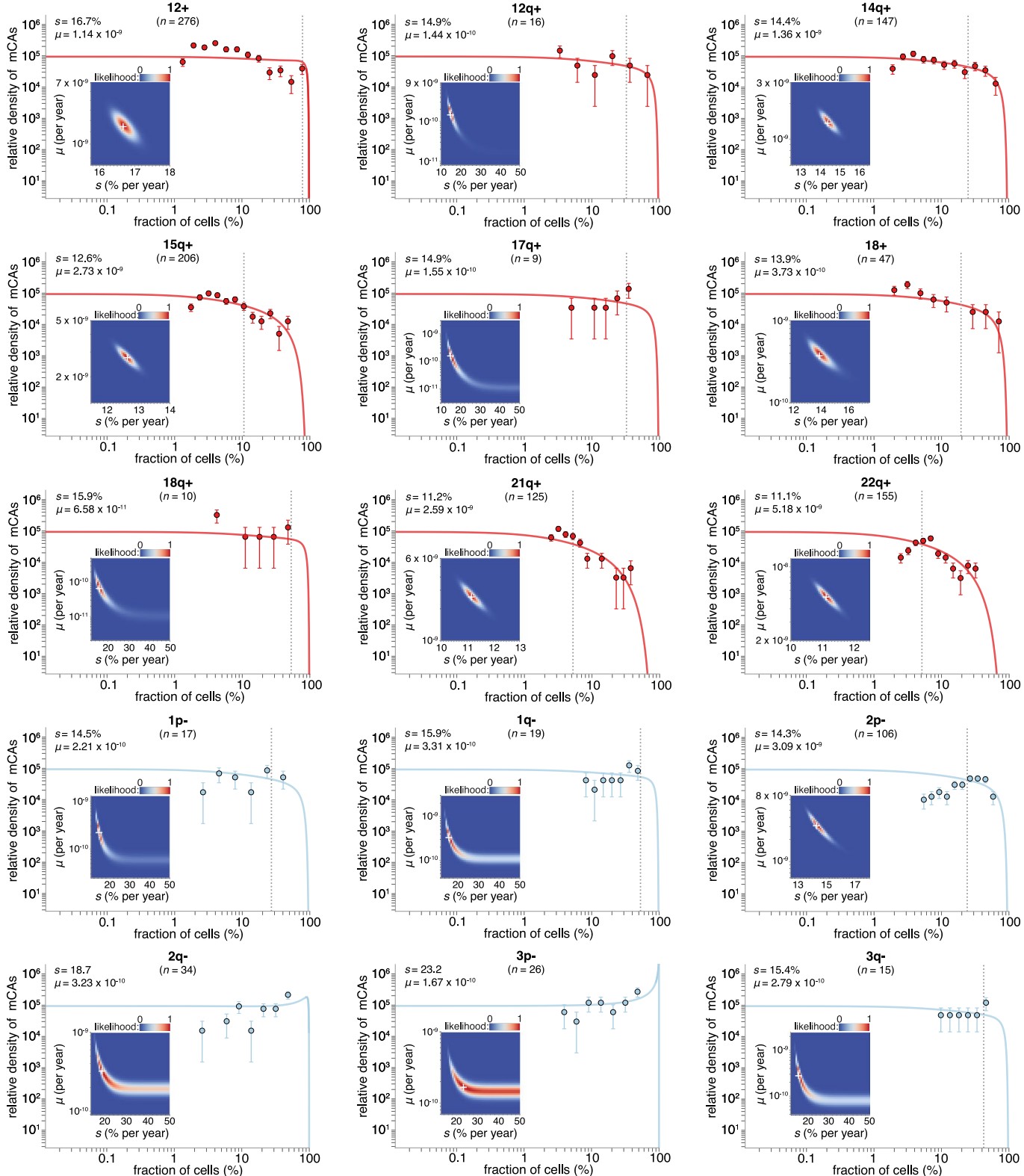

**Extended Data Fig. 2 | Parameter estimation for individual mCAs: part 2.** The cell fraction probability density histogram is shown for each mCA (datapoints) with the theory distribution (solid line) fitted using maximum likelihood approaches (red = gains, blue = losses, yellow = CN-LOH). Datapoints are presented as mean values +/− SEM. Grey vertical dashed line shows the fitted $\phi$ parameter $\left(\frac{e^{st}-1}{Ns}\right)$, where the exponential fall-off in densities occurs. The white cross on the maximum likelihood heatmap marks the most likely mutation rate ($\mu$) and fitness effect ($s$). $n$ = number of individuals (with a single mCA) in whom the mCA was detected. Gains (+) are shown in red, losses (−) are shown in blue.

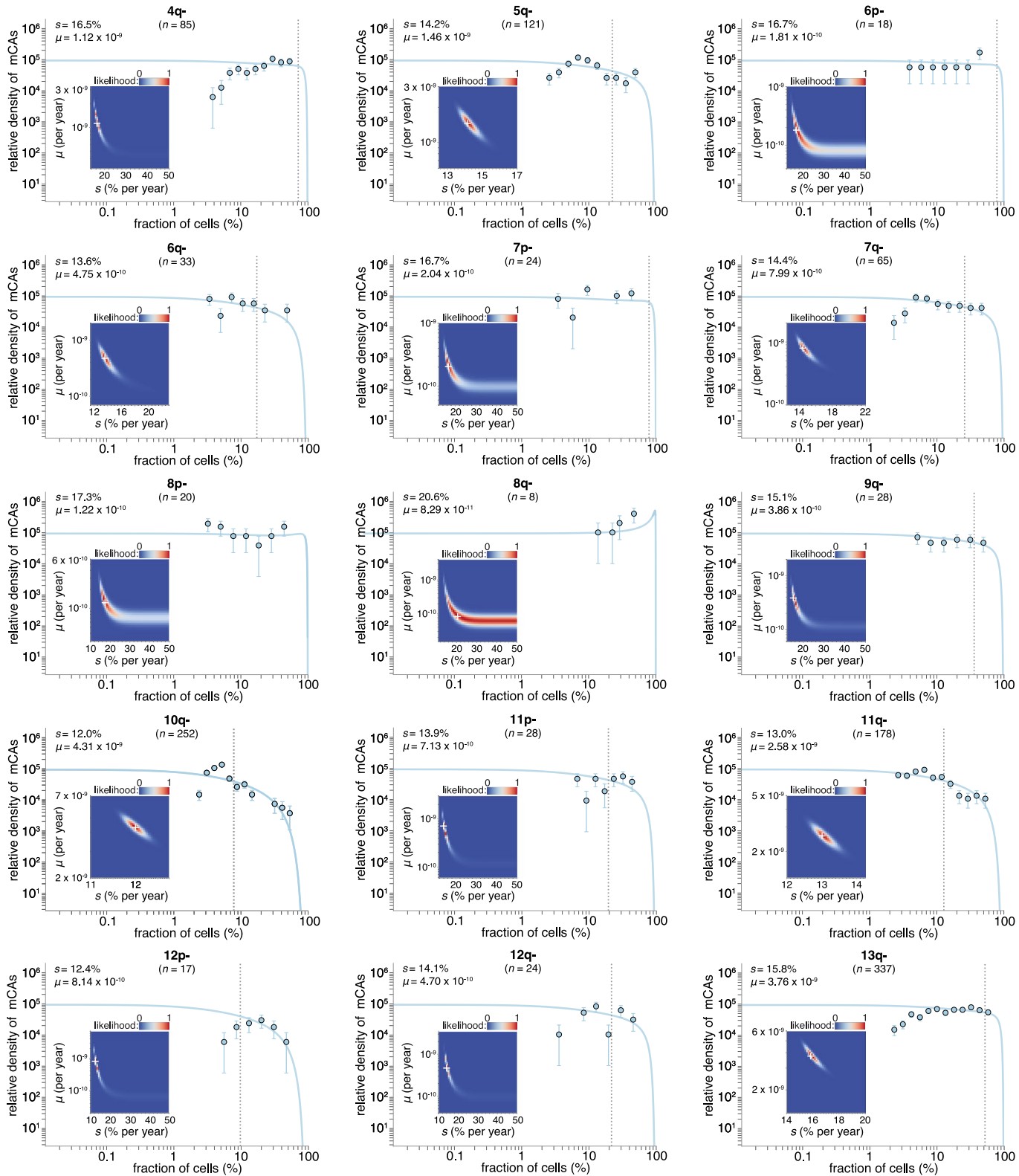

**Extended Data Fig. 3 | Parameter estimation for individual mCAs: part 3.** The cell fraction probability density histogram is shown for each mCA (datapoints) with the theory distribution (solid line) fitted using maximum likelihood approaches (red = gains, blue = losses, yellow = CN-LOH). Datapoints are presented as mean values +/− SEM. Grey vertical dashed line shows the fitted $\phi$ parameter $\left(\frac{e^{st}-1}{Ns}\right)$, where the exponential fall-off in densities occurs. The white cross on the maximum likelihood heatmap marks the most likely mutation rate ($\mu$) and fitness effect ($s$). $n$ = number of individuals (with a single mCA) in whom the mCA was detected.

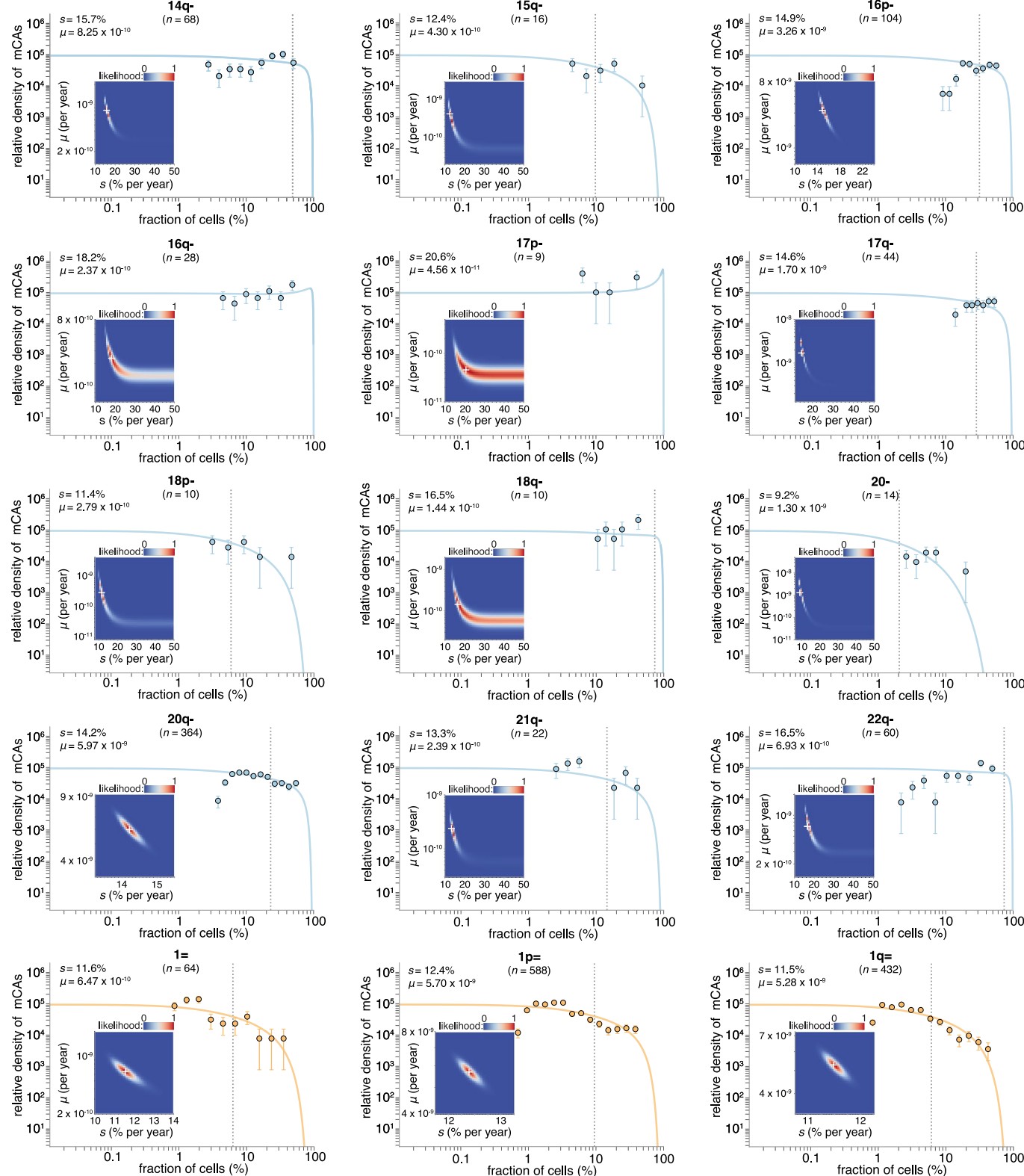

**Extended Data Fig. 4 | Parameter estimation for individual mCAs: part 4.** The cell fraction probability density histogram is shown for each mCA (datapoints) with the theory distribution (solid line) fitted using maximum likelihood approaches (red = gains, blue = losses, yellow = CN-LOH). Datapoints are presented as mean values +/− SEM. Grey vertical dashed line shows the fitted $\phi$ parameter $\left(\frac{e^{st}-1}{Ns}\right)$, where the exponential fall-off in densities occurs. The white cross on the maximum likelihood heatmap marks the most likely mutation rate ($\mu$) and fitness effect ($s$). $n$ = number of individuals (with a single mCA) in whom the mCA was detected. Losses (−) are shown in blue and CN-LOH (=) events are shown in yellow.

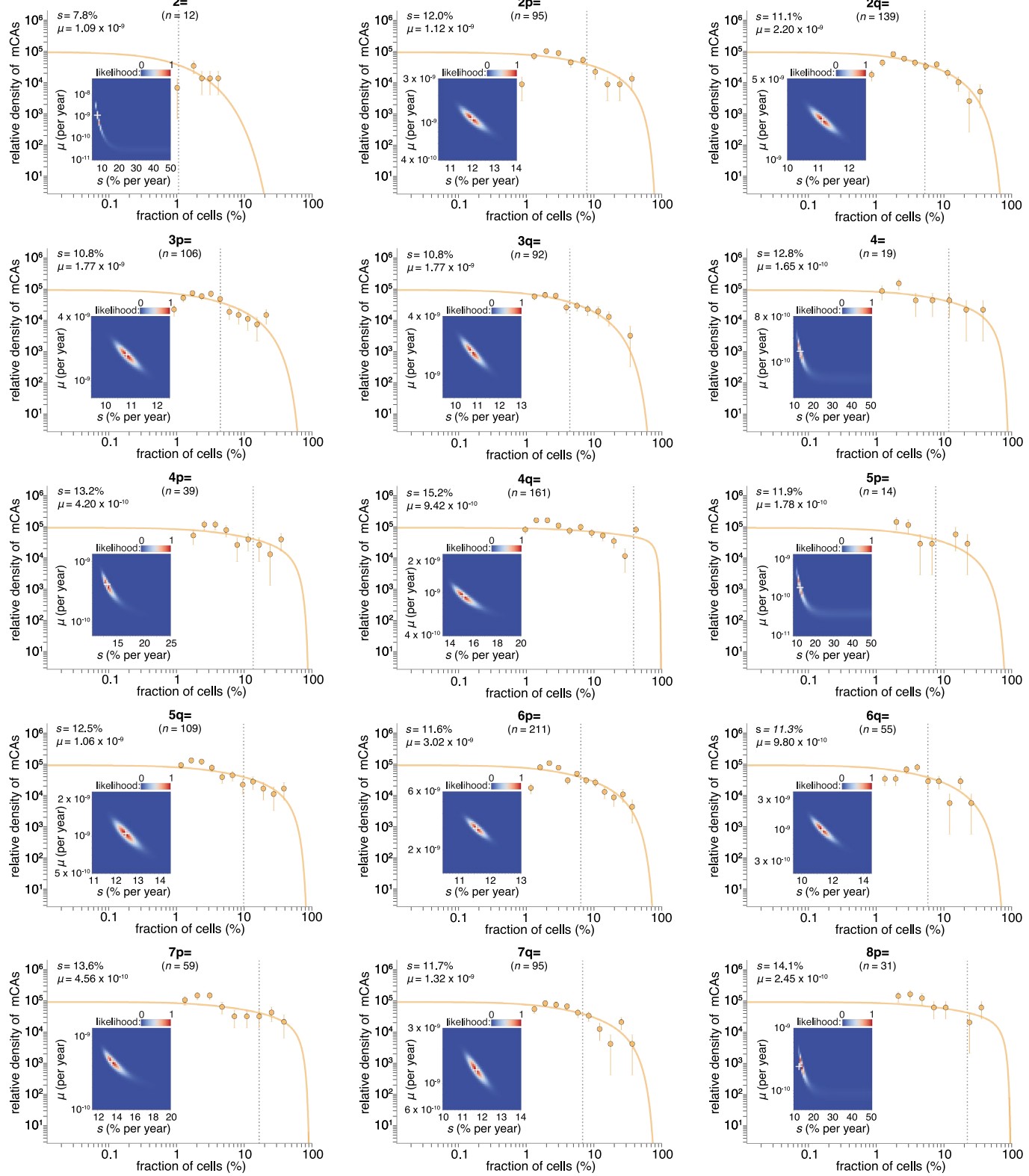

**Extended Data Fig. 5 | Parameter estimation for individual mCAs: part 5.** The cell fraction probability density histogram is shown for each mCA (datapoints) with the theory distribution (solid line) fitted using maximum likelihood approaches (red = gains, blue = losses, yellow = CN-LOH). Datapoints are presented as mean values +/− SEM. Grey vertical dashed line shows the fitted $\phi$ parameter $\left(\frac{e^{st}-1}{Ns}\right)$, where the exponential fall-off in densities occurs. The white cross on the maximum likelihood heatmap marks the most likely mutation rate ($\mu$) and fitness effect ($s$). $n$ = number of individuals (with a single mCA) in whom the mCA was detected.

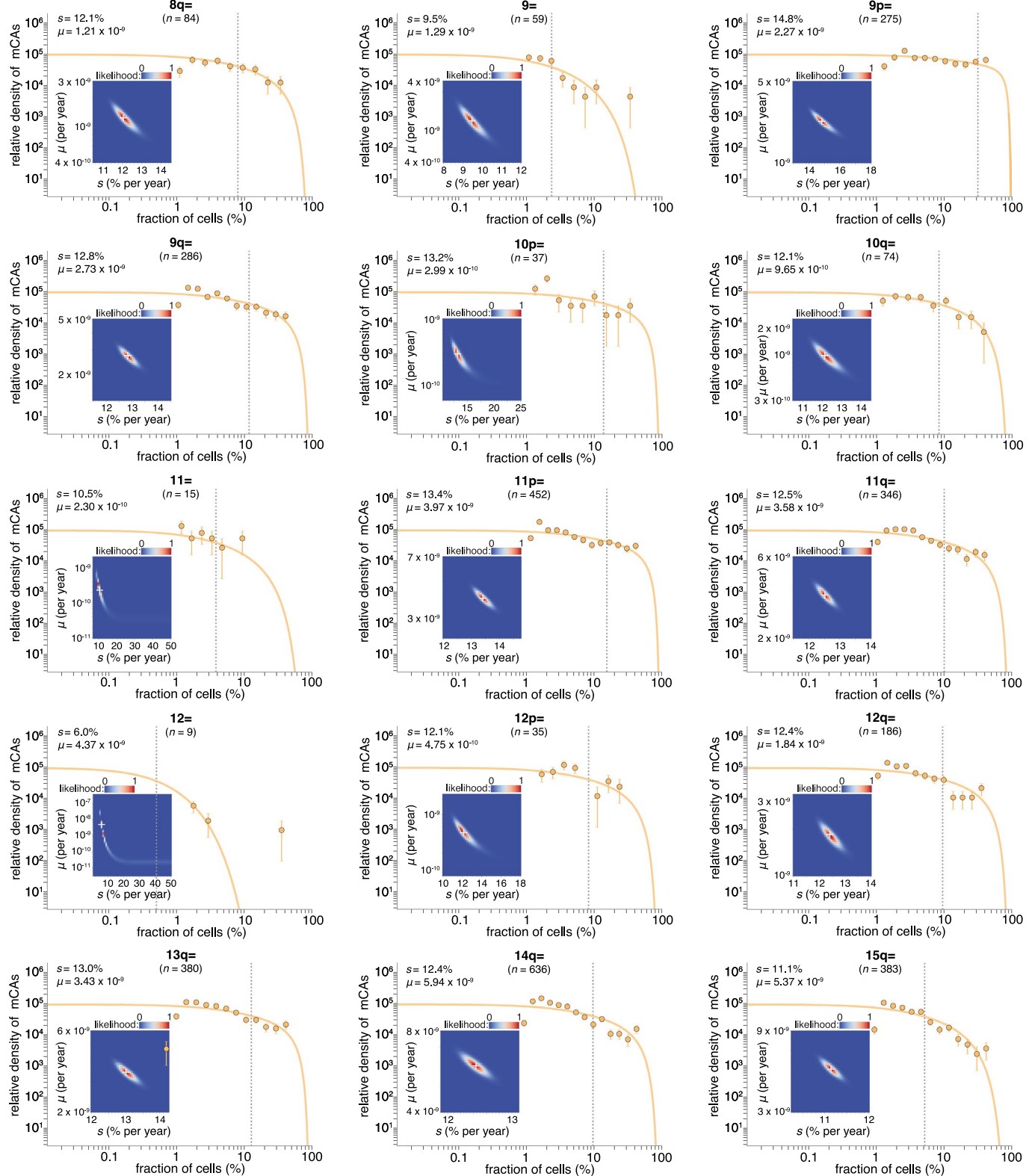

**Extended Data Fig. 6 | Parameter estimation for individual mCAs: part 6.** The cell fraction probability density histogram is shown for each mCA (datapoints) with the theory distribution (solid line) fitted using maximum likelihood approaches (red = gains, blue = losses, yellow = CN-LOH). Datapoints are presented as mean values +/− SEM. Grey vertical dashed line shows the fitted $\phi$ parameter $\left(\frac{e^{St}-1}{Ns}\right)$, where the exponential fall-off in densities occurs. The white cross on the maximum likelihood heatmap marks the most likely mutation rate ($\mu$) and fitness effect ($s$). $n$ = number of individuals (with a single mCA) in whom the mCA was detected.

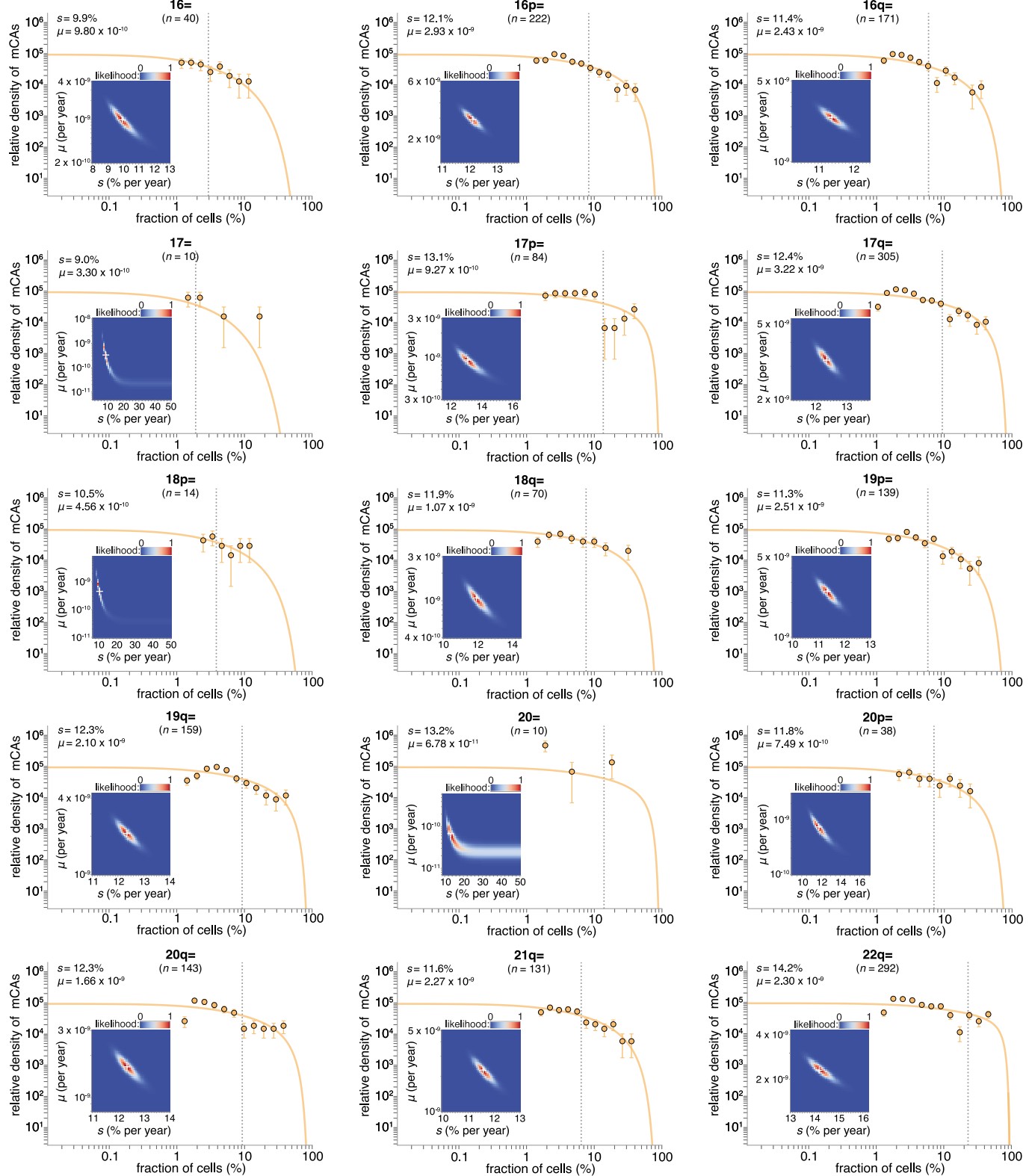

**Extended Data Fig. 7 | Parameter estimation for individual mCAs: part 7.** The cell fraction probability density histogram is shown for each mCA (datapoints) with the theory distribution (solid line) fitted using maximum likelihood approaches (red = gains, blue = losses, yellow = CN-LOH). Datapoints are presented as mean values +/− SEM. Grey vertical dashed line shows the fitted $\phi$ parameter $\left(\frac{e^{st}-1}{Ns}\right)$,

where the exponential fall-off in densities occurs. The white cross on the maximum likelihood heatmap marks the most likely mutation rate ($\mu$) and fitness effect ($s$). $n$ = number of individuals (with a single mCA) in whom the mCA was detected.

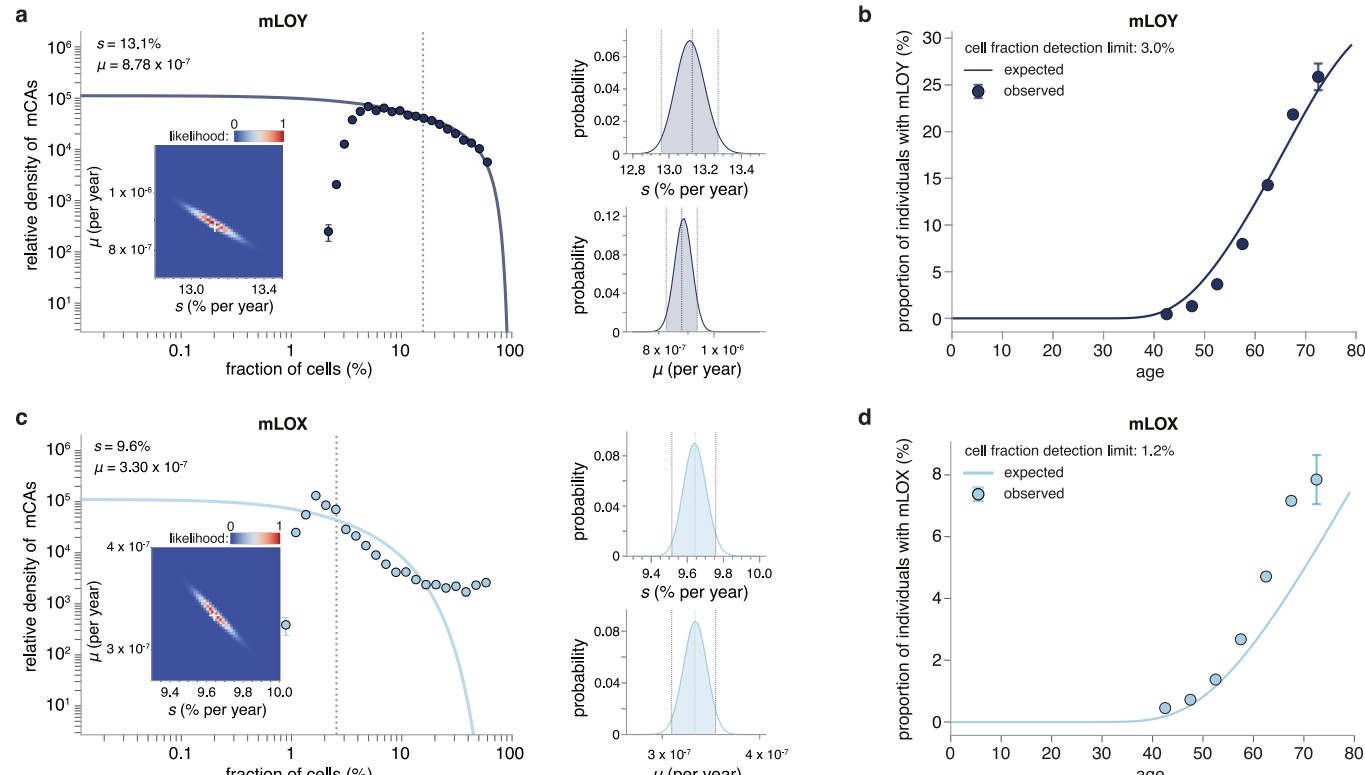

**Extended Data Fig. 8 | Parameter estimation for mLOY and mLOX. a**. The cell fraction probability density histogram is shown for mLOY (datapoints, $n = 22,367$ observations) with the theory distribution (solid line) fitted using maximum likelihood approaches. Datapoints are presented as mean values +/− SEM. Grey vertical dashed line shows the fitted $\phi$ parameter $\left(\frac{e^{st}-1}{Ns}\right)$, where the exponential fall-off in densities occurs. The white cross on the maximum likelihood heatmap marks the most likely mutation rate ($\mu$) and fitness effect ($s$). The two small plots show the distribution of likelihoods for $s$ and $\mu$, with the blue vertical line representing the most likely value. 95% confidence intervals are shown shaded in blue. **b**. Predicted age prevalence of mLOY (solid line) using inferred $s$ and $\mu$, compared to observed age prevalence (datapoints, $n = 22,367$ observations). Error bars represent sampling error (+/− 1 SD). The cell fraction limit of detection, for the predicted age prevalence, was taken as the minimum cell fraction

observed for mLOY, multiplied by 1.5. **c**. The cell fraction probability density histogram is shown for mLOX (datapoints, $n = 8577$ observations) with the theory distribution (solid line) fitted using maximum likelihood approaches. Datapoints are presented as mean values +/− SEM. Grey vertical dashed line shows the fitted $\phi$ parameter $\left(\frac{e^{st}-1}{Ns}\right)$, where the exponential fall-off in densities occurs. The white cross on the maximum likelihood heatmap marks the most likely mutation rate ($\mu$) and fitness effect ($s$). The two small plots show the distribution of likelihoods for $s$ and $\mu$, with the blue vertical line representing the most likely value. 95% confidence intervals are shown shaded in blue. **d**. Predicted age prevalence of mLOX (solid line) using inferred $s$ and $\mu$, compared to observed age prevalence (datapoints, $n = 8577$ observations). Error bars represent sampling error (+/− 1 SD). The cell fraction limit of detection, for the predicted age prevalence, was taken as the minimum cell fraction observed for mLOX, multiplied by 1.5.

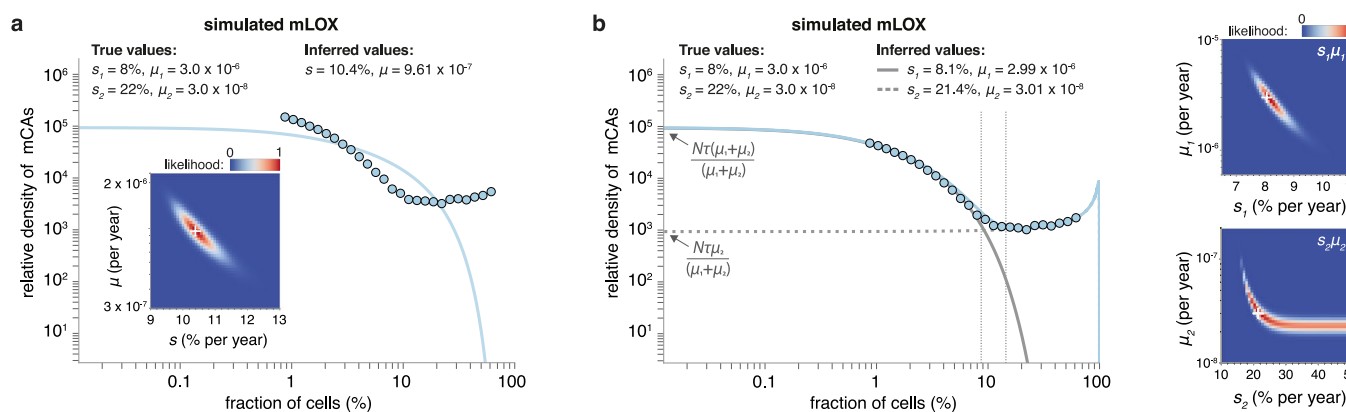

**Extended Data Fig. 9 | Simulated mLOX calls with 2 different mutation rates and fitness effects.** Simulated mLOX calls were generated at a high rate but conferring a low fitness ($\mu_1$,$s_1$) or at a low rate but conferring a high fitness ($\mu_2$,$s_2$). The cell fraction probability density histogram is shown for these mLOX calls (datapoints, $n = 38,198$ mLOX calls). Parameter estimation, using a maximum likelihood approach to fit a single theory distribution (solid line) with a single fitness effect ($s$) and single mutation rate ($\mu$), is shown. Datapoints are presented as mean values +/− SEM. The white cross on the maximum likelihood heatmap marks the most likely inferred $s$ and $\mu$. **b**. Parameter estimation, using a maximum likelihood approach to fit a distribution (solid blue line) made up of 2 distinct mLOX events: one with a high mutation rate ($\mu_1$), but low fitness ($s_1$) (solid grey line) and another with a low mutation rate ($\mu_s$), but high fitness ($s_2$) (dashed grey line). Vertical black dashed lines show the cell fraction split points used for fitting the lower and upper cell fraction datapoints to the theory distribution. Datapoints are presented as mean values +/− SEM. The white cross on the maximum likelihood heatmaps marks the most likely insferred $s_1$,$\mu_1$,$s_2$ and $\mu_2$. $n = 38,198$ mLOX calls.

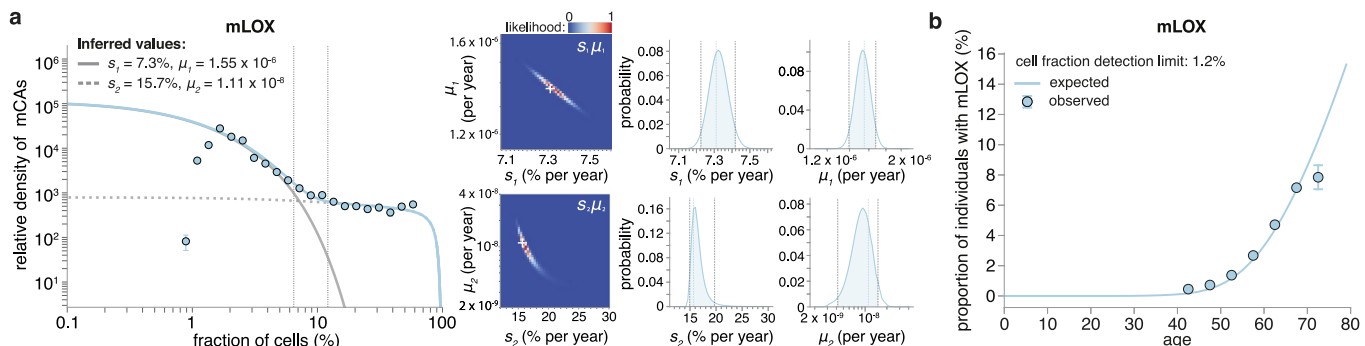

**Extended Data Fig. 10 | mLOX detected among ~260,000 female UK Biobank participants. a**. Parameter estimation for mLOX ($n$ = 8577 observations), using a maximum likelihood approach to fit 2 separate theory distributions (solid line and dashed line), each with their own single fitness effect ($s$) and mutation rate ($\mu$). Vertical black dashed lines show the cell fraction split points used for fitting the lower and upper cell fraction datapoints to the theory distribution. Datapoints are presented as mean values +/− SEM. The white crosses on the maximum likelihood heatmaps marks the most likely $s_1, \mu_1, s_2$ and $\mu_2$. Distribution of likelihoods for $s_1, \mu_1, s_2$ and $\mu_2$ are shown, with the blue vertical line representing the most likely value. 95% confidence intervals are shown shaded in blue. **b**. Predicted age prevalence for mLOX (solid line) using inferred $s$ and $\mu$, compared to observed age prevalence (datapoints, $n$ = 8577 observations). Error bars represent sampling error (+/−1 SD). The cell fraction limit of detection was taken as the minimum cell fraction observed for mLOX, multiplied by 1.5.

# Reporting Summary

## Statistics

For all statistical analyses, confirm that the following items are present in the figure legend, table legend, main text, or Methods section.

| n/a | Confirmed | |
|---|---|---|
| ☐ | ☒ | The exact sample size (*n*) for each experimental group/condition, given as a discrete number and unit of measurement |
| ☒ | ☐ | A statement on whether measurements were taken from distinct samples or whether the same sample was measured repeatedly |
| ☐ | ☒ | The statistical test(s) used AND whether they are one- or two-sided<br>*Only common tests should be described solely by name; describe more complex techniques in the Methods section.* |
| ☒ | ☐ | A description of all covariates tested |
| ☒ | ☐ | A description of any assumptions or corrections, such as tests of normality and adjustment for multiple comparisons |
| ☐ | ☒ | A full description of the statistical parameters including central tendency (e.g. means) or other basic estimates (e.g. regression coefficient) AND variation (e.g. standard deviation) or associated estimates of uncertainty (e.g. confidence intervals) |
| ☐ | ☒ | For null hypothesis testing, the test statistic (e.g. *F*, *t*, *r*) with confidence intervals, effect sizes, degrees of freedom and *P* value noted<br>*Give P values as exact values whenever suitable.* |
| ☒ | ☐ | For Bayesian analysis, information on the choice of priors and Markov chain Monte Carlo settings |
| ☒ | ☐ | For hierarchical and complex designs, identification of the appropriate level for tests and full reporting of outcomes |
| ☐ | ☒ | Estimates of effect sizes (e.g. Cohen's *d*, Pearson's *r*), indicating how they were calculated |

*Our web collection on statistics for biologists contains articles on many of the points above.*

## Software and code

Policy information about availability of computer code

| Data collection | No new data was collected. |
|---|---|
| Data analysis | Custom code used for data analysis is available on Blundell Lab GitHub page: https://github.com/the-blundell-lab/mCA-mutation-rates-fitness-consequences (DOI: 10.5281/zenodo.19662539) . The following publicly available software packages were used in analysis and Figure generation: Jupyter notebook 6.4.5, Python 3.7.4, Pandas 1.3.4, NumPy 1.20.1, Matplotlib 3.4.3, Scipy 1.7.1, csv 1.0, seaborn 0.11.2. |

For manuscripts utilizing custom algorithms or software that are central to the research but not yet described in published literature, software must be made available to editors and reviewers. We strongly encourage code deposition in a community repository (e.g. GitHub). See the Nature Portfolio guidelines for submitting code & software for further information.

## Data

Policy information about availability of data

All manuscripts must include a data availability statement. This statement should provide the following information, where applicable:

- Accession codes, unique identifiers, or web links for publicly available datasets
- A description of any restrictions on data availability
- For clinical datasets or third party data, please ensure that the statement adheres to our policy

The mCA calls used in our analysis are available from UK Biobank (Return 3094), via an application process described at http://www.ukbiobank.ac.uk/using-the-

## Human research participants

Policy information about <u>studies involving human research participants and Sex and Gender in Research.</u>

| | |
|---|---|
| Reporting on sex and gender | Where enough data was available, we calculated sex-specific fitness effects and mutation rates for mCAs (Figure 2c and Supplementary Note 3). Sex was defined as reported by UK Biobank. |
| Population characteristics | The mCA calls were generated by Loh et al (doi: 10.1038/s41586-020-2430-6) using SNP array data from 482,789 UK Biobank participants, out of a total 502,650 UK Biobank participants who had the following characteristics: Average age 56.52 years (range 37-73 years); Sex: 54.4% female, 45.6% male; Ethnicity: 94% White British, 2% Asian or Asian British, 1.6% Black or Black British, 0.6% Mixed, 0.3% Chinese, 0.9% Other. |
| Recruitment | No new study participants were recruited for this research. |
| Ethics oversight | The North West Multi-centre Research Ethics Committee (MREC) reviewed and approved the UK Biobank scientific protocol and operational procedures (REC reference number: 21/NW/0157) and all participants provided signed informed consent at enrolment. |

Note that full information on the approval of the study protocol must also be provided in the manuscript.

# Field-specific reporting

Please select the one below that is the best fit for your research. If you are not sure, read the appropriate sections before making your selection.

☒ Life sciences          ☐ Behavioural & social sciences          ☐ Ecological, evolutionary & environmental sciences

For a reference copy of the document with all sections, see <u>nature.com/documents/nr-reporting-summary-flat.pdf</u>

# Life sciences study design

All studies must disclose on these points even when the disclosure is negative.

| | |
|---|---|
| Sample size | No statistical methods were used to predetermine sample size. The mCA calls used in our analysis (Loh et al 2020 (doi: 10.1038/s41586-020-2430-6) and UK Biobank Return 3094) were generated by analysis of all available UK Biobank samples. |
| Data exclusions | When calling mCAs, Loh et al excluded individuals with low genotyping quality (B-allele frequency s.d. >0.11 at heterozygous sites), individuals with evidence of possible sample contamination, and individuals who had withdrawn consent. For the mLOY calls (from UK Biobank Return 3094), which were based on BAF and LRR measurements in PAR1, we only included those >2MB in size, to reduce the risk of including focal loss events on the Y chromosome. For the mLOX calls (from UK Biobank Return 3094) we only included those >125MB in size, to reduce the risk of including focal loss events on the X chromosome. For fitness effect and mutation rate parameter estimation, mLOX, mLOY and simulated mCA calls were each downsampled to 50 calls (using NumPy 1.20.1 numpy.random.sample with numpy.random.seed(seed = 3, version =2) to reduce compute time. |
| Replication | Not applicable - experimental replication was not attempted in the generation of the mCA calls used in our analysis. |
| Randomization | For fitness effect and mutation rate parameter estimation, mLOX, mLOY and simulated mCA calls were each downsampled to 50 calls (using NumPy 1.20.1 numpy.random.sample with numpy.random.seed(seed = 3, version =2) to reduce compute time. |
| Blinding | Blinding was not performed for this study as the identify of the mCA call was essential for our analysis. |

# Reporting for specific materials, systems and methods

We require information from authors about some types of materials, experimental systems and methods used in many studies. Here, indicate whether each material, system or method listed is relevant to your study. If you are not sure if a list item applies to your research, read the appropriate section before selecting a response.

## Materials & experimental systems

| n/a | Involved in the study |
|-----|------------------------|
| ☒ | Antibodies |
| ☒ | Eukaryotic cell lines |
| ☒ | Palaeontology and archaeology |
| ☒ | Animals and other organisms |
| ☒ | Clinical data |
| ☒ | Dual use research of concern |

## Methods

| n/a | Involved in the study |
|-----|------------------------|
| ☒ | ChIP-seq |
| ☒ | Flow cytometry |
| ☒ | MRI-based neuroimaging |

