## [Peer Review File · Nature Genetics]

Peer Review Information

Manuscript Title: Mutation rates and fitness consequences of mosaic chromosomal alterations in blood.

Corresponding author name(s): Dr Caroline Watson and Dr Jamie Blundell

Reviewer Comments & Decisions:

Decision Letter, initial version:

5th Oct 2022

Dear Dr Blundell,

Your Article, "Mutation rates and fitness consequences of mosaic chromosomal alterations in blood" has now been seen by 3 referees. You will see from their comments below that while they find your work of interest, some important points are raised. We are interested in the possibility of publishing your study in Nature Genetics, but would like to consider your response to these concerns in the form of a revised manuscript before we make a final decision on publication.

To guide the scope of the revisions, the editors discuss the referee reports in detail within the team, including with the chief editor, with a view to identifying key priorities that should be addressed in revision and sometimes overruling referee requests that are deemed beyond the scope of the current study. In this case, we'd like you to address all the comments in full - experimentally where possible or textually where appropriate.

We therefore invite you to revise your manuscript taking into account all reviewer and editor comments. Please highlight all changes in the manuscript text file. At this stage we will need you to upload a copy of the manuscript in MS Word .docx or similar editable format.

*1) Include a “Response to referees” document detailing, point-by-point, how you addressed each referee comment. If no action was taken to address a point, you must provide a compelling argument. This response will be sent back to the referees along with the revised manuscript.

*2) If you have not done so already please begin to revise your manuscript so that it conforms to our Article format instructions, available

[here](http://www.nature.com/ng/authors/article_types/index.html).

*3) Include a revised version of any required Reporting Summary: <https://www.nature.com/documents/nr-reporting-summary.pdf>

We hope to receive your revised manuscript within four to eight weeks. If you cannot send it within this time, please let us know.

Nature Genetics is committed to improving transparency in authorship. As part of our efforts in this direction, we are now requesting that all authors identified as ‘corresponding author’ on published papers create and link their Open Researcher and Contributor Identifier (ORCID) with their account on the Manuscript Tracking System (MTS), prior to acceptance. ORCID helps the scientific community achieve unambiguous attribution of all scholarly contributions. You can create and link your ORCID from the home page of the MTS by clicking on ‘Modify my Springer Nature account’. For more information

please visit www.springernature.com/orcid.

Sincerely,

Safia Danovi
Editor
Nature Genetics

Referee expertise:

Referee #1: CH, population genetics

Referee #2: CH, computational biology/modelling

Referee #3: cancer genomics, cancer evolution

Reviewers' Comments:

Reviewer #1:

Remarks to the Author:

In this manuscript, Watson et al., apply a population genetic framework that they previously developed to estimate mutation rate and fitness effects for CH driven by SNVs which here they apply to CH driven by CNVs using the UK Biobank dataset. They use the CNV calls previously generated by Loh et al using the MoChA algorithm. They descriptively compare the mutation frequency and fitness effects between CNVs and SNVs and find examples of CNVs that deviate from their age-related growth rates suggesting extrinsic factors beyond age might drive the expansion of these events. The manuscript is well-written and clear. This is a novel application of their previously developed method. Many of the findings are interesting and I think could be made even more interesting with some additional analyses. Specifically, it would be worthwhile leveraging the scale of the UKBB to understand mechanisms by which some CNVs show age and sex-related deviation from their predictions. There are also some methodologic concerns I have that I detail below.

The main broad concern I have regarding their methods is the application of their methods to CNV calls given that the sensitivity and detection rate for CNV events is driven by both the size of the event, the cell fraction and the type of the event. The Loh et al method for mCA detection uses a Hidden Markov

model to detect stretches of BAF at heterozygous sites. Thus, smaller events will be less likely to be detected since there are fewer heterozygous SNPs. The caveat here is that for very small events with high BAF deviation these are actually removed as likely germline events. Thus, I would think that the actual mutational frequency for smaller events would be underestimated since these events would be more common than observed. The authors state that CN-LOH events occur at the highest rate. However, as described by Loh et al in supplementary note 1 from PMID: 29995854 these types of events are the easiest to detect. Using simulations perhaps could the authors estimate the sensitivity of detection for different classes of mutational events and how the number of heterozygous sites influences this? Taking into account the sensitivity of detection might change some of their findings. Similarly, is comparing the mutation rate and fitness effects of SNVs to CNVs at face value fair without taking into account the sensitivity of detection?

In order to further understand the basis of the observed sex deviations and deviations in age-dependence could the authors look at how different environmental factors influence copy number events of interest? If they hypothesize that germline genetic factors play a role, perhaps study what germline genetic factors predispose to copy number variants of interest.

Finally, when they describe the association between mCA fitness and cancer risk they state that only 4-9 years of follow-up are available. However, currently, the median follow-up exceeds 10 years. How are they defining hematologic malignancy here?

Reviewer #2:

Remarks to the Author:

In this manuscript the authors analyzed a shared dataset of mosaic chromosomal alterations (mCAs) including 2389 gain (+), 3718 loss (-) and 8185 CN-LOH (=) events within ~500,000 individuals from the UK biobank better described in Loh et al. Nature 2020. They partition events across types (gain/loss/CN-LOH) and chromosome arms and for each bin they estimate mutation rate (μ) and fitness effect (s) starting from the number of events and the reported cell fraction estimates

While the manuscript is well written and the analysis highlights some interesting differences across mCA types not described in Loh et al. Nature 2020, I find the results of the model of limited utility from the point of view of predictability and I have been wondering across the manuscript whether a more descriptive approach of the distribution of cell fractions across mCA types would be better suited than the additional complexity from the presented approach

Ultimately though, an old saying goes that models are all wrong and they are only as good as what they can predict. The model presented is quite complicated and in this manuscript it only predicts the odds ratio for any blood cancer (Fig. 5 which is likely simply the result of CN-LOHs being observed at lower cell fractions). The identified sex differences for selected mCAs are something that was already observed in Loh et al. 2020 and it can be inferred without estimating mutation rate and fitness effect

Major comments:

What is the correlation between the estimated mutation rate (μ) and the prevalence of each event? What is the correlation between the estimated fitness effect (s) and the fraction of events with cell fraction higher than 5%? I do wonder here if μ and s are very similar to the two other more descriptive statistics and whether it would be better to use the descriptive statistics. Maybe reporting a comparison would help better understand the importance of μ and s

The authors never discuss how the sampling bias of mCAs can affect their model. CN-LOHs for example are larger and at low cell fractions they cause deviations in the BAF twice as large as deletions and duplications, as they affect two haplotypes since they can be described as a duplication and a deletion at the same time

At the same time heterozygosity 'dropping out', which is shortly explained in Supplementary Material 1, can also bias sampling of events at large cell fractions. For example, an event at 50% cell fraction would cause a BAF deviation, of 10%, 16.7%, 25% for, respectively, a duplication, a deletion, and a CN-LOH. Could this affect the fitness effect? Are the authors sure that their model is robust against these biases? I did not feel sure at all after reading the manuscript and I was particularly concerned that many of the outcomes could be driven by technicalities rather than biological phenomena

A scatterplot showing the two-parameter estimates for duplication, deletions, and CN-LOHs would be nice (maybe instead of panel Fig. 2b) to better assess how differently these three categories were fitted

Minor comments:

Introduction: Our estimates reveal that highly fit mCAs (growth rates $\geq 10\%$ per year) occur at a rate of ~ 1 per 10 million cells per year, approximately 10-fold lower than equivalently fit SNVs. While occurring at a relatively low rate, the fitness consequences of these mutations can be dramatic, expanding at rates of up to 15-20% per year. <- This should be in the discussion rather than in the introduction

Mutation rates and fitness effects of mCAs: the fitness effect of losses are systematically higher, with most fitness effects being between $\sim 14\text{-}20\%$ per year. However, as a class, losses occur at a combined rate of $\sim 4 \times 10^{-8}$ per cell per year, 2.3-fold lower than CN-LOH. Gains appear to have a broad range of fitness effects, but occur at the lowest combined mutation rate of $\sim 2 \times 10^{-8}$ per cell per year <- how do we really know that this is not in large part due to the differential rate of detectability of different types of events? Certainly CN-LOHs are measured with a higher prevalence just due to the fact that they are easier to detect

Sex differences in fitness effects and mutation rates: we infer that the observed higher prevalence of 10q- in women is due to a ~ 4 -fold higher mutation rate in women, with limited evidence for any sex bias in fitness effect <- as there is no association with age with this is likely simply that the prevalence of the event is ~ 4 times higher in females than males and Extended Data Fig. 1b in Loh et al. 2020 makes that

already abundantly clear. Furthermore, this event is observed in relation to FRA10B expansions and so it seems weird to try to fit it into a model that does not take into account the prevalence of long FRA10B germline haplotypes

Age dependence of mCAs: Overall, the observed prevalence of gain and loss events in both men and women is in close agreement with the predicted prevalence <- this seems quite subjective ... is there really close agreement?

Fig. 4 <- this figure again seems to reiterate the CN-LOH detection bias from Loh et al. 2020 and, since it cannot be independently tested, it seems inappropriate to show. Also ascertained rates of detected mCAs in Japan (see Figure 2b from Terao et al. Nature 2020) Finland (see Supplementary Figure S3 from Koskela et al. medRxiv 2021), which has a wider range of age in participants, make it clear that the true prevalence is severely underestimated by Fig. 4

mCA fitness effects and cancer risk: we do indeed find a significant correlation between the fitness effect and odds ratio of subsequent blood cancer (Pearson $R = 0.6$, $p = 0.03$) <- I suppose this is not significant within each type (gain/loss/CN-LOH). Again, I find hard to believe this is not simply an artifact of the fact that CN-LOH are detected at lower cell fractions

Fit mCAs occur at a lower rate relative to fit SNVs: Strikingly, the total mutation rate to highly fit mCAs ($s > 10\%$ per year) is over 10-fold lower than the total mutation rate to highly fit SNVs <- isn't this just restating the fact that SNV mutations (at least those in key DNMT3A, TET2, ASXL1, PPM1D, JAK2 genes) are much more prevalent than mCAs?

Cell-extrinsic influences and age-specific effects: 5q- shows a higher mutation rate in women, but a higher fitness effect in men, suggesting extrinsic effects play a role <- could this just be statistical noise? How do I know that this observation is robust to random sampling? As presented it is not very believable. There should be a p-values associated with statements like this that survive multiple hypotheses correction

Fig. S1 <- FRA10B seems localized in the wrong place

Reviewer #3:

Remarks to the Author:

Watson and Blundell present an analysis entitled 'Mutation rates and fitness consequences of mosaic chromosomal alterations in blood'. Based on existing autosomal mCA calls from the UK Biobank the authors adjust their existing population genetics framework to calculate mutation rates and fitness coefficients (growth rates) of clones harbouring somatic mCAs. This is a novel and powerful analysis revealing that mCAs exhibit growth rates similar to those seen for CG driver gene mutations, but mutate

at around 10x lower rate. The authors also demonstrate that while many mCAs exhibit the expected age incidence, some do not, and thus warrant further analysis.

Overall I congratulate the authors for a timely and insightful analysis. The paper, figures and supplementary figures are clearly presented. The comprehensive analysis fills a knowledge gap in our current understanding of the dynamics of CH.

I only have a few comments and questions motivated by curiosity rather than fundamental concerns.

My main question would be how the authors deal with population heterogeneity introduced by driver gene mutation driven CH, germline predisposition or other factors. While this is partially discussed in the manuscript and the authors show the clustering of alterations in fewer than expected individuals, I'd be interested to see a further investigation.

How many of the mCA calls overlap with known driver gene mutations? It's intriguing to see the high fitness effects associated with mCAs and I wondered whether any occur on the back of a driver gene mutated CH case (which reaches frequencies of around 10% in the UKB). Perhaps data from this study could be used: <http://dx.doi.org/10.1038/s41588-022-01121-z>.

Another analysis could include an investigation of individuals with >1 alterations (about 10% judging from Figure S4) and whether this leads to higher growth rates.

Furthermore, it could be interesting to assess whether growth is altered in a population subset carrying specific alleles found to promote the prevalence of CH with mCA. Examples include TCL1A, FRA1 and other variants, both reported in <http://dx.doi.org/10.1038/ng.3821> and <http://dx.doi.org/10.1038/s41586-018-0321-x>.

On a different note, the analysis only considers only autosomal mCAs. It could be insightful to complement the analysis with an investigation of alterations of the allosomes. Data should also be available, as this has been investigated, eg. in <http://dx.doi.org/10.1038/ng.3821>. This could be especially interesting as mLOY is very prevalent in ageing men.

Author Rebuttal to Initial comments

See Attached PDF

Mutation rates and fitness consequences of mosaic chromosomal alterations in blood: Response to reviewers' comments

We thank the editor and all 3 reviewers for their constructive comments and questions. We address each of the reviewers' comments below.

Reviewer 1

"In this manuscript, Watson et al., apply a population genetic framework that they previously developed to estimate mutation rate and fitness effects for CH driven by SNVs which here they apply to CH driven by CNVs using the UK Biobank dataset. They use the CNV calls previously generated by Loh et al using the MoChA algorithm. They descriptively compare the mutation frequency and fitness effects between CNVs and SNVs and find examples of CNVs that deviate from their age-related growth rates suggesting extrinsic factors beyond age might drive the expansion of these events. The manuscript is well-written and clear. This is a novel application of their previously developed method. Many of the findings are interesting and I think could be made even more interesting with some additional analyses. Specifically, it would be worthwhile leveraging the scale of the UKBB to understand mechanisms by which some CNVs show age and sex-related deviation from their predictions."

response 1

We appreciate these constructive comments from the reviewer and outline detailed responses below.

"There are also some methodologic concerns I have that I detail below.

The main broad concern I have regarding their methods is the application of their methods to CNV calls given that the sensitivity and detection rate for CNV events is driven by both the size of the event, the cell fraction and the type of the event. The Loh et al method for mCA detection uses a Hidden Markov model to detect stretches of BAF at heterozygous sites. Thus, smaller events will be less likely to be detected since there are fewer heterozygous SNPs. The caveat here is that for very small events with high BAF deviation these are actually removed as likely germline events. Thus, I would think that the actual mutational frequency for smaller events would be underestimated since these events would be more common than observed. The authors state that CN-LOH events occur at the highest rate. However, as described by Loh et al in supplementary note 1 from PMID: 29995854 these types of events are the easiest to detect. Using simulations perhaps could the authors estimate the sensitivity of detection for different classes of mutational events and how the number of heterozygous sites influences this? Taking into account the sensitivity of detection might change some of their findings. Similarly, is comparing the mutation rate and fitness effects of SNVs to CNVs at face value fair without taking into account the sensitivity of detection?"

response 2

The reviewer is correct that smaller mCAs are harder to detect and thus these events would potentially have a higher false negative rates in the data. However our method is largely insensitive to these issues. In fact this is a key strength of our framework. Instead of inferring rates by counting the number of events, our approach jointly estimates mutation rate (μ) and fitness effect (s) by fitting the predicted distribution (governed by these two parameters and the distribution of ages). It therefore integrates joint information about how many mCA events there are in each frequency interval. This is important because it means our estimates are not dominated by the large number of events at low cell fractions where false negative rates would have a large effect on estimates.

To demonstrate that our method is robust to false negatives at low cell fraction we repeated our analysis, but only included mCAs whose cell fraction were high enough that they were not substantially affected by false negatives. Our inferred estimates of s and μ were largely unchanged, showing that our method is robust to false negatives at low cell fraction. We have included this analysis (Supplementary Fig. 28) within a new section in the supplement (Supplementary Material 3: Effect of mCA detection limits on fitness and mutation rate estimates) (shown below), which also includes an analysis of the effects of lower and upper cell fraction limits of detection on our ability to infer mCA fitness effects and mutation rates.

To summarise, our approach is not very sensitive to the total number of events but rather the precise number of events at each frequency interval. Since most higher frequency intervals are not affected by false negatives these issues do not substantially affect our inferences.

addition to supplement: Supplementary Material 3:

Effect of mCA detection limits on fitness and mutation rate estimates.**mCA cell fraction detection limits**

The upper cell fraction limit of detection, for the mCA calls generated by Loh et al¹⁴, was 67% for losses and 54% for CN-LOH events, corresponding to BAF deviations >0.25 . This was due to the analytical approach used for calling, which resulted in heterozygous SNPs 'dropping out' of the data if BAF deviations were >0.25 . The lower cell fraction limit of detection depends on both the mCA length (with longer events being easier to detect) and the mCA type, with CN-LOH producing greater BAF deviations than loss and gain events of the same cell fraction. In order to provide a conservative estimate for the lower limit of detection for an mCA of a given class, e.g. CN-LOH, we recorded the lowest cell fraction detected for each mCA in the class, multiplied this by 1.5 and took the maximum of these values (Supplementary Table 4). Because we are taking the maximum of the lower thresholds, we expect that the false negative rate for most chromosomes will be low.

Supplementary Table 4: mCA-class specific lower and upper cell fraction limits of detection. The lowest detected cell fraction for each mCA in the class, multiplied by 1.5 (to reduce the false negative rate), was calculated and the maximum of these values, across all mCAs in the class, was used as the mCA-class specific lower limit of detection.

	Gain	Losses	CN-LOH
mCA class-specific lower cell fraction limit of detection	2.5%	4.1%	1.5%
mCA class-specific upper cell fraction limit of detection	100%	67%	54%

Simulated data to determine effect of mCA detection limits

In order to assess how cell fraction detection limits impact our estimates of fitness effects and mutation rates, we performed extensive simulations of the process. Our framework is able to accurately recover the ground-truth values for both fitness and mutation rate across a wide-range of parameter values. This range is slightly reduced because of cell fraction detection limits (outlined below), but these simulations show that we expect the vast majority of our inferences to be accurate.

We generated simulated mCA calls using a custom written stochastic process implemented in Python. Briefly:

- The code simulates a population (`population`), which is a list of all unique clones, and their abundances, present in the population at a given time.
- Clones are tuples composed of (`clone_ID`, `clone_size`) where `clone_size` is the integer number of cells comprising the clone, and, where `clone_ID` is a list of (`mutation_ID`, `fitness_effect`) pairs for all unique mutations that have accumulated in that clone.
- `mutation_ID` uniquely labels each independent mutation to have entered the population and is updated via a counter called `last_mutation_ID`, thus a `mutation_ID=5` means that was the 5th mutation to occur in the population.
- `fitness_effect` is the selective effect, s , of the mutation and is defined at the start of the simulation. All mutated clones generated by the simulation will have the same defined fitness effect.
- The dynamics are implemented using two functions: `mutate` (which generate new clones) and `divide` (which modifies the clone sizes of existing clones) in discrete time steps, where units of time are measured in years.
- The function `mutate` creates new clones by querying each clone in the current population and determining the number of daughter clones each gives rise to by drawing a Poisson random variate with mean $\text{clone_size} \times \mu \times dt$, where $\mu = 4.35 \times 10^{-9}$ is the mutation rate for each cell in the clone. New clones are added to the list of all clones with an initial `clone_size` = 1. In this simple simulation, further mutations cannot occur if the clone already carries a mutation.
- The function `divide` updates the clone sizes of existing clones determining the difference in the number of cell-births and cell-deaths occurring in that clone. In the time interval dt these are calculated via

$$\text{births} = X(\lambda \times B \times \text{clone_size} \times dt)$$

where $X(g)$ is a Poisson random variate with mean g and where $\lambda = 5$ is the total cell division rate (i.e. we assume 5 HSC divisions per year. Similarly the number of deaths is

$$\text{deaths} = X(\lambda \times D \times \text{clone_size} \times dt)$$

addition to supplement: Supplementary Material 3: *cont...*

....

The fitness differences are implemented by allowing birth-rates to depend on the fitness of the mutations in the clone:

$$B = B_0 \times (1 + F)$$

where $D = B_0 = 0.2$. Thus, when there is no fitness advantage ($F = 0$) the clone performs neutral drift due to matched birth and death rates i.e. one in every 5 cell divisions results in a self-renewal, and another 1 in 5 results in a symmetric differentiation event. The total fitness advantage (F) is determined by summing the fitness effects of all the mutations the clone has acquired ($F = \sum_i s_i$). In this simple simulation, where further mutations cannot occur if a clone already carries a mutation, F is simply equal to the mutation fitness effect defined at the start of the simulation.

- The `clone_size` and `population_size` are then recorded at any time (age) required and are used to calculate the cell fraction, `cell_fraction`, where `cell_fraction = clone_size/population_size`.

For each integer fitness effect in the range 7% to 35% per year, we generated simulated mCA calls across ~500,000 individuals, using the actual ages from UK Biobank. At fitness effects <7%, there were not enough mCA calls across the ~500,000 individuals to generate a meaningful cell fraction density histogram to infer the fitness effects and mutation rates.

Effect of mCA detection limits on fitness and mutation rate inferences

If the exponential fall-off in cell fraction (ϕ in eq. 1) occurs at a higher cell fraction than the upper cell fraction limit of detection for the mCA, this might be expected to impair our ability to accurately infer the fitness effect of that mCA. Similarly, if ϕ occurs at a lower cell fraction than the lower cell fraction limit of detection for the mCA, this might be expected to impair our ability to infer both the fitness effect and the mutation rate of the mCA. To test this, we used our simulated mCA calls to assess the effect of the lower and upper cell fraction limit of detection on i) our fitness effect inferences (Supplementary Fig. 15, Supplementary Table 5), and ii) our mutation rate inferences (Supplementary Fig. 16). Fitness effects and mutation rates were inferred as described in Supplementary Material 2.

Fitness effect inferences

When there is no upper cell fraction limit of detection (Supplementary Fig. 15a) (as is the case for gains), we are able to reliably infer the fitness effect of mCAs to at least the maximum simulated fitness effect (35% per year), although the upper error bars become large for fitness effects $\geq 26\%$ per year. When there is an upper cell fraction limit of detection this affects our ability to infer the fitness effect of high fitness mCAs (Supplementary Fig. 15b). For CN-LOH events, which have an upper cell fraction limit of detection of 54%, we can reliably infer the fitness effect up to 21% per year. For loss events, which have an upper cell fraction limit of detection of 67% per year, we can reliably infer the fitness effect up to 28% per year.

A lower cell fraction limit of detection affects our ability to infer the fitness effect of low fitness mCAs (Supplementary Fig. 15a). For CN-LOH and gain events, which have a lower cell fraction limits of detection of 1.5% and 2.5% respectively, we can infer the fitness effect down to 7% per year. For loss events, which have a lower cell fraction limit of detection of 4.1%, we can infer the fitness effect down to 9% per year.

Supplementary Table 5: mCA class-specific limits for fitness effect inferences. For ~500,000 individuals with age-distribution as per UK Biobank, with cell fraction limits of detection as per Supplementary Table 8.

	lowest inferable fitness effect	highest inferable fitness effect
gains	7%	$\geq 35\%$
losses	9%	28%
CN-LOH	7%	21%

Mutation rate inferences

Whether or not there is an upper cell fraction limit of detection makes little difference to our ability to reliably infer mutation rates (Supplementary Fig. 16a-b), because our framework uses the amplitude of cell fraction densities at low cell fractions to infer mutation rates. The lower cell fraction limit of detection and the fitness effect of the mCA affects how many mCAs we see at low cell fractions, which is reflected by the slight decrease in accuracy of our mutation rate inferences at fitness effects $\leq 10\%$ per year (Supplementary Fig. 16a-b). Nonetheless, for our lowest fitness mCAs (Fig. 2, CN-LOH at fitness effects of $\leq 9\%$ per year), our mutation rate inferences are still accurate to within a factor of 2.

addition to supplement: Supplementary Material 3: cont...

....

Supplementary Figure 15: Effect of upper and lower mCA detection limits on fitness inferences. **a.** Actual and inferred fitness effects for simulated data with no upper cell fraction limit of detection, but with mCA-class-specific lower cell fraction limits of detection. **b.** Actual and inferred fitness effects for simulated data with mCA-class-specific upper cell fraction limits of detection and mCA-class-specific lower cell fraction limits of detection. Gains are not shown because their mCA-class-specific upper cell fraction limit of detection is 100%. Error bars show 95% confidence intervals.

Supplementary Figure 16: Effect of upper and lower mCA detection limits on mutation rate inferences, according to mCA fitness effect. **a.** Inferred mutation rates for simulated data with no upper cell fraction limit of detection, but with mCA-class-specific lower cell fraction limits of detection. The actual mutation rate (4.35×10^{-9} per year) is represented by the horizontal dashed line. **b.** Inferred mutation rates for simulated data with mCA-class-specific upper cell fraction limits of detection and mCA-class-specific lower cell fraction limits of detection. The actual mutation rate (4.35×10^{-9} per year) is represented by the horizontal dashed line. Gains are not shown because their mCA-class-specific upper cell fraction limit of detection is 100%. Error bars show 95% confidence intervals.

addition to supplement: Supplementary Material 3: cont...

....

Effect of false negatives at low cell fraction on fitness and mutation rate inferences

Smaller mCAs are harder to detect and so these events could potentially have a higher false negative rates in the data. A key strength of our framework is that it is largely insensitive to these issues. By fitting the predicted distribution (governed by mutation rate and fitness effect), our framework integrates joint information about how many mCA events there are in each cell fraction interval. This means our estimates are not dominated by the large number of events at low cell fractions where false negative rates would have a large effect on estimates.

To demonstrate that our method is robust to false negatives at low cell fraction, we reanalysed all the (non-simulated) mCA data, but only included mCAs whose cell fractions were high enough that they were not substantially affected by false negatives (Supplementary Fig. 17). This shows that our inferences are robust even in the presence of false negatives at low cell fraction (Supplementary Fig. 17d-e).

Supplementary Figure 17: Effect of false negatives at low cell fraction on fitness and mutation rate inferences. a-c. Cell fraction density histograms for all gains (a), losses (b) and CN-LOH events (c). Vertical dashed lines indicate the cell fraction below which the density of mCAs starts to fall off and thus likely represents the limit of of reliable mCA detection where false negatives become more prevalent. mCAs at cell fractions lower than this cut-off were excluded to assess what effect this had on fitness and mutation rate inferences. d-e. Correlation between mutation rate (μ) (d) and fitness (s) inferences (e) with and without data trimming at the cell fractions highlighted in a-c. Faded data points represent mCAs where the total number of data points was <8 after the lower cell fraction mCAs were trimmed (i.e. below the number of datapoints required to form a meaningful cell fraction density histogram).

“In order to further understand the basis of the observed sex deviations and deviations in age-dependence could the authors look at how different environmental factors influence copy number events of interest? If they hypothesize that germline genetic factors play a role, perhaps study what germline genetic factors predispose to copy number variants of interest.”

response 3

This is an excellent suggestion. We agree that undertaking a systematic analysis of mCAs with unexpected age-dependence and their interaction with environmental and/or genetic factors is an important area of research. We have updated Fig. 3 to better highlight mCAs which exhibit unexpected age dependence (now ranking the mCAs by deviation from expected age dependence) to enable further study (as shown below). The original Loh studies (2018, 2020) did undertake an analysis of whether prevalence of mCAs was associated with germline variants. Interestingly 6 of the 7 mCAs (1p CN-LOH (associated with *MPL* in *cis*), 1q CN-LOH (associated with *FH* in *cis*), 8q CN-LOH (associated with *NBN* in *cis*), 10q- loss (associated with *FRA10B* in *cis*), 12q CN-LOH (associated with *SH2B3* in *cis*), 15q CN-LOH (associated with *TM2D3* and *TARLS2* in *cis*) that Loh et al find associated with inherited variants exhibit significant deviation from the age dependence predicted by our model.

changes to main text figure:

Fig. 3: Age dependence of mCAs. a-c, Observed and expected prevalence of gains (a), losses (b) and CN-LOH (c) events for men and women. Expected prevalence (solid lines) calculated by summing the expected prevalence of each mCA in the mCA class. d, Deviation from expected age-dependence for each mCA observed ≥ 30 times in men and ≥ 30 times in women, with examples from each mCA class (see Supplementary Material 5 for age dependence plots for all mCAs). Deviation from expected age-dependence calculated as the relative difference between the observed and expected age dependence gradients. Positive deviation reflects mCAs which show more increase in prevalence with age than expected. Negative deviation reflects mCAs which show less increase in prevalence with age than expected. mCAs whose deviation was statistically significant ($p < 0.05$) are highlighted with an asterisk (see Supplementary Material 5).

“Finally, when they describe the association between mCA fitness and cancer risk they state that only 4-9 years of follow-up are available. However, currently, the median follow-up exceeds 10 years. How are they defining hematologic malignancy here?”

response 4

The data on subsequent haematological cancer diagnosis in individuals with mCAs was obtained from the Loh 2020 analysis, in which 4-9 years of follow-up was available. The definition of haematologic malignancy is any blood cancer. We agree that going back and using the additional 4 years of follow-up would possibly increase statistical power in the association between fitness effect and odds ratio of blood cancer. However given our access agreement with the biobank this would incur substantial delays and add only a modest amount of data. We have, however, updated Fig. 4 to now include all autosomal mCAs with a non-zero odds ratio of any blood cancer (as shown below).

changes to main text and figure:

mCA fitness effects and cancer risk

Because the growth rate of an mCA could, in part, control the probability of acquiring subsequent driver mutations, we reasoned that the fitness effect of mCAs may be correlated with their subsequent risk of haematological malignancy. Despite 98 mCAs having fitness effects $>10\%$, however, only 20 of these were significantly associated with subsequent haematological malignancy diagnosis during 4-9 years of UK Biobank follow-up¹⁴. While this length of follow-up likely means we are underpowered to detect all mCAs associated with cancer risk, it could be that some mCAs drive benign clonal expansions, as has been observed in other tissues, e.g. NOTCH1 in the oesophagus²⁰. Nonetheless, conditioning on mCAs that have a non-zero odds ratio of haematological malignancy, we do indeed find a significant correlation between the fitness effect and odds ratio of any subsequent blood cancer (Pearson $R = 0.59$, $p = 4.9 \times 10^{-5}$) (Fig. 4).

Fig. 4: mCA fitness effects and blood cancer risk. The relationship between inferred fitness effect and odds ratio (OR) of any blood cancer is shown for autosomal mCAs with a non-zero OR (14) and which were observed in ≥ 30 individuals. The mCAs highlighted in bold are those that Loh et al determined to have a statistically significant increased risk of any blood cancer¹⁴. Pearson correlation coefficient and 95% confidence intervals (grey shaded area) are shown. The blood cancers were diagnosed >1 year after DNA collection (within 4-9 years follow-up) in individuals with no previous cancer and with no restriction on blood counts at assessment.

Reviewer 2

"In this manuscript the authors analyzed a shared dataset of mosaic chromosomal alterations (mCAs) including 2389 gain (+), 3718 loss (-) and 8185 CN-LOH (=) events within 500,000 individuals from the UK biobank better described in Loh et al. *Nature* 2020. They partition events across types (gain/loss/CN-LOH) and chromosome arms and for each bin they estimate mutation rate (μ) and fitness effect (s) starting from the number of events and the reported cell fraction estimates.

While the manuscript is well written and the analysis highlights some interesting differences across mCA types not described in Loh et al. *Nature* 2020, I find the results of the model of limited utility from the point of view of predictability and I have been wondering across the manuscript whether a more descriptive approach of the distribution of cell fractions across mCA types would be better suited than the additional complexity from the presented approach

Ultimately though, an old saying goes that models are all wrong and they are only as good as what they can predict. The model presented is quite complicated and in this manuscript it only predicts the odds ratio for any blood cancer (Fig. 5 which is likely simply the result of CN-LOHs being observed at lower cell fractions). The identified sex differences for selected mCAs are something that was already observed in Loh et al. 2020 and it can be inferred without estimating mutation rate and fitness effect"

response 5

There have been a number of major publications over the last year providing direct evidence that variants in clonal haematopoiesis and blood malignancies arise at a constant rate and subsequently expand exponentially at rates depending on the identity of the variant and its environment (Fabre et al. *Nature* 2022, Robertson et al. *Nature Medicine* 2022, Williams et al. *Nature* 2022). The model that we developed in (Watson et al. *Science* 2020) and adapted here for mCAs is the simplest possible framework for understanding how these features combine to shape the distribution of mCAs across people as they age. We do appreciate that the analysis can be technical in places, however it is important to realise that different quantitative features of the cell fraction distribution provide insights in to the different processes, e.g. the role of fitness, mutation rate and genetic drift. While we acknowledge that some of the mathematical details will only be fully appreciated by a subset of the readership (hence why we have put these in a detailed supplementary information) we have tried to make the key concepts and ideas as accessible as possible.

Major comments:

"What is the correlation between the estimated mutation rate (μ) and the prevalence of each event? What is the correlation between the estimated fitness effect (s) and the fraction of events with cell fraction higher than 5%? I do wonder here if μ and s are very similar to the two other more descriptive statistics and whether it would be better to use the descriptive statistics. Maybe reporting a comparison would help better understand the importance of μ and s "

response 6

The key point here is that the prevalence of a particular variant in blood (and how that prevalence changes with age) is shaped by *both* the variant's mutation rate and its fitness effect. Furthermore, prevalence is strongly influenced by the sensitivity of the assay and false negative rates at low cell fractions. Our framework is not sensitive to these issues (see response 2 to reviewer 1 and additional responses below). Reporting prevalence instead of μ is problematic because (i) it is not easily comparable across studies which have different sensitivities and across different variants with varying detectability (ii) it does not elucidate the cause of that prevalence (i.e. higher mutation rate or higher fitness?). Reporting a metric of "fraction of events with a cell fraction above 5%" rather than s is also problematic as this is confounded by mutation rate and requires arbitrary thresholds (why 5%?). We have included below a plot (Figure 1) showing how the mutation rate and fitness effect correlate with the metrics the reviewer suggested, but these metrics are far less meaningful than the ones we have used.

“The authors never discuss how the sampling bias of mCAs can affect their model. CN-LOHs for example are larger and at low cell fractions they cause deviations in the BAF twice as large as deletions and duplications, as they affect two haplotypes since they can be described as a duplication and a deletion at the same time”

response 7

We thank the reviewer for this comment. Reviewer 1 raised a similar point and we refer the reviewer to our Response 2, where we have included a new section we have added to the supplement (Supplementary Material 5: Effect of mCA detection limits on fitness and mutation rate estimates). In this new Supplementary Material, we discuss how sampling biases in mCAs at both low cell fraction and high cell fraction might influence our fitness and mutation rate estimates. The key point is that whilst different mCAs will have different biases at low cell fractions, our framework is largely insensitive to the low-cell fraction events because it is not simply based on, for example, prevalence.

“At the same time heterozygosity ‘dropping out’, which is shortly explained in Supplementary Material 1, can also bias sampling of events at large cell fractions. For example, an event at 50% cell fraction would cause a BAF deviation, of 10%, 16.7%, 25% for, respectively, a duplication, a deletion, and a CN-LOH. Could this affect the fitness effect? Are the authors sure that their model is robust against these biases? I did not feel sure at all after reading the manuscript and I was particularly concerned that many of the outcomes could be driven by technicalities rather than biological phenomena”

response 8

This is an excellent point. Whilst mCAs dropping out at high cell fraction due to technical effects was discussed briefly, it was not sufficiently clear in our original submission. In the data we use, CN-LOH events cannot be called above 54% cell fraction while losses cannot be called above 67% cell fraction. To assess how these high-cell fraction cut-offs affect our estimates of fitness effect we simulated data with a range of different fitness effects (7%-35%) for a population of the same size, and with the same age distribution, as the UK Biobank. We used our framework to infer the fitness effect of the mCAs with and without high cell fraction drop-out. The reviewer is correct in that the high cell fraction cut-offs do limit our ability to infer the fitness for highly fit losses and CN-LOH events, however we can provide a lower bound on their fitness (>21% for fit CN-LOH events, >28% for fit losses). The full effect of the high cell fraction cut-offs on our fitness and mutation rate inferences can now be seen in Supplementary Fig. 15 and 16 and are discussed in the new Supplementary Material 3 (see Response 2 for this new Supplementary Material section).

In response to the reviewers comments we realised that, independent of the high cell fraction cut-off, we were able to improve the robustness of our fitness inferences by fitting the cell fraction distribution on a

logit scale rather than on a simple log scale. This provides an improvement in correctly inferring the fitness effects of highly fit variants, which we extensively checked with simulated data. The intuition for this is that very highly fit variants 'bunch up' close to 100% cell fraction and this feature is more clearly delineated on a logit scale rather than on a simple log scale. We have therefore applied this improved approach on all our fitness and mutation rate estimates. We show below, in Figure 2, how the original and new fitness differences (and mutation rates) compare to each other. The only difference of note is that some of the highly fit variants have modified fitness effects for the reasons described above.

Figure 2: a. Comparison of fitness effects inferred using our original method (fitting cell fraction distributions on log scale) vs. our new method (fitting cell fraction distributions on a logit scale). b. Comparison of mutation rates inferred using our original method (fitting cell fraction distributions on log scale) vs. our new method (fitting cell fraction distributions on a logit scale).

“A scatterplot showing the two-parameter estimates for duplication, deletions, and CN-LOHs would be nice (maybe instead of panel Fig. 2b) to better assess how differently these three categories were fitted”

response 9

Scatter plots showing the joint distribution of these fitted parameters are shown in the figure below (Figure 3). However we believe that plotting the distribution of fitness effects as we do in Fig. 2b is more informative, as it is easier to visualise which fitness effects contribute the highest mutation rates.

Figure 3: Fitness effects and mutation rates for a. gains, b. losses, and c. CN-LOH events.

Minor comments:

“Introduction: Our estimates reveal that highly fit mCAs (growth rates ≥10% per year) occur at a rate of ~1 per 10 million cells per year, approximately 10-fold lower than equivalently fit SNVs. While occurring at a relatively low rate, the fitness consequences of these mutations can be dramatic, expanding at rates of up to 15-20% per year. <- This should be in the discussion rather than in the introduction”

response 10

We will leave this to editorial discretion, but we feel that stating important take-homes up front can sometimes be a good way of getting the key points across.

“Mutation rates and fitness effects of mCAs: the fitness effect of losses are systematically higher, with most fitness effects being between 14-20% per year. However, as a class, losses occur at a combined rate of $\sim 4 \times 10^{-8}$ per cell per year, 2.3-fold lower than CN-LOH. Gains appear to have a broad range of fitness effects, but occur at the lowest combined mutation rate of $\sim 2 \times 10^{-8}$ per cell per year <- how do we really know that this is not in large part due to the differential rate of detectability of different types of events? Certainly CN-LOHs are measured with a higher prevalence just due to the fact that they are easier to detect”

response 11

Please see Response 2 and Response 8.

“Sex differences in fitness effects and mutation rates: we infer that the observed higher prevalence of 10q- in women is due to a 4-fold higher mutation rate in women, with limited evidence for any sex bias in fitness effect <- as there is no association with age with this is likely simply that the prevalence of the event is ~ 4 times higher in females than males and Extended Data Fig. 1b in Loh et al. 2020 makes that already abundantly clear. Furthermore, this event is observed in relation to FRA10B expansions and so it seems weird to try to fit it into a model that does not take into account the prevalence of long FRA10B germline haplotypes”

response 12

The key point here is that prevalence is shaped by both mutation rate (μ), fitness effect (s) and sensitivity of detection. Our method enables us to disentangle these effects. It is plausible that the increased prevalence of 10q- in women could have been an effect of the mCA being fitter in women, however the data suggest it is a mutation rate effect.

The point about the association with FRA10B is well taken and, as highlighted in the paper, the cases where our model fails to capture the age dependence behaviour of mCAs correctly may highlight mCAs with interesting germline predispositions and FRA10B is an excellent example of this.

“Age dependence of mCAs: Overall, the observed prevalence of gain and loss events in both men and women is in close agreement with the predicted prevalence <- this seems quite subjective ... is there really close agreement?”

response 13

In response to this reviewer's comment we have now calculated the statistical significance of the difference between the observed and predicted age dependence for individual mCAs that had sufficient data (observed ≥ 30 times in men and ≥ 30 times in women). Our updated method for quantifying the difference between the observed and predicted age dependence is described in Supplementary Material 5 (shown below) and the results are shown in updated Fig. 3d (shown below). We now highlight the direction of the difference (stronger or weaker age dependence than predicted) and which mCAs showed statistically significant deviation from the expected age dependence. We thank the reviewer for this comment.

change to supplement: Supplementary Material 5: Age dependence of mCAs

Quantifying the deviation from the expected age dependence

To quantify any deviation from the expected age dependence, we calculated the relative difference between the gradients of the observed and expected age dependence (Supplementary Fig. 30). A maximum likelihood approach was used to calculate the gradients (m_{obs} or m_{exp}), minimising the L2 norm between the observed or expected prevalence (P) and $y = mt + C$ across 3 age (t) groups (age 40-50, 50-60, 60-70), optimising m and C . To account for error in prevalence, the square distance was multiplied by $T/2P$, where T is the total number of people in the age group. An age group was excluded from both observed and expected gradient estimation if the observed prevalence in that age group was 0 (because the prevalence would not be expected to be increasing linearly at that age). To determine whether any deviation from expected age dependence was statistically significant, a distribution of difference probability curve was calculated (for observed vs. expected age dependence gradients) and the p -value of the difference was calculated as the area under the curve where the difference ≤ 0 . If the p -value was < 0.05 , the deviation from expected age dependence was deemed statistically significant. The relative difference, $(m_{\text{obs}} - m_{\text{exp}}) / m_{\text{exp}}$, between the observed and expected age dependence gradients for individual mCAs is shown in Fig. 3d and the p -values in Supplementary Table 9.

change to supplement: Supplementary Material 5: Age dependence of mCAs: cont...

Supplementary Figure 30: Quantifying deviation from expected age dependence. a. A maximum likelihood approach was used to calculate the gradients (m_{obs} or m_{exp}) of the observed and expected age dependence. Deviation from expected age dependence was calculated as the relative difference between these gradients. b. Probability distributions of the gradients for the observed and expected age dependence. c. The p -value for the difference between the gradients was calculated from the distribution of gradient differences.

Supplementary Table 9: Quantification of deviation from expected age dependence Linear gradients for the observed and expected age dependence were calculated for mCAs observed ≥ 30 times in men and ≥ 30 times in women. p -values were calculated from the area under the distribution of gradient difference probability curve where the difference ≤ 0 .

mCA	sex	Expected gradient ($\times 10^{-6}$)	Observed gradient ($\times 10^{-6}$)	Gradient difference ($\times 10^{-6}$)	95% C.I. difference ($\times 10^{-6}$)	Relative difference	p -value
8+	female	6.72	2.57	4.15	-3.33 - 10.60	-0.62	1.35×10^{-1}
	male	4.56	5.77	1.21	-4.98 - 6.88	0.27	3.56×10^{-1}
12+	female	23.80	34.30	10.50	-0.55 - 20.20	0.44	2.74×10^{-2}
	male	32.80	36.90	4.09	-10.60 - 17.70	0.12	2.91×10^{-1}
14q+	female	12.10	13.50	1.41	-5.95 - 7.83	0.12	3.43×10^{-1}
	male	19.70	24.40	4.75	-5.12 - 13.80	0.24	1.78×10^{-1}
15q+	female	7.18	12.10	4.96	-9.60 - 18.70	0.69	2.48×10^{-1}
	male	57.50	119.40	61.90	27.10 - 92.20	1.08	2.93×10^{-4}
21q+	female	16.50	-0.28	16.80	4.24 - 28.50	-1.02	5.31×10^{-3}
	male	14.50	14.80	0.34	-8.18 - 8.18	0.02	4.73×10^{-1}
22q+	female	19.30	17.00	2.29	-6.53 - 10.60	-0.12	3.05×10^{-1}
	male	18.00	8.20	9.80	0.40 - 19.00	-0.54	2.14×10^{-2}

mCA	sex	Expected gradient ($\times 10^{-6}$)	Observed gradient ($\times 10^{-6}$)	Gradient difference ($\times 10^{-6}$)	95% C.I. difference ($\times 10^{-6}$)	Relative difference	p -value
2p-	female	20.60	7.68	12.90	4.27 - 21.40	-0.63	2.23×10^{-3}
	male	14.80	10.60	4.23	-3.33 - 11.20	-0.28	1.39×10^{-1}
4q-	female	5.19	8.10	2.91	-4.60 - 10.30	0.56	2.26×10^{-1}
	male	15.20	8.10	7.12	-2.27 - 15.00	-0.47	6.28×10^{-2}
5q-	female	20.70	11.10	9.61	0.51 - 18.70	-0.46	2.35×10^{-2}
	male	6.07	5.92	0.15	-7.42 - 6.72	-0.02	4.91×10^{-1}
7q-	female	7.52	7.61	0.09	-5.45 - 5.05	0.01	4.89×10^{-1}
	male	7.88	10.70	2.85	-3.94 - 9.39	0.36	2.15×10^{-1}
10q-	female	36.40	-7.78	44.20	30.70 - 56.40	-1.21	$< 1 \times 10^{-4}$
	male	11.20	3.84	7.38	-1.21 - 15.80	-0.66	5.29×10^{-2}
11q-	female	17.40	14.90	2.46	-15.20 - 18.80	-0.14	3.96×10^{-1}
	male	29.20	15.70	13.40	0.66 - 24.30	-0.46	2.33×10^{-2}
13q-	female	24.30	32.30	8.07	-4.17 - 19.70	0.33	1.11×10^{-1}
	male	45.80	54.90	9.16	-9.85 - 26.50	0.20	1.64×10^{-1}
14q-	female	0.72	6.21	5.49	-0.27 - 10.60	7.68	3.02×10^{-2}
	male	13.10	9.83	3.27	-4.70 - 10.30	-0.25	2.17×10^{-1}
20q-	female	32.80	27.60	5.18	-5.91 - 15.00	-0.16	1.74×10^{-1}
	male	65.90	64.10	1.76	-16.30 - 17.70	-0.03	4.28×10^{-1}

change to supplement: Supplementary Material 5: Age dependence of mCAs: *cont...*

mCA	sex	Expected gradient ($\times 10^{-6}$)	Observed gradient ($\times 10^{-6}$)	Gradient difference ($\times 10^{-6}$)	95% C.I. difference ($\times 10^{-6}$)	Relative difference	p-value
1p=	female	51.10	21.00	30.10	10.00 - 48.20	-0.59	1.71×10^{-3}
	male	58.20	45.40	12.80	-9.44 - 33.50	-0.22	1.22×10^{-1}
1q=	female	38.90	12.70	26.20	10.00 - 40.90	-0.67	6.97×10^{-4}
	male	44.80	23.20	21.60	2.58 - 38.60	-0.48	1.28×10^{-2}
2p=	female	11.30	6.22	5.05	-3.18 - 12.40	-0.45	1.09×10^{-1}
	male	9.01	7.24	1.78	-7.21 - 9.96	-0.20	3.51×10^{-1}
2q=	female	16.40	1.90	14.50	5.00 - 23.20	-0.88	1.79×10^{-3}
	male	13.80	0.71	13.10	2.53 - 22.70	-0.95	7.63×10^{-3}
3p=	female	10.80	0.78	9.98	1.87 - 16.80	-0.93	6.34×10^{-3}
	male	12.50	10.90	1.58	-7.55 - 10.20	-0.13	3.71×10^{-1}
3q=	female	8.05	3.95	4.10	-2.73 - 10.00	-0.51	1.14×10^{-1}
	male	11.10	8.53	2.54	-4.55 - 9.39	-0.23	2.43×10^{-1}
4q=	female	12.50	14.70	2.16	-7.67 - 11.00	0.17	3.33×10^{-1}
	male	7.43	41.00	33.60	9.55 - 54.10	4.52	2.13×10^{-3}
5q=	female	10.40	3.31	7.05	-0.37 - 13.80	-0.68	3.07×10^{-2}
	male	9.03	4.00	5.03	-4.33 - 13.70	-0.56	1.48×10^{-1}
6p=	female	21.10	8.36	12.70	1.52 - 22.70	-0.60	1.27×10^{-2}
	male	20.90	8.19	12.70	0.51 - 23.70	-0.61	1.74×10^{-2}
7q=	female	10.40	3.47	6.98	-0.32 - 13.30	-0.67	2.67×10^{-2}
	male	7.94	1.96	5.98	-1.98 - 13.30	-0.75	6.40×10^{-2}
8q=	female	7.94	-0.82	8.76	1.54 - 15.10	-1.10	6.50×10^{-3}
	male	12.40	10.70	1.77	-6.57 - 9.72	-0.14	3.44×10^{-1}
9p=	female	23.30	13.50	9.82	-2.63 - 21.50	-0.42	6.07×10^{-2}
	male	26.90	36.40	9.52	-6.11 - 23.90	0.35	1.12×10^{-1}
9q=	female	25.20	16.60	8.56	-5.56 - 20.70	-0.34	1.10×10^{-1}
	male	26.40	26.40	0.01	-14.20 - 13.10	0.00	5.02×10^{-1}
11p=	female	38.20	17.50	20.70	4.88 - 35.50	-0.54	5.80×10^{-3}
	male	41.00	14.40	26.60	8.18 - 44.50	-0.65	3.50×10^{-3}
11q=	female	26.50	20.80	5.79	-7.48 - 17.80	-0.22	1.89×10^{-1}
	male	38.00	41.00	2.92	-13.40 - 18.00	0.08	3.64×10^{-1}
12q=	female	16.50	3.69	12.80	2.37 - 22.30	-0.78	7.08×10^{-3}
	male	15.50	13.00	2.51	-8.27 - 13.00	-0.16	3.23×10^{-1}
13q=	female	28.90	32.60	3.77	-11.10 - 17.50	0.13	3.11×10^{-1}
	male	42.90	15.40	27.50	8.64 - 45.10	-0.64	2.15×10^{-3}
14q=	female	51.30	28.50	22.80	2.58 - 40.40	-0.44	1.10×10^{-2}
	male	57.20	49.50	7.68	-13.60 - 28.20	-0.13	2.37×10^{-1}
15q=	female	44.20	21.10	23.10	7.78 - 37.50	-0.52	1.47×10^{-3}
	male	33.90	28.70	5.21	-11.50 - 21.20	-0.15	2.71×10^{-1}
16p=	female	20.90	8.84	12.00	0.51 - 22.70	-0.58	2.29×10^{-2}
	male	25.10	10.80	14.30	2.63 - 25.70	-0.57	1.02×10^{-2}
16q=	female	14.10	7.93	6.16	-1.92 - 14.20	-0.44	7.57×10^{-2}
	male	17.40	6.86	10.50	0.40 - 20.60	-0.61	2.38×10^{-2}
17p=	female	7.88	0.48	7.40	0.36 - 14.20	-0.94	2.10×10^{-2}
	male	7.88	6.97	0.90	-6.04 - 7.09	-0.11	3.96×10^{-1}
17q=	female	27.10	9.58	17.50	4.24 - 29.70	-0.65	6.04×10^{-3}
	male	30.60	9.75	20.80	6.36 - 34.60	-0.68	2.84×10^{-3}
19p=	female	15.70	4.28	11.50	2.27 - 19.50	-0.73	7.14×10^{-3}
	male	14.30	5.82	8.44	-1.15 - 17.30	-0.59	4.72×10^{-2}
19q=	female	16.40	10.60	5.83	-3.18 - 14.50	-0.36	1.07×10^{-1}
	male	17.30	9.95	7.39	-3.82 - 17.40	-0.43	9.53×10^{-2}
20q=	female	14.20	12.50	1.65	-7.00 - 9.67	-0.12	3.53×10^{-1}
	male	13.10	-2.68	15.80	5.44 - 25.20	-1.20	1.55×10^{-3}
21q=	female	13.00	-1.14	14.10	4.89 - 21.80	-1.09	8.55×10^{-4}
	male	15.90	8.74	7.18	-1.92 - 15.70	-0.45	6.47×10^{-2}
22q=	female	24.50	20.60	3.92	-8.64 - 15.90	-0.16	2.78×10^{-1}
	male	24.80	26.90	2.00	-11.60 - 14.60	0.08	3.84×10^{-1}

Fig. 3: Age dependence of mCAs. **a-c**, Observed and expected prevalence of gains (a), losses (b) and CN-LOH (c) events for men and women. Expected prevalence (solid lines) calculated by summing the expected prevalence of each mCA in the mCA class. **d**, Deviation from expected age-dependence for each mCA observed ≥ 30 times in men and ≥ 30 times in women, with examples from each mCA class (see Supplementary Material 6 for age dependence plots for all mCAs). Deviation from expected age-dependence calculated as the relative difference between the observed and expected age dependence gradients. Positive deviation reflects mCAs which show more increase in prevalence with age than expected. Negative deviation reflects mCAs which show less increase in prevalence with age than expected. mCAs whose deviation was statistically significant ($p < 0.05$) are highlighted with an asterisk (see Supplementary Material 6).

“Fig. 4 - this figure again seems to reiterate the CN-LOH detection bias from Loh et al. 2020 and, since it cannot be independently tested, it seems inappropriate to show. Also ascertained rates of detected mCAs in Japan (see Figure 2b from Terao et al. Nature 2020) Finland (see Supplementary Figure S3 from Koskela et al. medRxiv 2021), which has a wider range of age in participants, make it clear that the true prevalence is severely underestimated by Fig. 4”

response 14

This is a good point and we agree with the reviewer here. We have now cut the figure from the main text. It is interesting that the prevalence in the papers highlighted by the reviewer (which span a wider range of ages) show a higher prevalence than we would predict based on extrapolating the UK Biobank data. One possible explanation for this is that we are underpowered to detect very low fitness mCAs (see points added to the discussion under Response 7). At older ages these low fitness mCAs would become increasingly detectable and thus would be expected to drive a stronger age dependence than predicted by our model.

“mCA fitness effects and cancer risk: we do indeed find a significant correlation between the fitness effect and odds ratio of subsequent blood cancer (Pearson $R = 0.6$, $p = 0.03$) <- I suppose this is not significant within each type (gain/loss/CN-LOH). Again, I find hard to believe this is not simply an artifact of the fact that CN-LOH are detected at lower cell fractions”

response 15

We thank the reviewer for this question. To test this we took each class of mCA and asked whether the correlation remains significant within each class, using all mCAs with a non-zero odds ratio of any blood cancer (see below, Figure 4). While the correlation appears to remain for all 3 classes, it only achieves statistical significance in CN-LOH. We agree with the more general point being made by the reviewer that the association of fitness with risk has some limitations and we believe these limitations are addressed in the discussion.

Figure 4: mCA fitness effects and blood cancer risk for a. gains, b. losses, and c. CN-LOH events.. The relationship between inferred fitness effect and odds ratio (OR) of any blood cancer is shown for autosomal mCAs with a non-zero OR¹⁴ and which were observed in ≥ 30 individuals. The mCAs highlighted in bold are those that Loh et al determined to have a statistically significant increased risk of any blood cancer¹⁴. Pearson correlation coefficient and 95% confidence intervals (grey shaded area) are shown. The blood cancers were diagnosed >1 year after DNA collection (within 4-9 years follow-up) in individuals with no previous cancer and with no restriction on blood counts at assessment.

“Fit mCAs occur at a lower rate relative to fit SNVs: Strikingly, the total mutation rate to highly fit mCAs ($s > 10\%$ per year) is over 10-fold lower than the total mutation rate to highly fit SNVs <- isn't this just restating the fact that SNV mutations (at least those in key DNMT3A, TET2, ASXL1, PPM1D, JAK2 genes) are much more prevalent than mCAs?”

response 16

As discussed above, the prevalence is determined by the combined effects of mutation rate and fitness effect and it could have been the case that SNVs were more prevalent because of systemically higher fitness effects. However, the fitness effects of SNVs and mCAs appear to be similar and so we determine that the difference in prevalence is driven predominantly by differences in mutation rate. It is also important to stress that prevalence is in large part controlled by limits of sensitivities and so comparing prevalence of SNVs and mCAs would be problematic because of their different detectabilities.

“Cell-extrinsic influences and age-specific effects: 5q- shows a higher mutation rate in women, but a higher fitness effect in men, suggesting extrinsic effects play a role <- could this just be statistical noise? How do I know that this observation is robust to random sampling? As presented it is not very believable. There should be p-values associated with statements like this that survive multiple hypotheses correction”

response 17

We did indeed calculate these p -values, which can be found in Supplementary Tables 6-8 in ‘Supplementary Material 4: Sex differences in mCA fitness effects and mutation rates’.

“Fig. S1 <- FRA10B seems localized in the wrong place”

response 18

We thank the reviewer for noticing this. We have now corrected Fig. S1 to show FRA10B in the correct location.

figure changed in Supplementary Material 1

Supplementary Figure 1: mCAs detected among ~500,000 UK Biobank participants in Loh et al 2020¹⁴: part 1. Each mCA is represented as a horizontal line. Gain events are shown in red, loss events in blue and CN-LOH events in yellow. Genes recurrently mutated in clonal haematopoiesis or haematological malignancies which may be putative target genes for loss, gain or CN-LOH events are labelled in blue, red and orange respectively.

Reviewer 3

“Watson and Blundell present an analysis entitled ‘Mutation rates and fitness consequences of mosaic chromosomal alterations in blood’. Based on existing autosomal mCA calls from the UK Biobank the authors adjust their existing population genetics framework to calculate mutation rates and fitness coefficients (growth rates) of clones harbouring somatic mCAs. This is a novel and powerful analysis revealing that mCAs exhibit growth rates similar to those seen for CG driver gene mutations, but mutate at around 10x lower rate. The authors also demonstrate that while many mCAs exhibit the expected age incidence, some do not, and thus warrant further analysis.

Overall I congratulate the authors for a timely and insightful analysis. The paper, figures and supplementary figures are clearly presented. The comprehensive analysis fills a knowledge gap in our current understanding of the dynamics of CH.”

response 19

We thank the reviewer for these kind comments.

“I only have a few comments and questions motivated by curiosity rather than fundamental concerns.

My main question would be how the authors deal with population heterogeneity introduced by driver gene mutation driven CH, germline predisposition or other factors. While this is partially discussed in the manuscript and the authors show the clustering of alterations in fewer than expected individuals, I’d be interested to see a further investigation.”

response 20

We agree with the reviewer that the broader-than-expected distribution of mCAs (and for that matter SNVs) across individuals is an intriguing observation which has also been observed in a number of other studies including in blood (e.g. Young et al. Nature Communications 2016 and Genovese et al. NEJM 2014) as well as in other tissues e.g. bladder epithelium (Lawson et al. Science 2020). As the reviewer suggests there are a number of plausible reasons for such effects including germline predisposition and extrinsic factors. However a thorough investigation into either of these factors is a substantial undertaking that would be outside the scope of our current work. Indeed we are not aware of a paper that has investigated the association between multiple driver clonal haematopoiesis vs. single driver clonal haematopoiesis in a GWAS for example.

“How many of the mCA calls overlap with known driver gene mutations? It's intriguing to see the high fitness effects associated with mCAs and I wondered whether any occur on the back of a driver gene mutated CH case (which reaches frequencies of around 10% in the UKB). Perhaps data from this study could be used: <http://dx.doi.org/10.1038/s41588-022-01121-z>.”

response 21

This is an excellent suggestion and we agree that a systemic analysis of the co-occurrence of somatic SNV driver mutations and mCAs would be very interesting. However, because the vast majority of mCAs and SNVs occur at low cell fraction, assessing co-occurrence from two independent bulk assays would generally not be possible. There are a handful of examples where co-occurrence can be assessed with certainty (e.g. a JAK2 V617F at >50% VAF, where a high cell fraction 9p= (CN-LOH) is also present). Indeed, JAK2 V617F and 9p= is one example where you can see many instances of a double-hit event, where the first driver mutation is an SNV (JAK2 V617F), with a subsequent CN-LOH event. These were reported in the original Loh 2018 paper.

“Another analysis could include an investigation of individuals with >1 alterations (about 10% judging from Figure S4) and whether this leads to higher growth rates.”

response 22

We had exactly the same thought during this work. Indeed, we spent a considerable amount of time thinking about this idea. We did indeed do the analysis where we assessed whether the fitness effect of an mCA that occurs on its own in an individual is significantly different to the fitness effect of the mCA when it co-occurs with at least one other mCA in that person. Our reasoning (perhaps the same as the reviewer?) was that some of these events will be multiple mutants, i.e. cells carrying both mCAs which may drive faster growth. We have now included our fitness effect and mutation rate inferences from this analysis in Supplementary Tables 1-3 and have also included a plot in the supplement (Supplementary Fig. 14, shown below) which shows the fitness effect of each mCA when it occurs in isolation vs. when it co-occurs with any other mCA. As the reviewer can see, the results of this analysis are inconclusive. We suspect the reason for this is that there will be many cases where the multiple mCAs are not co-occurring in the same clone and thus one may expect the reverse effect - a slowing of the growth rate due to clonal competition.

tables amended in Supplementary Material 2

Supplementary Table 1: Fitness effects and mutation rates for gain events. Fitness effects and mutation rates were calculated for mCAs that were observed in at least 8 individuals. Fitness effects and mutation rates were calculated using data from individuals who had a single mCA ('Single mCA' column) and also from individuals who also had additional mCA(s) ('mCA + additional mCA(s)' column).

mCA	Observed number			Single mCA				mCA + additional mCA(s)			
	Single mCA	+ 1 other	+ ≥ 2 other	s	s 95% C.I.	μ	μ 95% C.I.	s	s 95% C.I.	μ	μ 95% C.I.
1p+	23	2	1	14.05	13.03 - 17.11	0.41	0.12 - 0.65	-	-	-	-
1q+	15	7	14	14.18	12.59 - 29.31	0.15	0.02 - 0.27	15.78	14.18 - 45.22	0.22	0.02 - 0.35
2p+	8	2	6	14.90	13.27 - 48.37	0.14	0.01 - 0.37	-	-	-	-
3+	30	39	32	14.86	13.14 - 38.00	0.21	0.03 - 0.30	13.88	13.27 - 14.90	0.60	0.41 - 0.80
3p+	8	0	4	14.08	12.45 - 47.55	0.25	0.01 - 0.95	-	-	-	-
3q+	17	17	26	14.52	13.01 - 20.18	0.13	0.04 - 0.25	14.71	13.57 - 17.57	0.46	0.20 - 0.74
5+	21	0	6	9.27	8.24 - 11.31	1.49	0.31 - 3.72	-	-	-	-
5p+	32	5	4	10.34	9.60 - 11.68	1.81	0.63 - 3.66	-	-	-	-
5q+	9	5	5	15.43	13.86 - 47.64	0.09	0.01 - 0.18	9.59	8.67 - 45.41	1.93	0.01 - 5.31
6p+	13	4	5	14.18	12.59 - 46.82	0.36	0.01 - 0.74	-	-	-	-
6q+	8	0	0	12.81	11.98 - 47.52	0.80	0.01 - 2.00	-	-	-	-
8+	75	15	30	16.57	14.94 - 19.51	0.42	0.22 - 0.59	15.18	14.12 - 18.37	0.32	0.16 - 0.48
9+	46	14	10	15.61	14.08 - 21.22	0.29	0.10 - 0.43	18.16	15.71 - 48.37	0.10	0.03 - 0.18
9p+	8	5	2	13.58	11.89 - 47.46	0.10	0.01 - 0.25	-	-	-	-
9q+	18	5	4	14.10	12.55 - 19.53	0.15	0.04 - 0.25	-	-	-	-
12+	276	112	100	13.47	12.98 - 13.88	2.77	2.24 - 3.28	15.89	15.09 - 16.82	1.44	1.08 - 1.76
12q+	16	7	7	14.90	13.27 - 33.67	0.14	0.02 - 0.28	12.45	11.63 - 34.49	0.33	0.02 - 0.63
14q+	147	8	7	14.05	13.56 - 14.79	1.56	1.12 - 1.89	12.29	11.43 - 25.14	0.28	0.02 - 0.52
15q+	206	15	2	12.62	12.27 - 12.98	2.73	2.19 - 3.22	14.08	13.27 - 21.43	0.14	0.03 - 0.21
17q+	9	5	5	14.90	13.27 - 47.55	0.16	0.01 - 0.31	14.08	13.27 - 45.92	0.14	0.01 - 0.24
18+	47	44	80	13.86	13.04 - 15.15	0.37	0.23 - 0.52	13.19	12.63 - 13.87	1.54	1.06 - 1.95
18q+	10	7	10	15.88	14.33 - 46.90	0.07	0.01 - 0.13	13.51	12.41 - 19.02	0.25	0.05 - 0.48
21q+	125	13	14	11.16	10.73 - 11.65	2.59	1.86 - 3.34	14.48	13.29 - 18.46	0.26	0.09 - 0.36
22q+	155	23	13	11.12	10.77 - 11.48	5.18	3.72 - 6.68	13.27	12.29 - 14.90	0.56	0.26 - 0.88

figure added in Supplementary Material 2

Supplementary Figure 14: Fitness effects and mutation rates for mCAs observed as single events vs observed with additional mCAs. Only mCAs that were observed in 8 or more individuals that also had additional mCAs are shown.

“Furthermore, it could be interesting to assess whether growth is altered in a population subset carrying specific alleles found to promote the prevalence of CH with mCA. Examples include *TCL1A*, *FRA1* and other variants, both reported in <http://dx.doi.org/10.1038/ng.3821> and <http://dx.doi.org/10.1038/s41586-018-0321-x>.”

response 23

This is an excellent suggestion and is indeed work that we are currently undertaking. As the reviewer suggests, it will be interesting to assess whether variants predispose to clonal haematopoiesis, e.g. *CHEK*, *TERT* (and those suggested by the reviewer) are associating with clonal haematopoiesis due to increasing the mutation rate or increasing the fitness effect or a combination of the two.

“On a different note, the analysis only considers only autosomal mCAs. It could be insightful to complement the analysis with an investigation of alterations of the allosomes. Data should also be available, as this has been investigated, eg. in <http://dx.doi.org/10.1038/ng.3821>. This could be especially interesting as mLOY is very prevalent in ageing men.”

response 24

We thank the reviewer for this suggestion - we agree that including estimates of fitness effects and mutation rates for allosomes is an interesting comparison. We have now performed an analysis of the mosaic loss of X (mLOX) and mosaic loss of Y (mLOY) calls generated by Loh et al from ~500,000 UK Biobank participants (available from UK Biobank as part of Return 3094). We now discuss these results in the main text (see below), have included the analysis in a new section in the supplement (Supplementary Material 3) and have updated the abstract (see below).

change to main text: Results

Mosaic loss of Y (mLOY) and X (mLOX)

Mosaic loss of the Y chromosome (mLOY) and mosaic loss of the X chromosome (mLOX) are commonly observed in blood, at increasing prevalence with age^{13,21}. We reasoned that our framework could determine whether this high prevalence was due to increased fitness effects or anomalously high mutation rates. To check this we considered the cell fraction distribution of all 22,367 mLOY events across the 220,893 men in UK Biobank (Fig. 5a). The distribution of mLOY cell fractions is in remarkably close agreement with the predictions of our framework, conferring a modest fitness effect (s) of ~13% per year, and a mutation rate (μ) of $\sim 9 \times 10^{-7}$ per year, which is ~1000-fold higher than the typical autosomal mCA mutation rate (Fig. 5b). The close agreement between the predicted and observed age dependence for mLOY provides further support for our fitness effect and mutation rate inferences (Supplementary Fig. 36b).

change to main text: Results *cont...*

We next considered the 8577 mLOX events across 261,889 women. In contrast to mLOY, the observed distribution of mLOX cell fractions is clearly inconsistent with the predictions of our framework based on a single fitness effect and single mutation rate (Supplementary Fig. 37a). This inconsistency is further evidenced by the marked discrepancy between the observed and expected age dependence (Supplementary Fig. 37b). The way in which the observed distribution of mLOX cell fractions deviates from our null model, with two distinct plateaus, is highly suggestive of two distinct mLOX events occurring at different rates and conferring different fitness effects. To check this intuition, we simulated mLOX events with two possible combinations of mutation rate and fitness effect and observed cell fraction distributions very similar to those observed in the UK Biobank data (Supplementary Material 6). We therefore modified our framework to predict the distribution of cell fractions that would be observed for two distinct fitness effects occurring at two distinct mutation rates for mLOX. This modified framework provides a much closer agreement to the observed cell fraction estimates in UK Biobank (Fig. 5c). It predicts that there are two possible mLOX events: one of these confers a large fitness effect ($s \approx 18\%$ per year), but occurs at a low rate ($\mu \approx 6 \times 10^{-9}$ per year), while the other confers a low fitness effect ($s \approx 7\%$ per year), but occurs at a rate similar to mLOY ($\mu \approx 1.4 \times 10^{-6}$ per year) (Fig. 5b). Furthermore, this modified framework provides a much closer agreement between the predicted and observed age dependence for mLOX.

Fig. 5: mLOY and mLOX. **a**, The distribution of mLOY cell fractions (datapoints) is in close agreement with the predictions of our framework (solid lines). **b**, Inferred mutation rates for autosomal mCAs (violin plot), mLOX₁, mLOY and mLOX₂. The middle horizontal line in the violin plot shows the mean mutation rate and the top and bottom horizontal lines indicate the minimum and maximum autosomal mCA mutation rates. Error bars on the mLOX and mLOY datapoints represent 95% confidence intervals. **c**, The distribution of mLOX cell fractions is suggestive of a framework involving two distinct mLOX events (blue solid line): one event (mLOX₁) occurring at high mutation rate with low fitness (green dashed line) and another event (mLOX₂) occurring at low mutation rate and high fitness (purple dotted line). **d**, Possible explanation for two mLOX events: one involving loss of the active X (Xa) and one involving loss of the inactive X (Xi).

change to main text: Discussion

Mosaic loss of sex chromosomes

Because mLOY and mLOX are common events in blood, they provide good statistical power for testing the predictions of our framework. These events demonstrate why total number of events detected above technical sensitivity limits across all individuals can be a misleading statistic. Our framework reveals that while mLOY is more prevalent than mLOX, the overall mutation rate to mLOX is higher, but it is less prevalent because it confers a modest fitness effect. The behaviour of mLOX is further demonstration of why our framework is an important null model for understanding the fitness effects and mutation rates of mosaic events. mLOX provides the strongest evidence for deviating from this null model and is more consistent with a model in which mLOX can confer two distinct fitness effects which occur at two different rates. While we do not have direct evidence for the cause of this, we speculate that it could be related to the fact that the X-chromosome that is lost will either be 'active' (Xa) or 'inactive' (Xi). This could plausibly result in two distinct effects occurring at two different rates. Previous studies have shown that high cell fraction mLOX typically results from the loss of the 'inactive' X-chromosome (Xi)³², suggesting that this could be the event that drives the higher fitness effect (Fig. 5b). More generally, these examples demonstrate how quantitative features in the distribution of cell fractions, and their evolution with age, can reveal insights above the underlying biology.

addition to supplement: Supplementary Material 6

Mosaic loss of Y (mLOY) and mosaic loss of X (mLOX)

Cell fraction estimates for mosaic loss of chromosome Y (mLOY) and mosaic loss of chromosome X (mLOX) are available from UK Biobank, as part of the return of mCA calls (Return 3094) generated by Loh et al from 482,789 UK Biobank participants¹⁴. As with the autosomal mCA calls, mLOY and mLOX calls were generated by transforming the UK Biobank SNP array genotyping intensities on the Y and X chromosome into \log_2 R ratios (LRR) and B-allele frequencies (BAF) to obtain measures of total and relative allelic intensities respectively. There is therefore a sharp cut-off at mLOY and mLOX cell fractions $\geq 67\%$, corresponding to BAF deviations >0.25 , above which heterozygous SNPs 'drop out' out of the data.

mLOY

mLOY was detected in 22,367 men. To calculate mutation rate (μ) and fitness effect (s) for mLOY, a maximum likelihood approach was used, as described in Supplementary Material 2 (Supplementary Fig. 36a). Using our inferred parameters for s and μ , we calculated the expected prevalence of mLOY, as a function of age (using the approach described in Supplementary Material 5) and found our predictions to be in line with the observed age dependence of mLOY (Supplementary Fig. 36b). The behaviour of mLOY is therefore consistent with the underlying assumptions of our framework, in which mCAs are acquired stochastically at a constant rate throughout life and then expand with an mCA-specific intrinsic fitness effect.

Supplementary Figure 36: mLOY detected among ~220,000 male UK Biobank participants. **a.** Parameter estimation for mLOY. The cell fraction probability density histogram is shown for mLOY (datapoints) with the theory distribution (solid line) fitted using maximum likelihood approaches. Error bars represent sampling noise. Grey vertical dashed line shows the fitted ϕ parameter ($\frac{e^{st}-1}{Ns}$), where the exponential fall-off in densities occurs. The white cross on the maximum likelihood heatmap marks the most likely μ and s . The two small plots show the distribution of likelihoods for s and μ , with the blue vertical line representing the most likely value. 95% confidence intervals are shown shaded in blue. **b.** Predicted age prevalence for mLOY (solid line) using inferred s and μ , compared to observed age prevalence (datapoints). The cell fraction limit of detection was taken as the minimum cell fraction observed for mLOY, multiplied by 1.5.

mLOX

mLOX was detected in 8577 women. Unlike mLOY, the shape of the mLOX cell fraction distribution was clearly inconsistent with that predicted by our framework. Indeed, attempting to fit a cell fraction distribution predicted by our framework, to infer the mutation rate (μ) and fitness effect (s) for mLOX (using the approach described in Supplementary Material 2), revealed a poor fit to the data (Supplementary Fig. 37a). The age prevalence predicted by these inferred parameters was also significantly different to the observed age dependence of mLOX (Supplementary Fig. 37b).

Supplementary Figure 37: mLOX detected among ~260,000 female UK Biobank participants. **a.** Parameter estimation for mLOX. The cell fraction probability density histogram is shown for mLOX (datapoints) with the theory distribution (solid line) fitted using maximum likelihood approaches. Error bars represent sampling noise. Grey vertical dashed line shows the fitted ϕ parameter ($\frac{e^{st}-1}{Ns}$), where the exponential fall-off in densities occurs. The white cross on the maximum likelihood heatmap marks the most likely μ and s . The two small plots show the distribution of likelihoods for s and μ , with the blue vertical line representing the most likely value. 95% confidence intervals are shown shaded in blue. **b.** Predicted age prevalence for mLOX (solid line) using inferred s and μ , compared to observed age prevalence (datapoints). The cell fraction limit of detection was taken as the minimum cell fraction observed for mLOX, multiplied by 1.5.

addition to supplement: Supplementary Material 6 cont...

Considering two distinct mLOX events with different mutation rates and fitness effects: Simulated data

We considered whether the shape of the mLOX cell fraction distribution (Supplementary Fig. 37a) could be explained by 2 distinct mutational events underlying LOX: one which occurs at a high mutation rate (μ_1) but confers a small fitness effect (s_1) and another that occurs at a much lower rate (μ_2) but confers a substantially larger fitness effect (s_2). To test this, we generated simulated mLOX calls across ~500,000 people (as in Supplementary Material 2), but with 2 possible combinations of mutation rate and fitness effect (μ_1, s_1) and (μ_2, s_2) for the mLOX event. The distribution of the simulated mLOX cell fractions (Supplementary Fig. 37a) was indeed similar to the distribution observed for mLOX in UK Biobank (Supplementary Fig. 37a). Attempting to fit a single theory distribution to the simulated mLOX distribution, to infer the mutation rate (μ) and fitness effect (s), revealed a poor fit to the data and inaccurate inferences for s and μ .

When the product of the difference between the fitness effects of two mLOX events and the age, T , is large enough ($(s_2 - s_1)T \gg 1$), and when the two mutation rates are substantially different ($\mu_1 \gg \mu_2$), a qualitative feature emerges in the distribution characterised by a large peak at low cell fractions (set by the high mutation rate, low fitness event (which falls off at $\phi_1 \approx \exp(s_1 T)/N\tau s_1$), followed by a plateau out to higher cell fractions (set by $\phi_2 \approx \exp(s_2 T)/N\tau s_2$). This separation of regimes means that the low cell fraction events are dominated by one event and the high cell fraction events are dominated by the other. It is therefore possible to estimate the (μ_1, s_1) and (μ_2, s_2) parameters by fitting to the low cell fraction vs. the high cell fraction distributions independently. We therefore divided the histogram into two sets of datapoints: the lower cell fraction datapoints included mLOX events whose cell fraction was below the point at which the fall-off in cell fraction densities starts to plateau (lower split point = vertical dashed line at ~9% cell fraction in Supplementary Fig. 38b). The upper cell fraction datapoints included mLOX events whose cell fractions were above the cell fraction that was 3 histogram 'bins' above the lower split point (upper split point = vertical dashed line at ~14% cell fraction in Supplementary Fig. 38b), to avoid fitting independent distributions to datapoints where there is expected to be the most contribution from both distributions together). A theory distribution was then fit to each of the two datapoint sets, using a maximum likelihood approach, to infer the parameters μ_1, s_1 (for the distribution at lower cell fraction) and μ_2, s_2 (for the distribution at higher cell fraction). Using this approach, we were able to infer very close to the ground-truth values for these parameters (Supplementary Fig. 38b).

Supplementary Figure 38: Simulated mLOX calls with 2 different mutation rates and fitness effects. **a.** Simulated mLOX calls were generated at a high rate but conferring a low fitness (μ_1, s_1) or at a low rate but conferring a high fitness (μ_2, s_2). The cell fraction probability density histogram is shown for these mLOX calls (datapoints). Parameter estimation, using a maximum likelihood approach to fit a single theory distribution (solid line) with a single s and single μ , is shown. Error bars represent sampling noise. The white cross on the maximum likelihood heatmap marks the most likely inferred μ and s . **b.** Parameter estimation, using a maximum likelihood approach to fit a distribution (solid blue line) made up of 2 distinct mLOX events: one with a high mutation rate (μ_1), but low fitness (s_1) (solid grey line) and another with a low mutation rate (μ_2), but high fitness (s_2) (dashed grey line). Vertical black dashed lines show the cell fraction split points used for fitting the lower and upper cell fraction datapoints to the theory distribution. Error bars represent sampling noise. The white cross on the maximum likelihood heatmaps marks the most likely inferred s_1, μ_1, s_2 and μ_2 .

Considering two distinct mLOX events with different mutation rates and fitness effects: UK Biobank mLOX data

Dividing the UK Biobank mLOX cell fraction histogram into two sets of datapoints and fitting a theory distribution to each of the two datapoint sets, revealed a much better fit to the data (Supplementary Fig. 39a). This enabled us to infer the fitness effects and mutation rates of two distinct mLOX events, one which occurs with a high mutation rate ($\mu_1 = 1.4 \times 10^{-6}$ per year), but low fitness effect ($s_1 = 7.5\%$ per year); and another which occurs with a low mutation rate ($\mu_2 = 6.5 \times 10^{-9}$ per year), but high fitness effect ($s_2 = 17.7\%$ per year).

addition to supplement: Supplementary Material 6 cont...

Using the inferred parameters for s_1, μ_1, s_2 and μ_2 , we calculated the expected combined prevalence of the two distinct mLOX events, as a function of age. In contrast to our expected prevalence inferred from a single fitted theory distribution (Supplementary Fig. 37), our predictions based on two distinct mLOX events are in excellent agreement with the observed age dependence of mLOX in UK Biobank (Supplementary Fig. 39b). The behaviour of mLOX is therefore consistent with there being two possible distinct mLOX events, each of which behaves according to the underlying assumptions of our framework: each of the two mLOX events is acquired stochastically at a constant rate throughout life and then expands with a specific intrinsic fitness effect. We speculate that this likely has a biological origin in that loss of either the active X or inactive X likely confer different fitness effects.

Supplementary Figure 39: mLOX detected among ~260,000 female UK Biobank participants. a. Parameter estimation for mLOX, using a maximum likelihood approach to fit 2 separate theory distributions (solid line and dashed line), each with their own single s and μ . Error bars represent sampling noise. The white crosses on the maximum likelihood heatmaps marks the most likely s_1, μ_1, s_2 and μ_2 . Distribution of likelihoods for s_1, μ_1, s_2 and μ_2 are shown, with the blue vertical line representing the most likely value. 95% confidence intervals are shown shaded in blue. **b.** Predicted age prevalence for mLOY (solid line) using inferred s and μ , compared to observed age prevalence (datapoints).

Using the inferred parameters for s_1, μ_1, s_2 and μ_2 , we calculated the expected combined prevalence of the two distinct mLOX events, as a function of age. In contrast to our expected prevalence inferred from a single fitted theory distribution (Supplementary Fig. 16), our predictions based on two distinct mLOX events are in excellent agreement with the observed age dependence of mLOX in UK Biobank (Supplementary Fig. 18b). The behaviour of mLOX is therefore consistent with there being two possible distinct mLOX events, each of which behaves according to the underlying assumptions of our framework: each of the two mLOX events is acquired stochastically at a constant rate throughout life and then expands with a specific intrinsic fitness effect.

Supplementary Table 4: Fitness effects and mutation rates for mLOY and mLOX.

mCA	occurrences	Fitness effect (s) (% per year)		Mutation rate (μ) ($\times 10^{-9}$ / year)	
		s	s 95% C.I.	μ	μ 95% C.I.
mLOY	22367	13.17	13.13 - 13.21	861	840 - 881
mLOX ₁	8577	7.46	7.40 - 7.53	1349	1256 - 1423
mLOX ₂		17.67	16.20 - 43.39	6.47	3.20 - 9.43

change to main text: Abstract

Abstract

Mosaic chromosomal alterations (mCAs) are commonly detected in many cancers and have been found to arise decades before diagnosis. A quantitative understanding of the rate at which these events occur and their functional consequences could improve cancer risk prediction and our understanding of somatic evolution in healthy tissues. Here we use clone size estimates of mCAs from the blood of ~500,000 participants in the UK Biobank to estimate the mutation rates and fitness consequences of acquired gain, loss and copy-neutral loss of heterozygosity (CN-LOH) events at the chromosomal arm level. Most mCAs have moderate to high fitness effects (growth rates of 10–20% per year), but occur at a low rate, being over 10-fold less common than equivalently fit single nucleotide variants (SNVs). Striking counter-examples to this are mosaic loss of X (mLOX) and Y (mLOY), which we estimate to occur at ~1000-fold higher mutation rates than autosomal mCAs, but confer only modest fitness advantages. Whilst the majority of mCAs increase in prevalence with age in a way that is consistent with a constant growth rate, we find specific examples of mCAs whose behaviour deviates from this, suggesting fitness effects for these mCAs may depend on inherited variants, extrinsic factors or distributions of fitness effects. A notable example is mLOX, whose behaviour we find to be more consistent with two separate events with distinct fitness effects and mutation rates. We speculate that this may be related to loss of the inactive versus active X-chromosome.

Decision Letter, first revision:

22nd May 2023

Dear Dr. Blundell,

Thank you for submitting your revised manuscript "Mutation rates and fitness consequences of mosaic chromosomal alterations in blood" (NG-A60886R). It has now been seen by the original referees and their comments are below. The reviewers find that the paper has improved in revision, and therefore we'll be happy in principle to publish it in Nature Genetics, pending minor revisions to address Reviewer #2's queries and to comply with our editorial and formatting guidelines.

Sincerely,

Safia Danovi
Editor
Nature Genetics

Reviewer #1 (Remarks to the Author):

The authors did a fantastic job in addressing my comments and the comments of the other reviewers. The additional simulations have addressed my methodological concerns and the additional analyses have added to the biologic interest of the paper.

Reviewer #2 (Remarks to the Author):

The manuscript is quite improved after review and I find the additional mLOY and mLOX analyses quite compelling. I thank the reviewer for the "Correlation between number of mCA observations and mCA mutation rate" though I do think that this figure needs a placement in the manuscript, even if as a supplementary figure and with reported correlation estimates, to help the reader better understand the novelty and importance of the estimated mutation rate (μ) and the estimated fitness effect (s) compared to the more descriptive parameters of prevalence and fraction of events with cell fraction higher than 5%. The new Fig. 4 is compelling, but it is still not clear in my mind whether the fitness effect predicts which mCA types have higher risk of turning into blood cancer better than the simpler statistic of proportion of mCA > 5% cell fraction.

Major comments:

The new Fig. 5 is very interesting, especially since it is trained on two mCA events for which there is a large amount of data. I think the possible explanation provided by the authors for why there might be two different mLOX events is inappropriate though. Rather than the inactive and the active X being lost as two different events, it could easily be the case that the inactive X is lost in two different cell types instead. I am very skeptical that the active X can be lost at any appreciable frequency as we don't see any other large autosome being lost in its entirety. I suppose it is possible that the active X is lost and the inactive X is reactivated, but this to me feels all a bit too speculative for a main figure. Have the authors thought instead about the fact that, while two mLOY events in the same individual at low cell fractions would be interpreted as an mLOY event with cell fraction the sum of the two cell fractions, two mLOX events in the same individual at low cell fractions could be interpreted either as an mLOX event with cell fraction the sum of the two cell fractions or as an mLOX event with cell fraction the difference of the two cell fractions if the two events happened in two cells that inactivated a different X chromosome. This is due to the fact that mLOX at low cell fractions is entirely detected through BAF deviations and not through a shift in the LRR. Another possibility is that mLOX at low cell fractions is not mLOX at all and maybe it is related to an overall imbalance between cells that inactivated one X chromosome and cells that inactivated the other X chromosome, which is known to increase with age. In short, while I find the observed differential dynamic between mLOX and mLOY as one of the most fascinating aspects of the manuscript now, I would be careful about speculating what the cause is without supporting data

Minor comments:

ACKNOWLEDGMENTS: mLOY and mLOX calls are available by request via application to UK Biobank (Return 3094) (<https://www.ukbiobank.ac.uk>) <- the resource at <https://biobank.ndph.ox.ac.uk/ukb/dset.cgi?id=3094> includes both autosomal and sex chromosome mCAs, so it is not clear to me why only mLOY and mLOX are mentioned

Reviewer #3 (Remarks to the Author):

The authors have addressed all my comments and added a very interesting analysis of allosomic LOH.

Author Rebuttal, first revision:

Mutation rates and fitness consequences of mosaic chromosomal alterations in blood: Response to reviewers' comments on revised manuscript

We thank the reviewers for their constructive comments. We address Reviewer 2's remaining comments below.

Reviewer 2

"The manuscript is quite improved after review and I find the additional mLOY and mLOX analyses quite compelling. I thank the reviewer for the "Correlation between number of mCA observations and mCA mutation rate" though I do think that this figure needs a placement in the manuscript, even if as a supplementary figure and with reported correlation estimates, to help the reader better understand the novelty and importance of the estimated mutation rate (μ) and the estimated fitness effect (s) compared to the more descriptive parameters of prevalence and fraction of events with cell fraction higher than 5%. The new Fig. 4 is compelling, but it is still not clear in my mind whether the fitness effect predicts which mCA types have higher risk of turning into blood cancer better than the simpler statistic of proportion of mCA > 5% cell fraction."

response 1

We have now included an additional section in the supplement (Supplementary Note 8) on the 'Relationship between mCA mutation rate or fitness effect and more descriptive metrics', where we describe our rationale for using 'mutation rate' and 'fitness effect' metrics. We have also included the reviewer's requested correlation plots in this section (as shown below).

addition to supplement: Supplementary Note 8:

Relationship between mCA mutation rate or fitness effect and more descriptive metrics.

Descriptive metrics, such as a variant's prevalence, or the proportion observed at higher cell fractions are sometimes used as a proxy for a variant's mutation rate and fitness effect respectively. However, the prevalence of a particular variant (and how that prevalence changes with age) is shaped by *both* the variant's mutation rate and its fitness effect. Furthermore, prevalence is strongly influenced by the sensitivity of the assay and false negative rates at low cell fractions. A strength of our framework is that it is not sensitive to these issues (Supplementary Note 2). Reporting prevalence instead of μ is problematic because (i) it is not easily comparable across studies which have different sensitivities and across different variants with varying detectability (ii) it does not elucidate the cause of that prevalence (i.e. higher mutation rate vs. higher fitness). Reporting a metric of "fraction of events with a cell fraction above e.g. 5%" rather than s is also problematic as this is confounded by mutation rate and requires arbitrary cell fraction thresholds. The relationship between mCA observations vs. mutation rate and mCA mutation rate vs. proportion at >5% cell fraction are shown in Supplementary Fig. 27.

Supplementary Figure 27: a. Correlation between number of mCA observations and mCA mutation rate. mCAs with similar mutation rates can have very different prevalence, e.g. some mCAs with mutation rates of 1×10^{-9} per year were observed <10 times, whereas others with a similar mutation rate were observed >150 times. This is because the mCA prevalence is determined by both the mutation rate and the fitness effect, with higher fitness effects resulting in more mCAs expanding to detectable levels. **b. Correlation between proportion of mCA events at cell fraction >5% and mCA fitness effect.** Whilst mCAs with higher fitness effects are generally more likely to be found at higher cell fraction, mCAs observed in a similar proportion of people at >5% cell fraction can have very different fitness effects, e.g. some mCAs in which >80% were found at >5% cell fraction had fitness effects of ~12% per year, whereas others had fitness effects of >28% per year.

Major comments:

"The new Fig. 5 is very interesting, especially since it is trained on two mCA events for which there is a large amount of data. I think the possible explanation provided by the authors for why there might be two different mLOX events is inappropriate though. Rather than the inactive and the active X being lost as two different events, it could easily be the case that the inactive X is lost in two different cell types instead. I am very skeptical that the active X can be lost at any appreciable frequency as we don't see any other large autosome being lost in its entirety. I suppose it is possible that the active X is lost and the inactive X is reactivated, but this to me feels all a bit too speculative for a main figure.

Have the authors thought instead about the fact that, while two mLOY events in the same individual at low cell fractions would be interpreted as an mLOY event with cell fraction the sum of the two cell fractions, two mLOX events in the same individual at low cell fractions could be interpreted either as an mLOX event with cell fraction the sum of the two cell fractions or as an mLOX event with cell fraction the difference of the two cell fractions if the two events happened in two cells that inactivated a different X chromosome. This is due to the fact that mLOX at low cell fractions is entirely detected through BAF deviations and not through a shift in the LRR. Another possibility is that mLOX at low cell fractions is not mLOX at all and maybe it is related to an overall imbalance between cells that inactivated one X chromosome and cells that inactivated the other X chromosome, which is known to increase with age. In short, while I find the observed differential dynamic between mLOX and mLOY as one of the most fascinating aspects of the manuscript now, I would be careful about speculating what the cause is without supporting data"

response 2

We thank the reviewer for raising the interesting point about the technical difficulty in distinguishing two or more mLOY or mLOX events in the same individual at low cell fraction. To address this, we have added a section in the supplement (Supplementary Note 7) (see below) where we have performed an analysis to test whether this could result in the qualitative features observed in the mLOX and mLOY distributions. We show that the effects described by the reviewer would result in features in the distributions that are not observed in the data.

addition to supplement: Supplementary Note 7:**Distribution of cell fractions for high mutation rate events.**

A key difference between allosomal and autosomal mCAs is that we infer both mLOY and mLOX to occur at a high mutation rate, such that the product $\theta = N\tau\mu$ may no longer be a very small parameter ($\theta \ll 1$). In this regime it is likely that multiple independent allosomal mCA events expand at the same time within the same individual. Because of the way in which mCA cell fractions are called, this results in mLOY cell fraction being recorded as the sum of the independent cell fractions in that individual. For mLOX, the cell fraction recorded would be the absolute difference between the sum of the independent mLOX events affecting the maternal X-chromosome and the sum of the independent mLOY events affecting the paternal X-chromosome. To test whether these effects could result in the qualitative features observed in the mLOY and mLOX distributions in the UK Biobank data we simulated mCA calls at 3 different mutation rates ($\mu = 1 \times 10^{-6}$, $\mu = 1 \times 10^{-5}$ and $\mu = 1 \times 10^{-4}$), corresponding to $\theta = 0.1$, $\theta = 1$ and $\theta = 10$. We compared the simulated distribution of mCA cell fractions to the analytical expression (eq. 1) for the distribution of the sum of exponentially growing clones fed at a constant rate³¹.

$$\rho(l) = \frac{1}{\Gamma(\theta)} \times \frac{e^{-\frac{e^l}{\phi}/(1-e^l)}}{\left(\frac{e^l}{\phi}/(1-e^l)\right)^{1-\theta}} \times \frac{e^l}{(1-e^l)^2} \times \frac{1}{\phi} \quad \text{where } \theta = N\tau\mu \quad \text{and } \phi = \frac{e^{st} - 1}{N\tau s} \quad (1)$$

We considered whether the features in the distributions we observed in Fig. 5 may result from high mutation rate to mLOY and mLOX and the inability to distinguish independent events from the sum (mLOY) or the difference (mLOX). However, as can be seen in Supplementary Fig. 26, these effects produce features that are inconsistent with the features we observed in the real data, suggesting that these effects are not contributing. Multiple independent indistinguishable clones contributing significantly to the sum is also inconsistent with the mutation rates we infer for mLOY and mLOX where we estimate that the value of $\theta \sim 0.1$. In this parameter regime we do not expect these effects to play a significant role in shaping the distribution, as can be seen the first column of Supplementary Fig. 26.

addition to supplement: Supplementary Note 7: continued...

Supplementary Figure 26: Simulated mLOX calls with 2 different mutation rates and fitness effects. **a.** Simulated mLOX calls were generated at a high rate but conferring a low fitness (μ_1, s_1) or at a low rate but conferring a high fitness (μ_2, s_2). The cell fraction probability density histogram is shown for these mLOX calls (datapoints, $n = 38,198$ mLOX calls). Parameter estimation, using a maximum likelihood approach to fit a single theory distribution (solid line) with a single fitness effect (s) and single mutation rate (μ), is shown. Datapoints are presented as mean values \pm SEM. The white cross on the maximum likelihood heatmap marks the most likely inferred s and μ . **b.** Parameter estimation, using a maximum likelihood approach to fit a distribution (solid blue line) made up of 2 distinct mLOX events: one with a high mutation rate (μ_1), but low fitness (s_1) (solid grey line) and another with a low mutation rate (μ_2), but high fitness (s_2) (dashed grey line). Vertical black dashed lines show the cell fraction split points used for fitting the lower and upper cell fraction datapoints to the theory distribution. Datapoints are presented as mean values \pm SEM. The white cross on the maximum likelihood heatmaps marks the most likely inferred s_1, μ_1, s_2 and μ_2 . $n = 38,198$ mLOX calls.

With regards to our hypothesis for why there might be two different mLOX events (loss of the inactive vs. active X-chromosome), there is evidence to suggest that it is possible to lose the active X-chromosome (Machiela et al 2016). Nonetheless, we have now made it clear in the main text that there are other possible mechanisms that could explain two distinct fitness effects (changes highlighted in blue below).

amendment to main text: Discussion:

Because mLOY and mLOX are common events in blood, they provide good statistical power for testing our framework's predictions. These events demonstrate why prevalence, total number of events detected above technical sensitivity limits across all individuals, can be a misleading statistic (Supplementary Note 8). Our framework reveals that while mLOY is more prevalent than mLOX, the overall mutation rate to mLOX is higher, but it is less prevalent because it confers a modest fitness effect. The behaviour of mLOX is further demonstration of why our framework is an important null model for understanding the fitness effects and mutation rates of mosaic events. mLOX provides the strongest evidence for deviating from this null model and is more consistent with a model in which mLOX can confer two distinct fitness effects which occur at two different rates. While we do not have direct evidence for the cause of this, we speculate that it could be related to the fact that the X-chromosome that is lost will either be 'active' (X_a) or 'inactive' (X_i), however other explanations for two distinct fitness effects are possible. Loss of either the 'active' or 'inactive' X could plausibly result in two distinct effects occurring at two different rates. Previous studies have shown that high cell fraction mLOX typically results from the loss of the 'inactive' X-chromosome (X_i)³⁰, suggesting that this could be the event that drives the higher fitness effect (Fig. 5b). More generally, these examples demonstrate how quantitative features in the distribution of cell fractions, and their evolution with age, can reveal insights above the underlying biology.

Minor comments:

"ACKNOWLEDGMENTS: mLOY and mLOX calls are available by request via application to UK Biobank (Return 3094) (<https://www.ukbiobank.ac.uk>) <- the resource at <https://biobank.ndph.ox.ac.uk/ukb/dset.cgi?id=3094> includes both autosomal and sex chromosome mCAs, so it is not clear to me why only mLOY and mLOX are mentioned"

response 3

We have now appended the Data Availability statement as follows:

Data Availability statement:

The mCA calls used in our analysis are available from UK Biobank (Return 3094), via an application process described at <http://www.ukbiobank.ac.uk/using-the-resource/>. Autosomal mCA calls are also available, in anonymized form, on the Blundell lab Github page (<https://github.com/blundelllab/mCA-mutation-rates-fitness-consequences>) (DOI: 10.5281/zenodo.8097406) and from Loh, P-R., Genovese G. & McCarroll S. Monogenic and polygenic inheritance become instruments for clonal selection, Supplementary Data: <https://doi.org/10.1038/s41586-020-2430-6> (2020).

Final Decision Letter:

28th Jul 2023

Dear Jamie,

I am delighted to say that your manuscript "Mutation rates and fitness consequences of mosaic chromosomal alterations in blood." has been accepted for publication in an upcoming issue of Nature Genetics.

Your paper will be published online after we receive your corrections and will appear in print in the next available issue. You can find out your date of online publication by contacting the Nature Press Office (press@nature.com) after sending your e-proof corrections. Now is the time to inform your Public Relations or Press Office about your paper, as they might be interested in promoting its publication. This will allow them time to prepare an accurate and satisfactory press release. Include your manuscript tracking number (NG-A60886R1) and the name of the journal, which they will need when they contact our Press Office.

Please note that *Nature Genetics* is a Transformative Journal (TJ). Authors may publish their research with us through the traditional subscription access route or make their paper immediately open access through payment of an article-processing charge (APC). Authors will not be required to make a final decision about access to their article until it has been accepted. [Find out more about Transformative Journals](https://www.springernature.com/gp/open-research/transformative-journals)

Authors may need to take specific actions to achieve [compliance](https://www.springernature.com/gp/open-research/funding/policy-compliance-faqs) with funder and institutional open access mandates. If your research is supported by a funder that requires immediate open access (e.g. according to [Plan S principles](https://www.springernature.com/gp/open-research/plan-s-compliance)) then you should select the gold OA route, and we will direct you to the compliant route where possible. For authors selecting the subscription publication route, the journal's standard licensing terms will need to be accepted, including <https://www.nature.com/nature-portfolio/editorial-policies/self-archiving-and-license-to-publish>. Those licensing terms will supersede any other terms that the author or any third party may assert apply to any version of the manuscript.

Please note that Nature Portfolio offers an immediate open access option only for papers that were first submitted after 1 January, 2021.

If you have not already done so, we invite you to upload the step-by-step protocols used in this manuscript to the Protocols Exchange, part of our on-line web resource, natureprotocols.com. If you complete the upload by the time you receive your manuscript proofs, we can insert links in your article that lead directly to the protocol details. Your protocol will be made freely available upon publication of your paper. By participating in natureprotocols.com, you are enabling researchers to more readily reproduce or adapt the methodology you use. [Natureprotocols.com](https://natureprotocols.com) is fully searchable, providing your protocols and paper with increased utility and visibility. Please submit your protocol to <https://protocolexchange.researchsquare.com/>. After entering your nature.com username and password you will need to enter your manuscript number (NG-A60886R1). Further information can be found at <https://www.nature.com/nature-portfolio/editorial-policies/reporting-standards#protocols>

I am delighted that we've reached this stage. It's a lovely paper and I am excited to see it online and in print. Please pass on my congratulations to Caroline.

Have a great weekend.

Sincerely,

Safia Danovi
Editor
Nature Genetics